# Conic Blackwell Algorithm: Parameter-Free Convex-Concave Saddle-Point Solving

**Julien Grand-Clément**[*]
ISOM Department
HEC Paris
grand-clement@hec.fr

**Christian Kroer**
IEOR Department
Columbia University
christian.kroer@columbia.edu

## Abstract

We develop new parameter-free and scale-free algorithms for solving convex-concave saddle-point problems. Our results are based on a new simple regret minimizer, the Conic Blackwell Algorithm$^+$ (CBA$^+$), which attains $O(1/\sqrt{T})$ average regret. Intuitively, our approach generalizes to other decision sets of interest ideas from the Counterfactual Regret minimization (CFR$^+$) algorithm, which has very strong practical performance for solving sequential games on simplexes. We show how to implement CBA$^+$ for the simplex, $\ell_p$ norm balls, and ellipsoidal confidence regions in the simplex, and we present numerical experiments for solving matrix games and distributionally robust optimization problems. Our empirical results show that CBA$^+$ is a simple algorithm that outperforms state-of-the-art methods on synthetic data and real data instances, without the need for any choice of step sizes or other algorithmic parameters.

## 1  Introduction

We are interested in solving *saddle-point problems* (SPPs) of the form

$$\min_{\boldsymbol{x} \in \mathcal{X}} \max_{\boldsymbol{y} \in \mathcal{Y}} F(\boldsymbol{x}, \boldsymbol{y}), \tag{1}$$

where $\mathcal{X} \subset \mathbb{R}^n, \mathcal{Y} \subset \mathbb{R}^m$ are convex, compact sets, and $F : \mathcal{X} \times \mathcal{Y} \to \mathbb{R}$ is a subdifferentiable convex-concave function. Convex-concave SPPs arise in a number of practical problems. For example, the problem of computing a Nash equilibrium of a zero-sum game can be formulated as a convex-concave SPP, and this is the foundation of most methods for solving sequential zero-sum games [vS96, ZJBP07, TBJB15, KWKKS20]. They also arise in imaging [CP11], $\ell_\infty$-regression [ST18], Markov Decision Processes [Iye05, WKR13, ST18], and in distributionally robust optimization, where the max term represents the distributional uncertainty [ND16, BTHKM15]. In this paper we propose efficient, *parameter-free* algorithms for solving (1) in many settings, i.e., algorithms that do not require any tuning or choices of step sizes.

**Repeated game framework**  One way to solve convex-concave SPPs is by viewing the SPP as a repeated game between two players, where each step $t$ consists of one player choosing $\boldsymbol{x}_t \in \mathcal{X}$, the other player choosing $\boldsymbol{y}_t \in \mathcal{Y}$, and then the players observe the payoff $F(\boldsymbol{x}_t, \boldsymbol{y}_t)$. If each player employs a regret-minimization algorithm, then a well-known folk theorem says that the uniform average strategy generated by two regret minimizers repeatedly playing an SPP against each other converges to a solution to the SPP. We will call this the "repeated game framework" (see Section 2). There are already well-known algorithms for instantiating the above repeated game framework

---

[*]Julien Grand-Clément acknowledges the financial support of Hi! Paris for this project.

35th Conference on Neural Information Processing Systems (NeurIPS 2021).

for (1). For example, one can employ the *online mirror descent* (OMD) algorithm, which generates iterates as follows for the first player (and similarly for the second player):

$$\boldsymbol{x}_{t+1} = \arg\min_{\boldsymbol{x} \in \mathcal{X}} \langle \eta \boldsymbol{f}_t, \boldsymbol{x} \rangle + D(\boldsymbol{x} \| \boldsymbol{x}_t), \tag{2}$$

where $\boldsymbol{f}_t \in \partial_{\boldsymbol{x}} F(\boldsymbol{x}_t, \boldsymbol{y}_t)$ ($\partial_{\boldsymbol{x}}$ denotes here the set of subgradients as regards the variable $\boldsymbol{x}$), $D(\cdot \| \cdot)$ is a Bregman divergence which measures distance between pairs of points, and $\eta > 0$ is an appropriate step size. By choosing $D$ appropriately for $\mathcal{X}$, the update step (2) becomes efficient, and one can achieve an overall regret on the order of $O(\sqrt{T})$ after $T$ iterations. This regret can be achieved either by choosing a fixed step size $\eta = \alpha/L\sqrt{T}$, where $L$ is an upper bound on the norms of the subgradients visited $(\boldsymbol{f}_t)_{t \geq 0}$, or by choosing adaptive step sizes $\eta_t = \alpha/\sqrt{t}$, for $\alpha > 0$. This is problematic, as 1) the upper bound $L$ may be hard to obtain in many applications and may be too conservative in practice, and 2) adequately tuning the parameter $\alpha$ can be time- and resource-consuming, and even practically infeasible for very large instances, since we won't know if the step size will cause a slow convergence or a divergence until late in the optimization process. This is not just a theoretical issue, as we highlight in our numerical experiments (Section 4) and in the appendices (Appendices F). Similar results and challenges hold for the popular *follow the regularized leader* (FTRL) algorithm (see Appendix F).

The above issues can be circumvented by employing *adaptive* variants of OMD or FTRL, which lead to parameter- and scale-free algorithms that estimate the parameters through the observed subgradients, e.g., AdaHedge for the simplex setting [DRVEGK14] or AdaFTRL for general compact convex decisions sets [OP15]. Yet these adaptive variants have not seen practical adoption in large-sale game-solving, where regret-matching variants are preferred (see the next paragraph). As we show in our experiments, adaptive variants of OMD and FTRL perform much worse than our proposed algorithms. While these adaptive algorithms are referred to as parameter-free, this is only true in the sense that they are able to learn the necessary parameters. Our algorithm is parameter-free in the stronger sense that there are no parameters that even require learning. Formalizing this difference may be one interesting avenue for explaining the performance discrepancy on saddle-point problems.

**Regret Matching**  In this paper, we introduce alternative regret-minimization schemes for instantiating the above framework. Our work is motivated by recent advances on solving large-scale zero-sum sequential games. In the zero-sum sequential game setting, $\mathcal{X}$ and $\mathcal{Y}$ are simplexes, the objective function becomes $F(\boldsymbol{x}, \boldsymbol{y}) = \langle \boldsymbol{x}, \boldsymbol{A}\boldsymbol{y} \rangle$, and thus (1) reduces to a *bilinear* SPP. Based on this bilinear SPP formulation, the best practical methods for solving large-scale sequential games use the repeated game framework, where each player minimizes regret via some variant of *counterfactual regret minimization* (CFR, [ZJBP07]). Variants of CFR were used in every recent poker AI challenge, where poker AIs beat human poker players [BBJT15, MSB+17, BS18, BS19]. The CFR framework itself is a decomposition of the overall regret of the bilinear SPP into local regrets at each decision point in a sequential game [FKS19a]. The key to the practical performance of CFR-based algorithms seems to be three ingredients (beyond the CFR decomposition itself): (1) a particular regret minimizer called *regret matching*[+] (RM[+]) [TBJB15] which is employed at each decision point, (2) aggressive iterate averaging schemes that put greater weight on recent iterates (e.g. *linear averaging*, which weights iterate at period $t$ by $2t/T(T+1)$), and (3) an alternation scheme where the updates of the repeated game framework are performed in an asymmetric fashion. The CFR framework itself is specific to sequential bilinear games on simplexes, but these last three ingredients could potentially be generalized to other problems of the form (1). That is the starting point of the present paper.

The most challenging aspect of generalizing the above ingredients is that RM[+] is specifically designed for minimizing regret over a simplex. However, many problems of the form (1) have convex sets $\mathcal{X}, \mathcal{Y}$ that are not simplexes, e.g., box constraints or norm-balls for distributionally robust optimization [BTHKM15]. In principle, regret matching arises from a general theory called *Blackwell approachability* [Bla56, HMC00], and similar constructions can be envisioned for other convex sets. However, in practice the literature has only focused on developing concrete implementable instantiations of Blackwell approachability for simplexes. A notable deviation from this is the work of [ABH11], who showed a general reduction between regret minimization over general convex compact sets and Blackwell approachability. However, their general reduction still does not yield a practically implementable algorithm: among other things, their reduction relies on certain black-box projections that are not always efficient. We show how to implement these necessary projections for the setting where $\mathcal{X}$ and $\mathcal{Y}$ are simplexes, $\ell_p$ balls, and intersections of the $\ell_2$ ball with a hyperplane

(with a focus on the case where an $\ell_2$ ball is intersected with a simplex, which arises naturally as a confidence region). This yields an algorithm which we will refer to as the *conic Blackwell algorithm* (CBA), which is similar in spirit to the regret matching algorithm, but crucially generalizes to other decision sets. Motivated by the practical performance of RM⁺, we construct a variant of CBA which uses a thresholding operation similar to the one employed by RM⁺. We call this algorithm CBA⁺.

**Our contributions**   We introduce CBA⁺, a parameter-free algorithm which achieves $O(\sqrt{T})$ regret in the worst-case and generalizes the strong performances of RM⁺ for bilinear, simplex saddle-points solving to other more general settings. A major selling point for CBA⁺ is that it does not require any step size choices. Instead, the algorithm implicitly adjusts to the structure of the domains and losses by being instantiations of Blackwell's approachability algorithm. After developing the CBA⁺ algorithm, we then develop analogues of another crucial components for large-scale game solving. In particular, we prove a generalization of the folk theorem for the repeated game framework for solving (1), which allows us to incorporate polynomial averaging schemes such as linear averaging. We then show that CBA⁺ is compatible with linear averaging on the iterates. This mirrors the case of RM and RM⁺, where only RM⁺ is compatible with linear averaging on the iterates. We also show that both CBA and CBA⁺ are compatible with polynomial averaging when simultaneously performed on the regrets and the iterates. Combining all these ingredients, we arrive at a new class of algorithms for solving convex-concave SPPs. As long as efficient projection operations can be performed (which we show for several practical domains, including the simplex, $\ell_p$ balls and confidence regions in the simplex), one can apply the repeated game framework on (1), where one can use either CBA or CBA⁺ as a regret minimizer for $\mathcal{X}$ and $\mathcal{Y}$ along with polynomial averaging on the generated iterates to solve (1) at a rate of $O\left(1/\sqrt{T}\right)$.

We highlight the practical efficacy of our algorithmic framework on several domains. First, we solve two-player zero-sum matrix games and extensive-form games, where RM⁺ regret minimizer combined with linear averaging and alternation, and CFR⁺, lead to very strong practical algorithms [TBJB15]. We find that CBA⁺ combined with linear averaging and alternation leads to a comparable performance in terms of the iteration complexity, and may even slightly outperform RM⁺ and CFR⁺. On this simplex setting, we also find that CBA⁺ outperforms both AdaHedge and AdaFTRL. Second, we apply our approach to a setting where RM⁺ and CFR⁺ do not apply: distributionally robust empirical risk minimization (DR-ERM) problems. Across two classes of synthetic problems and four real data sets, we find that our algorithm based on CBA⁺ performs orders of magnitude better than online mirror descent and FTRL, as well as their optimistic variants, when using their theoretically-correct fixed step sizes. Even when considering adaptive step sizes, or fixed step sizes that are up to $10{,}000$ larger than those predicted by theory, our CBA⁺ algorithm performs better, with only a few cases of comparable performance (at step sizes that lead to divergence for some of the other non parameter-free methods). The fast practical performance of our algorithm, combined with its simplicity and the total lack of step sizes or parameters tuning, suggests that it should be seriously considered as a practical approach for solving convex-concave SPPs in various settings.

Finally, we make a brief note on accelerated methods. Our algorithms have a rate of convergence towards a saddle point of $O(1/\sqrt{T})$, similar to OMD and FTRL. In theory, it is possible to obtain a faster $O\left(1/T\right)$ rate of convergence when $F$ is differentiable with Lipschitz gradients, for example via mirror prox [Nem04] or other primal-dual algorithms [CP16]. However, our experimental results show that CBA⁺ is faster than optimistic variants of FTRL and OMD [SALS15], the latter being almost identical to the mirror prox algorithm, and both achieving $O(1/T)$ rate of convergence. A similar conclusion has been drawn in the context of sequential game solving, where the fastest $O(1/\sqrt{T})$ CFR-based algorithms have better practical performance than the theoretically-superior $O\left(1/T\right)$-rate methods [KWKKS20, KFS18]. In a similar vein, using *error-bound conditions*, it is possible to achieve a linear rate, e.g., when solving bilinear saddle-point problems over polyhedral decision sets using the extragradient method [Tse95] or optimistic gradient descent-ascent [WLZL20]. However, these linear rates rely on unknown constants, and may not be indicative of practical performance.

# 2 Game setup and Blackwell Approachability

As stated in Section 1, we will solve (1) using a repeated game framework. The first player chooses strategies from $\mathcal{X}$ in order to minimize the sequence of payoffs in the repeated game, while the second player chooses strategies from $\mathcal{Y}$ in order to maximize payoffs. There are $T$ iterations with indices $t = 1, \ldots, T$. In this framework, each iteration $t$ consists of the following steps:

1. Each player chooses strategies $\boldsymbol{x}_t \in \mathcal{X}, \boldsymbol{y}_t \in \mathcal{Y}$
2. First player observes $\boldsymbol{f_t} \in \partial_{\boldsymbol{x}} F(\boldsymbol{x}_t, \boldsymbol{y}_t)$ and uses $\boldsymbol{f_t}$ when computing the next strategy
3. Second player observes $\boldsymbol{g_t} \in \partial_{\boldsymbol{y}} F(\boldsymbol{x}_t, \boldsymbol{y}_t)$ and uses $\boldsymbol{g_t}$ when computing the next strategy

The goal of each player is to minimize their regret $R_{T,\boldsymbol{x}}, R_{T,\boldsymbol{y}}$ across the $T$ iterations:

$$R_{T,\boldsymbol{x}} = \sum_{t=1}^{T} \langle \boldsymbol{f}_t, \boldsymbol{x}_t \rangle - \min_{\boldsymbol{x} \in \mathcal{X}} \sum_{t=1}^{T} \langle \boldsymbol{f}_t, \boldsymbol{x} \rangle, \quad R_{T,\boldsymbol{y}} = \max_{\boldsymbol{y} \in \mathcal{Y}} \sum_{t=1}^{T} \langle \boldsymbol{g}_t, \boldsymbol{y} \rangle - \sum_{t=1}^{T} \langle \boldsymbol{g}_t, \boldsymbol{y}_t \rangle.$$

The reason this repeated game framework leads to a solution to the SPP problem (1) is the following folk theorem. Relying on $F$ being convex-concave and subdifferentiable, it connects the regret incurred by each player to the duality gap in (1).

**Theorem 2.1** (Theorem 1, [Kro20]). *Let* $(\bar{\boldsymbol{x}}_T, \bar{\boldsymbol{y}}_T) = \frac{1}{T} \sum_{t=1}^{T} (\boldsymbol{x}_t, \boldsymbol{y}_t)$ *for any* $(\boldsymbol{x}_t)_{t \geq 1}, (\boldsymbol{y}_t)_{t \geq 1}$. *Then*

$$\max_{\boldsymbol{y} \in \mathcal{Y}} F(\bar{\boldsymbol{x}}_T, \boldsymbol{y}) - \min_{\boldsymbol{x} \in \mathcal{X}} F(\boldsymbol{x}, \bar{\boldsymbol{y}}_T) \leq (R_{T,\boldsymbol{x}} + R_{T,\boldsymbol{y}})/T.$$

Therefore, when each player runs a regret minimizer that guarantees regret on the order of $O(\sqrt{T})$, $(\bar{\boldsymbol{x}}_T, \bar{\boldsymbol{y}}_T)_{T \geq 0}$ converges to a solution to (1) at a rate of $O\left(1/\sqrt{T}\right)$. Later we will show a generalization of Theorem 2.1 that will allow us to incorporate more aggressive averaging schemes that put additional weight on the later iterates. Given the repeated game framework, the next question becomes which algorithms to employ in order to minimize regret for each player. As mentioned in Section 1, for zero-sum games, variants of regret matching are used in practice.

**Blackwell Approachability**   Regret matching arises from the *Blackwell approachability* framework [Bla56]. In Blackwell approachability, a decision maker repeatedly takes decisions $\boldsymbol{x}_t$ from some convex decision set $\mathcal{X}$ (this set plays the same role as $\mathcal{X}$ or $\mathcal{Y}$ in (1)). After taking decision $\boldsymbol{x}_t$ the player observes a vector-valued affine payoff function $\boldsymbol{u}_t(\boldsymbol{x}) \in \mathbb{R}^n$. The goal for the decision maker is to force the average payoff $\frac{1}{t} \sum_{\tau=1}^{t} \boldsymbol{u}_\tau(\boldsymbol{x}_\tau)$ to approach some convex target $\mathcal{S}$. Blackwell proved that a convex target set $\mathcal{S}$ can be approached if and only if for every halfspace $\mathcal{H} \supseteq \mathcal{S}$, there exists $\boldsymbol{x} \in \mathcal{X}$ such that for every possible payoff function $\boldsymbol{u}(\cdot)$, $\boldsymbol{u}(\boldsymbol{x})$ is guaranteed to lie in $\mathcal{H}$. The action $\boldsymbol{x}$ is said to *force* $\mathcal{H}$. Blackwell's proof is via an algorithm: at iteration $t$, his algorithm projects the average payoff $\bar{\boldsymbol{u}} = \frac{1}{t-1} \sum_{\tau=1}^{t-1} \boldsymbol{u}_\tau(\boldsymbol{x}_\tau)$ onto $\mathcal{S}$, and then the decision maker chooses an action $\boldsymbol{x}_t$ that forces the tangent halfspace to $\mathcal{S}$ generated by the normal $\bar{\boldsymbol{u}} - \pi_{\mathcal{S}}(\bar{\boldsymbol{u}})$, where $\pi_{\mathcal{S}}(\bar{\boldsymbol{u}})$ is the orthogonal projection of $\bar{\boldsymbol{u}}$ onto $\mathcal{S}$. We call this algorithm *Blackwell's algorithm*; it approaches $\mathcal{S}$ at a rate of $O(1/\sqrt{T})$. It is important to note here that Blackwell's algorithm is rather a meta-algorithm than a concrete algorithm. Even within the context of Blackwell's approachability problem, one needs to devise a way to compute the forcing actions needed at each iteration, i.e., to compute $\pi_{\mathcal{S}}(\bar{\boldsymbol{u}})$.

**Details on Regret Matching**   Regret matching arises by instantiating Blackwell approachability with the decision space $\mathcal{X}$ equal to the simplex $\Delta(n)$, the target set $\mathcal{S}$ equal to the nonpositive orthant $\mathbb{R}^n_-$, and the vector-valued payoff function $\boldsymbol{u}_t(\boldsymbol{x}_t) = \boldsymbol{f}_t - \langle \boldsymbol{f}_t, \boldsymbol{x}_t \rangle \boldsymbol{e}$ equal to the regret associated to each of the $n$ actions (which correspond to the corners of $\Delta(n)$). Here $\boldsymbol{e} \in \mathbb{R}^n$ has one on every component. [HMC00] showed that with this setup, playing each action with probability proportional to its positive regret up to time $t$ satisfies the forcing condition needed in Blackwell's algorithm. Formally, regret matching (RM) keeps a running sum $\boldsymbol{r}_t = \sum_{\tau=1}^{t} (\boldsymbol{f}_\tau - \langle \boldsymbol{f}_\tau, \boldsymbol{x}_\tau \rangle \boldsymbol{e})$, and then action $i$ is played with probability $\boldsymbol{x}_{t+1,i} = [\boldsymbol{r}_{t,i}]^+ / \sum_{i=1}^{n} [\boldsymbol{r}_{t,i}]^+$, where $[\cdot]^+$ denotes thresholding at zero. By Blackwell's approachability theorem, this algorithm converges to zero average regret at a rate of $O(1/\sqrt{T})$. In zero-sum game-solving, it was discovered that a variant of

regret matching leads to extremely strong practical performance (but the same theoretical rate of convergence). In regret matching$^+$ (RM$^+$), the running sum is thresholded at zero at every iteration: $\boldsymbol{r}_t = [\boldsymbol{r}_{t-1} + \boldsymbol{f}_t - \langle \boldsymbol{f}_t, \boldsymbol{x}_t \rangle e]^+$, and then actions are again played proportional to $\boldsymbol{r}_t$. In the next section, we describe a more general class of regret-minimization algorithms based on Blackwell's algorithm for general sets $\mathcal{X}$, introduced in [ABH11]. Note that a similar construction of a general class of algorithms can be achieved through the *Lagrangian Hedging* framework of [Gor07]. It would be interesting to construct a CBA$^+$-like algorithm and efficient projection approaches for this framework as well.

# 3   Conic Blackwell Algorithm

We present the Conic Blackwell Algorithm Plus (CBA$^+$), a no-regret algorithm which uses a variation of Blackwell's approachability procedure [Bla56] to perform regret minimization on general convex compact decision sets $\mathcal{X}$. We will assume that losses are coming from a bounded set; this occurs, for example, if there exists $L_x, L_y$ (that we do not need to know), such that

$$\|\boldsymbol{f}\| \leq L_x, \|\boldsymbol{g}\| \leq L_y, \ \forall \, \boldsymbol{x} \in \mathcal{X}, \boldsymbol{y} \in \mathcal{Y}, \forall \, \boldsymbol{f} \in \partial_{\boldsymbol{x}} F(\boldsymbol{x}, \boldsymbol{y}), \forall \, \boldsymbol{g} \in \partial_{\boldsymbol{y}} F(\boldsymbol{x}, \boldsymbol{y}). \tag{3}$$

CBA$^+$ is best understood as a combination of two steps. The first is the basic CBA algorithm, derived from Blackwell's algorithm, which we describe next. To convert Blackwell's algorithm to a regret minimizer on $\mathcal{X}$, we use the reduction from [ABH11], which considers the conic hull $\mathcal{C} = \text{cone}(\{\kappa\} \times \mathcal{X})$ where $\kappa = \max_{\boldsymbol{x} \in \mathcal{X}} \|\boldsymbol{x}\|_2$. The Blackwell approachability problem is then instantiated with $\mathcal{X}$ as the decision set, target set equal to the polar $\mathcal{C}^\circ = \{\boldsymbol{z} : \langle \boldsymbol{z}, \hat{\boldsymbol{z}} \rangle \leq 0, \forall \hat{\boldsymbol{z}} \in \mathcal{C}\}$ of $\mathcal{C}$, and payoff vectors $(\langle \boldsymbol{f}_t, \boldsymbol{x}_t \rangle, -\boldsymbol{f}_t)$. The conic Blackwell algorithm (CBA) is implemented by projecting the average payoff vector onto $\mathcal{C}$, calling this projection $\alpha(\kappa, \boldsymbol{x})$ with $\alpha \geq 0$ and $\boldsymbol{x} \in \mathcal{X}$, and playing the action $\boldsymbol{x}$.

The second step in CBA$^+$ is to modify CBA to make it analogous to RM$^+$ rather than to RM. To do this, the algorithm does not keep track of the average payoff vector. Instead, we keep a running aggregation of the payoffs, where we always add the newest payoff to the aggregate, and then project the aggregate onto $\mathcal{C}$. More concretely, pseudocode for CBA$^+$ is given in Algorithm 1. This pseudocode relies on two functions: CHOOSEDECISION$_{\text{CBA+}}$ : $\mathbb{R}^{n+1} \to \mathbb{R}^n$, which maps the aggregate payoff vector $\boldsymbol{u}_t$ to a decision in $\mathcal{X}$, and UPDATEPAYOFF$_{\text{CBA+}}$ which controls how we aggregate payoffs. Given an aggregate payoff vector $\boldsymbol{u} = (\tilde{u}, \hat{\boldsymbol{u}}) \in \mathbb{R} \times \mathbb{R}^n$, we have

$$\text{CHOOSEDECISION}_{\text{CBA+}}(\boldsymbol{u}) = (\kappa/\tilde{u})\hat{\boldsymbol{u}}.$$

If $\tilde{u} = 0$, we just let CHOOSEDECISION$_{\text{CBA+}}(\boldsymbol{u}) = \boldsymbol{x}_0$ for some chosen $\boldsymbol{x}_0 \in \mathcal{X}$.

The function UPDATEPAYOFF$_{\text{CBA+}}$ is implemented by adding the most recent payoff to the aggregate payoffs, and then projecting onto $\mathcal{C}$. More formally, it is defined as

$$\text{UPDATEPAYOFF}_{\text{CBA+}}(\boldsymbol{u}, \boldsymbol{x}, \boldsymbol{f}, \omega, S) = \pi_{\mathcal{C}} \left( \frac{S}{S+\omega} \boldsymbol{u} + \frac{\omega}{S+\omega} \left( \langle \boldsymbol{f}, \boldsymbol{x} \rangle/\kappa, -\boldsymbol{f} \right) \right),$$

where $\omega$ is the weight assigned to the most recent payoff and $S$ the weight assigned to the previous aggregate payoff $\boldsymbol{u}$. Because of the projection step in UPDATEPAYOFF$_{\text{CBA+}}$, we always have $\boldsymbol{u} \in \mathcal{C}$, which in turn guarantees that CHOOSEDECISION$_{\text{CBA+}}(\boldsymbol{u}) \in \mathcal{X}$, since $\mathcal{C} = \text{cone}(\{\kappa\} \times \mathcal{X})$.

Let us give some intuition on the effect of projection onto $\mathcal{C}$. In a geometric sense, it is easier to visualize things in $\mathbb{R}^2$ with $\mathcal{C} = \mathbb{R}^2_+$ and $\mathcal{C}^\circ = \mathbb{R}^2_-$. The projection on $\mathcal{C}$ moves the vector along the edges of $\mathcal{C}^\circ$, maintaining the distance to $\mathcal{C}^\circ$ and moving toward the vector $\boldsymbol{0}$. This is illustrated in Figure 5 in Appendix B.1. From a game-theoretic standpoint, the projection on $\mathcal{C} = \mathbb{R}^2_+$ eliminates the components of the payoffs that are negative. It enables CBA+ to be less pessimistic than CBA, which may accumulate negative payoffs on actions for a long time and never resets the components of the aggregated payoff to 0, leading to some actions being chosen less frequently.

We will see in the next section that RM$^+$ is related to CBA$^+$ but replaces the exact projection step $\pi_{\mathcal{C}}(\boldsymbol{u})$ in UPDATEPAYOFF$_{\text{CBA+}}$ by a suboptimal solution to the projection problem. Let us also note the difference between CBA$^+$ and the algorithm introduced in [ABH11], which we have called

---

**Algorithm 1** Conic Blackwell Algorithm Plus (CBA$^+$)

---

1: **Input** A convex, compact set $\mathcal{X} \subset \mathbb{R}^n$, $\kappa = \max\{\|\boldsymbol{x}\|_2 \mid \boldsymbol{x} \in \mathcal{X}\}$.
2: **Algorithm parameters** Weights $(\omega_\tau)_{\tau \geq 1} \in \mathbb{R}^{\mathbb{N}}$.
3: **Initialization** $t = 1$, $\boldsymbol{x}_1 \in \mathcal{X}$.
4: Observe $\boldsymbol{f}_1$ then set $\boldsymbol{u}_1 = (\langle \boldsymbol{f}_1, \boldsymbol{x}_1 \rangle / \kappa, -\boldsymbol{f}_1) \in \mathbb{R} \times \mathbb{R}^n$.
5: **for** $t \geq 1$ **do**
6:     Choose $\boldsymbol{x}_{t+1} = \text{CHOOSEDECISION}_{\text{CBA}^+}(\boldsymbol{u}_t)$.
7:     Observe the loss $\boldsymbol{f}_{t+1} \in \mathbb{R}^n$.
8:     Update $\boldsymbol{u}_{t+1} = \text{UPDATEPAYOFF}_{\text{CBA}^+}(\boldsymbol{u}_t, \boldsymbol{x}_{t+1}, \boldsymbol{f}_{t+1}, \omega_{t+1}, \sum_{\tau=1}^t \omega_\tau)$.
9:     Increment $t \leftarrow t + 1$.

---

CBA. CBA uses different UPDATEPAYOFF and CHOOSEDECISION functions. In CBA the payoff update is defined as

$$\text{UPDATEPAYOFF}_{\text{CBA}}(\boldsymbol{u}, \boldsymbol{x}, \boldsymbol{f}, \omega, S) = \frac{S}{S + \omega} \boldsymbol{u} + \frac{\omega}{S + \omega} \left( \langle \boldsymbol{f}, \boldsymbol{x} \rangle / \kappa, -\boldsymbol{f} \right).$$

Note in particular the lack of projection as compared to CBA$^+$, analogous to the difference between RM and RM$^+$. The CHOOSEDECISION$_{\text{CBA}}$ function then requires a projection onto $\mathcal{C}$:

$$\text{CHOOSEDECISION}_{\text{CBA}}(\boldsymbol{u}) = \text{CHOOSEDECISION}_{\text{CBA}^+}(\pi_{\mathcal{C}}(\boldsymbol{u})).$$

Based upon the analysis in [Bla56], [ABH11] show that CBA with uniform weights (both on payoffs and decisions) guarantees $O(1/\sqrt{T})$ average regret. The difference between CBA$^+$ and CBA is similar to the difference between the RM and RM$^+$ algorithms. In practice, RM$^+$ performs significantly better than RM for solving matrix games, when combined with *linear averaging* on the decisions (as opposed to the uniform averaging used in Theorem 2.1). In the next theorem, we show that CBA$^+$ is compatible with linear averaging on decisions only. We present a detailed proof in Appendix B.

**Theorem 3.1.** *Consider* $(\boldsymbol{x}_t)_{t \geq 0}$ *generated by* CBA$^+$ *with uniform weights:* $\omega_\tau = 1, \forall \, \tau \geq 1$. *Let* $L = \max\{\|\boldsymbol{f}_t\|_2 \mid t \geq 1\}$ *and* $\kappa = \max\{\|\boldsymbol{x}\|_2 \mid \boldsymbol{x} \in \mathcal{X}\}$. *Then*

$$\frac{\sum_{t=1}^T t \langle \boldsymbol{f}_t, \boldsymbol{x}_t \rangle - \min_{\boldsymbol{x} \in \mathcal{X}} \sum_{t=1}^T t \langle \boldsymbol{f}_t, \boldsymbol{x} \rangle}{T(T+1)} = O\left(\kappa L / \sqrt{T}\right).$$

Note that in Theorem 3.1, we have *uniform* weights on the sequence of payoffs $(\boldsymbol{u}_t)_{t \geq 0}$, but *linearly increasing* weights on the sequence of decisions. The proof relies on properties specific to CBA$^+$, and it does not extend to CBA. Numerically it also helps CBA$^+$ but not CBA. In Appendix B, we show that both CBA and CBA$^+$ achieve $O\left(\kappa L / \sqrt{T}\right)$ convergence rates when using a weighted average on *both* the decisions and the payoffs (Theorems B.2-B.3). In practice, using linear averaging only on the decisions, as in Theorem 3.1, performs vastly better than linear averaging on both decisions and payoffs. We present empirical evidence of this in Appendix B.

We can compare the $O\left(\kappa L / \sqrt{T}\right)$ average regret for CBA$^+$ with the $O\left(\Omega L / \sqrt{T}\right)$ average regret for OMD [NY83, BTN01] and FTRL [AHR09, McM11], where $\Omega = \max\{\|\boldsymbol{x} - \boldsymbol{x}'\|_2 | \boldsymbol{x}, \boldsymbol{x}' \in \mathcal{X}\}$. We can always recenter $\mathcal{X}$ to contain $\boldsymbol{0}$, in which case the bounds for OMD/FTRL and CBA$^+$ are equivalent since $\kappa \leq \Omega \leq 2\kappa$. Note that the bound on the average regret for Optimistic OMD (O-OMD, [CYL$^+$12]) and Optimistic FTRL (O-FTRL, [RS13]) is $O\left(\Omega^2 L / T\right)$ in the game setup, a priori better than the bound for CBA$^+$ as regards the number of iterations $T$. Nonetheless, we will see in Section 4 that the empirical performance of CBA$^+$ is better than that of $O(1/T)$ methods. A similar situation occurs for RM$^+$ compared to O-OMD and O-FTRL for solving poker games [FKS19b, KWKKS20].

The following theorem gives the convergence rate of CBA$^+$ for solving saddle-points (1), based on our convergence rate on the regret of each player (Theorem 3.1). The proof is in Appendix C.

**Theorem 3.2.** *Let* $(\bar{\boldsymbol{x}}_T, \bar{\boldsymbol{y}}_T) = 2 \sum_{t=1}^T t (\boldsymbol{x}_t, \boldsymbol{y}_t) / (T(T+1))$, *where* $(\boldsymbol{x}_t)_{t \geq 0}, (\boldsymbol{y}_t)_{t \geq 0}$ *are generated by the repeated game framework with* CBA$^+$ *with uniform weights:* $\omega_\tau = 1, \forall \, \tau \geq 1$. *Let* $L = \max\{L_x, L_y\}$ *defined in* (3) *and* $\kappa = \max\{\max\{\|\boldsymbol{x}\|_2, \|\boldsymbol{y}\|_2\} \mid \boldsymbol{x} \in \mathcal{X}, \boldsymbol{y} \in \mathcal{Y}\}$. *Then*

$$\max_{\boldsymbol{y} \in \mathcal{Y}} F(\bar{\boldsymbol{x}}_T, \boldsymbol{y}) - \min_{\boldsymbol{x} \in \mathcal{X}} F(\boldsymbol{x}, \bar{\boldsymbol{y}}_T) = O\left(\kappa L / \sqrt{T}\right).$$

## 3.1 Efficient implementations of CBA⁺

To obtain an implementation of CBA⁺ and CBA, we need to efficiently resolve the functions CHOOSEDECISION$_{\mathsf{CBA+}}$ and UPDATEPAYOFF$_{\mathsf{CBA+}}$. In particular, we need to compute $\pi_{\mathcal{C}}(\boldsymbol{u})$, the orthogonal projection of $\boldsymbol{u}$ onto the cone $\mathcal{C}$, where $\mathcal{C} = \text{cone}(\{\kappa\} \times \mathcal{X})$:

$$\pi_{\mathcal{C}}(\boldsymbol{u}) \in \arg\min_{\boldsymbol{y} \in \mathcal{C}} \|\boldsymbol{y} - \boldsymbol{u}\|_2^2. \tag{4}$$

Even for CBA this problem must be resolved, since [ABH11] did not study whether (4) can be efficiently solved. It turns out that (4) can be computed in closed-form or quasi closed-form for many decision sets $\mathcal{X}$ of interest. Interestingly, parts of the proofs rely on *Moreau's Decomposition Theorem* [CR13], which states that $\pi_{\mathcal{C}}(\boldsymbol{u})$ can be recovered from $\pi_{\mathcal{C}^\circ}(\boldsymbol{u})$ and vice-versa, because $\pi_{\mathcal{C}}(\boldsymbol{u}) + \pi_{\mathcal{C}^\circ}(\boldsymbol{u}) = \boldsymbol{u}$. We present the detailed complexity results and the proofs in Appendix D.

**Simplex**   $\mathcal{X} = \Delta(n)$ is the classical setting used for matrix games. Also, for extensive-form games, CFR decomposes the decision sets (treeplexes) into a set of regret minimization problems over the simplex [FKS19a]. Here, $n$ is the number of actions of a player and $\boldsymbol{x} \in \Delta(n)$ represents a randomized strategy. In this case, $\pi_{\mathcal{C}}(\boldsymbol{u})$ can be computed in $O(n \log(n))$. Note that RM and RM⁺ are obtained by choosing a suboptimal solution to (4), avoiding the $O(n \log(n))$ sorting operation, whereas CBA and CBA⁺ choose optimally (see Appendix D). Thus, RM and RM⁺ can be seen as approximate versions of CBA and CBA⁺, where (4) is solved approximatively at every iteration. In our numerical experiments, we will see that CBA⁺ slightly outperforms RM⁺ and CFR⁺ in terms of iteration count.

$\ell_p$ **balls**   This is when $\mathcal{X} = \{\boldsymbol{x} \in \mathbb{R}^n \mid ; \|\boldsymbol{x}\|_p \leq 1\}$ with $p \geq 1$ or $p = \infty$. This is of interest for instance in distributionally robust optimization [BTHKM15, ND16], $\ell_\infty$ regression [ST18] and saddle-point reformulation of Markov Decision Process [JS20]. For $p = 2$, we can compute $\pi_{\mathcal{C}}(\boldsymbol{u})$ in closed-form, i.e., in $O(n)$ arithmetic operations. For $p \in \{1, \infty\}$, we can compute $\pi_{\mathcal{C}}(\boldsymbol{u})$ in $O(n \log(n))$ arithmetic operations using a sorting algorithm.

**Ellipsoidal confidence region in the simplex**   Here, $\mathcal{X}$ is an *ellipsoidal subregion of the simplex*, defined as $\mathcal{X} = \{\boldsymbol{x} \in \Delta(n) \mid \|\boldsymbol{x} - \boldsymbol{x}_0\|_2 \leq \epsilon_x\}$. This type of decision set is widely used because it is associated with confidence regions when estimating a probability distribution from observed data [Iye05, BdHP19]. It can also be used in the Bellman update for robust Markov Decision Process [Iye05, WKR13, GGC18]. We also assume that the confidence region is "entirely contained in the simplex": $\{\boldsymbol{x} \in \mathbb{R}^n | \boldsymbol{x}^\top \boldsymbol{e} = 1\} \bigcap \{\boldsymbol{x} \in \mathbb{R}^n \mid \|\boldsymbol{x} - \boldsymbol{x}_0\|_2 \leq \epsilon_x\} \subseteq \Delta(n)$, to avoid degenerate components. In this case, using a change of basis we show that it is possible to compute $\pi_{\mathcal{C}}(\boldsymbol{u})$ in closed-form, i.e., in $O(n)$ arithmetic operations.

**Other potential sets of interests**   Other important decision sets include sets based on Kullback-Leibler divergence $\{\boldsymbol{x} \in \Delta(n) \mid KL(\boldsymbol{x}, \boldsymbol{x}_0) \leq \epsilon_x\}$, or, more generally, $\phi$-divergence [BTDHDW⁺13]. For these sets, we did not find a closed-form solution to the projection problem (4). Still, as long as the domain $\mathcal{X}$ is a convex set, computing $\pi_{\mathcal{C}}(\boldsymbol{u})$ remains a convex problem, and it can be solved efficiently with solvers, although this results in a slower algorithm than with closed-form computations of $\pi_{\mathcal{C}}(\boldsymbol{u})$.

# 4   Numerical experiments

In this section we investigate the practical performances of our algorithms on several instances of saddle-point problems. We start by comparing CBA⁺ with RM⁺ in the matrix and extensive-form games setting. We then turn to comparing our algorithms on instances from the distributionally robust optimization literature. The code for all experiments is available in the supplemental material.

## 4.1   Matrix games on the simplex and Extensive-Form Games

Since the motivation for CBA⁺ is to obtain the strong empirical performances of RM⁺ and CFR⁺ on other decision sets than the simplex, we start by checking that CBA⁺ indeed provides comparable

performance on simplex settings. We compare these methods on matrix games

$$\min_{\boldsymbol{x}\in\Delta(n)} \max_{\boldsymbol{y}\in\Delta(m)} \langle\boldsymbol{x}, \boldsymbol{A}\boldsymbol{y}\rangle,$$

where $\boldsymbol{A}$ is the matrix of payoff, and on extensive-form games (EFGs). EFGs can also be written as SPPs with bilinear objective and $\mathcal{X}, \mathcal{Y}$ polytopes encoding the players' space of sequential strategies [vS96]. EFGs can be solved via simplex-based regret minimization by using the counterfactual regret minimization (CFR) framework to decompose regrets into local regrets at each simplex. Explaining CFR is beyond the scope of this work; we point the reader to [ZJBP07] or newer explanations [FKS19c, FKS19a]. For matrix games, we generate 70 synthetic 10-dimensional matrix games with $A_{ij} \sim U[0,1]$ and compare the most efficient algorithms for matrix games with linear averaging: CBA+ and RM+. We also compare with two other scale-free no-regret algorithms, AdaHedge [DRVEGK14] and AdaFTRL [OP15]. Figure 1a presents the duality gap of the current solutions vs. the number of steps. Here, both CBA+ and RM+ use *alternation*, which is a trick that is well-known to improve the performances of RM+ [TBJB15], where the repeated game framework is changed such that players take turns updating their strategies, rather than performing these updates simultaneously, see Appendix E.1 for details.[2]

For EFGs, we compare CBA+ and CFR$^+$ on many poker AI benchmark instances, including Leduc, Kuhn, search games and sheriff (see [FKS21] for game descriptions). We present our results in Figures 1b-1d. Additional details and experiments for EFGs are presented in Appendix E.4. Overall, we see in Figure 1 that CBA+ may slightly outperform RM+ and CFR$^+$, two of the strongest algorithms for matrix games and EFGs, which were shown to achieve the best empirical performances compared to a wide range of algorithms, including Hedge and other first-order methods [Kro20, KFS18, FKS19b]. For matrix games, AdaHedge and AdaFTRL are both outperformed by RM+ and CBA+; we present more experiments to compare RM+ and CBA+ in Appendix E.2, and more experiments with matrix games in Appendix E.3. Recall that our goal is to generalize these strong performance to other settings: we present our numerical experiments for solving distributionally robust optimization problems in the next section.

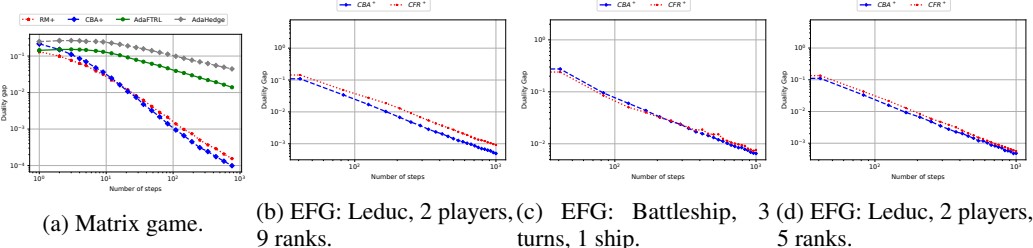

(a) Matrix game. (b) EFG: Leduc, 2 players, 9 ranks. (c) EFG: Battleship, 3 turns, 1 ship. (d) EFG: Leduc, 2 players, 5 ranks.

Figure 1: Comparison of CBA+ with RM+ and CFR$^+$ on matrix games and EFGs.

## 4.2 Distributionally Robust Optimization

**Problem setup** Broadly speaking, DRO attempts to exploit partial knowledge of the statistical properties of the model parameters to obtain risk-averse optimal solutions [RM19]. We focus on the following instance of distributionally robust classification with logistic losses [ND16, BTHKM15]. There are $m$ observed feature-label pairs $(\boldsymbol{a}_i, b_i) \in \mathbb{R}^n \times \{-1, 1\}$, and we want to solve

$$\min_{\boldsymbol{x}\in\mathbb{R}^n, \|\boldsymbol{x}-\boldsymbol{x}_0\|_2 \le R} \max_{\boldsymbol{y}\in\Delta(m), \|\boldsymbol{y}-\boldsymbol{y}_0\|_2^2 \le \lambda} \sum_{i=1}^{m} y_i \ell_i(\boldsymbol{x}), \tag{5}$$

where $\ell_i(\boldsymbol{x}) = \log(1 + \exp(-b_i \boldsymbol{a}_i^\top \boldsymbol{x}))$. The formulation (5) takes a worst-case approach to put more weight on misclassified observations and provides some statistical guarantees, e.g., it can be seen as a convex regularization of standard empirical risk minimization instances [DGN21].

We compare CBA+ (with linear averaging and alternation) with Online Mirror Descent (OMD), Optimistic OMD (O-OMD), Follow-The-Regularized-Leader (FTRL) and Optimistic FTRL (O-FTRL). We provide a detailed presentation of our implementations of these algorithms in Appendix

---

[2]We note that RM+ is guaranteed to retain its convergence rate under alternation. In contrast, we leave resolving this property for CBA+ to future work.

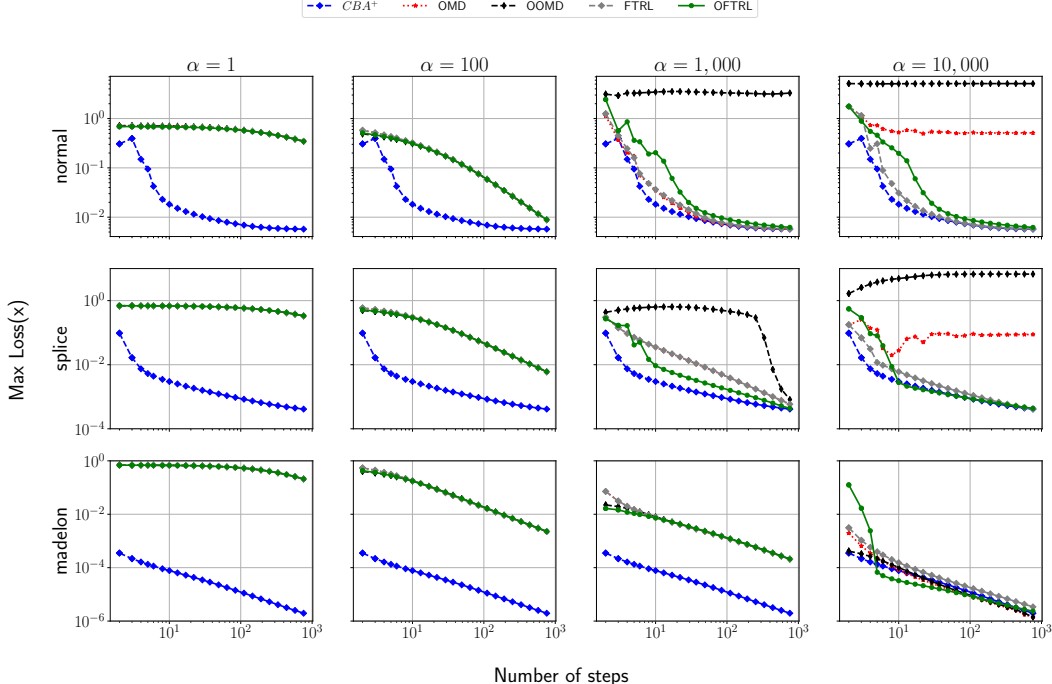

Figure 2: Comparisons of the performances of CBA$^+$ with OMD,FTRL,O-OMD and O-FTRL with fixed step sizes, on synthetic (with *normal* distribution) and real data sets (*splice* and *madelon*).

F. We compare the performances of these algorithms with CBA$^+$ on two synthetic data sets and four real data sets. We use linear averaging on decisions for all algorithms, and parameters $x_0 = 0, R = 10, y_0 = (1, ..., 1)/m, \lambda = 1/2m$ in eq. (5).

**Synthetic and real instances**   For the synthetic classification instances, we generate an optimal $x^* \in \mathbb{R}^n$, sample $a_i \sim N(0, I)$ for $i \in \{1, ..., m\}$, set labels $b_i = \text{sign}(a_i^\top x^*)$, and then we flip 10% of them. For the real classification instances, we use the following data sets from the libsvm website[3]: *adult*, *australian*, *splice*, *madelon*. Details about the empirical setting, the data sets and additional numerical experiments are presented in Appendix G.

**Choice of step sizes**   One of the main motivation for CBA$^+$ is to obtain a *parameter-free* algorithm. Choosing a *fixed* step size $\eta$ for the other algorithms requires knowing a bound $L$ on the norm of the instantaneous payoffs (see Appendix F.2 for our derivations of this upper bound). This is a major limitation in practice: these bounds may be very conservative, leading to small step sizes. We highlight this by showing the performance of all four algorithms, for various fixed step sizes $\eta = \alpha \times \eta_{\text{th}}$, where $\alpha \in \{1, 100, 1,000, 10,000\}$ is a multiplier and $\eta_{\text{th}}$ is the theoretical step size which guarantees the convergence of the algorithms for each instance. We detail the computation of $\eta_{\text{th}}$ in Appendix F.2. We present the results of our numerical experiments on synthetic and real data sets in Figure 2. Additional simulations with adaptive step sizes $\eta_t = 1/\sqrt{\sum_{\tau=1}^{t-1} \|f_\tau\|_2^2}$ [Ora19] are presented in Figure 3 and in Appendix G.

**Results and discussion**   In Figure 2, we present the worst-case loss of the current solution $\bar{x}_T$ in terms of the number of steps $T$. We see that when the step sizes is chosen as the theoretical step sizes guaranteeing the convergence of the non-parameter free algorithms ($\alpha = 1$), CBA$^+$ vastly outperforms all of the algorithms. When we take more aggressive step sizes, the non-parameter-free algorithms become more competitive. For instance, when $\alpha = 1,000$, OMD, FTRL and O-FTRL are competitive with CBA$^+$ for the experiments on synthetic data sets. However, for this same instance and $\alpha = 1,000$, O-OMD diverges, because the step sizes are far greater than the theoretical step sizes guaranteeing convergence. At $\alpha = 10,000$, both OMD and O-OMD diverge. The same type of performances also hold for the *splice* data set. Finally, for the *madelon* data set, the non

---

[3]https://www.csie.ntu.edu.tw/∼cjlin/libsvmtools/datasets/

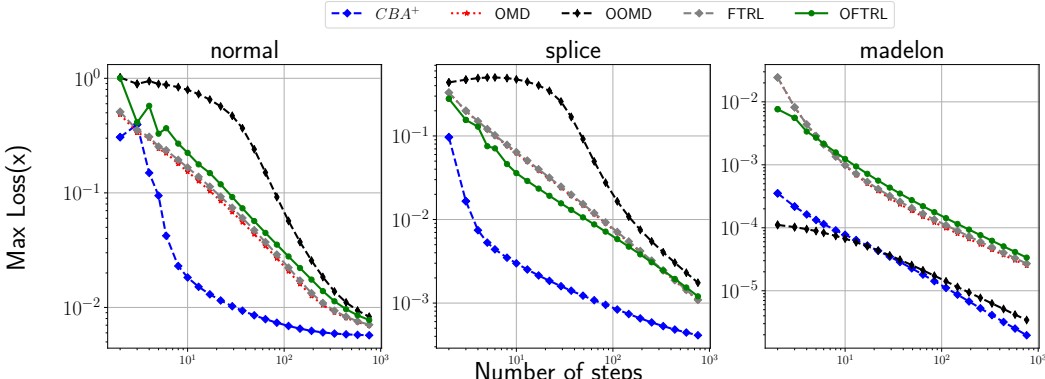

Figure 3: Comparisons of the performances of CBA$^+$, OMD,FTRL,O-OMD and O-FTRL with adaptive step sizes, on synthetic (with *normal* distribution) and real data sets (*splice* and *madelon*).

parameter-free algorithms start to be competitive with CBA$^+$ only when $\alpha = 10,000$. Again, we note that this range of step sizes $\eta$ is completely outside the values $\eta_{th}$ that guarantee convergence of the algorithms, and fine-tuning the algorithms is time- and resource-consuming. In contrast, CBA$^+$ can be used without wasting time on exploring and finding the best convergence rates, and with confidence in the convergence of the algorithm. Similar observations hold for adaptive step sizes (see Figure 3 and Appendix G). The overall poor performances of the optimistic methods (compared to their $O(1/T)$ average regret guarantees) may reflect their sensibility to the choice of the step sizes. Additional experiments in Appendix G with other real and synthetic EFG and DRO instances show the robustness of the strong performances of CBA$^+$ across additional problem instances.

**Running times compared to CBA$^+$** We would like to emphasize that all of our figures show the *number of steps* on the $x$-axis, and not the actual running times of the algorithms. Overall, CBA$^+$ converges to an optimal solution to the DRO instance (5) vastly faster than the other algorithms. In particular, empirically, CBA$^+$ is 2x-2.5x faster than OMD, FTRL and O-FTRL, and 3x-4x faster than O-OMD. This is because OMD, FTRL, O-OMD, and O-FTRL require binary searches at each step, see Appendix F. The functions used in the binary searches themselves require solving an optimization program (an orthogonal projection onto the simplex, see (33)) at each evaluation. Even though computing the orthogonal projection of a vector onto the simplex can be done in $O(n \log(n))$, this results in slower overall running time, compared to CBA$^+$ with (quasi) closed-form updates at each step. The situation is even worse for O-OMD, which requires two proximal updates at each iteration. We acknowledge that the same holds for CBA$^+$ compared to RM$^+$. In particular, CBA$^+$ is slightly slower than RM$^+$, because of the computation of $\pi_{\mathcal{C}}(\boldsymbol{u})$ in $O(n \log(n))$ operations at every iteration.

# 5 Conclusion

We have introduced CBA$^+$, a new algorithm for convex-concave saddle-point solving, that is 1) simple to implement for many practical decision sets, 2) completely parameter-free and does not attempt to lear any step sizes, and 3) competitive with, or even better than, state-of-the-art approaches for the best choices of parameters, both for matrix games, extensive-form games, and distributionally robust instances. Our paper is based on Blackwell approachability, which has been used to achieved important breakthroughs in poker AI in recent years, and we hope to generalize the use and implementation of this framework to other important problem instances. Interesting future directions of research include developing a theoretical understanding of the improvements related to alternation in our setting, designing efficient implementations for other widespread decision sets (e.g., based on Kullback-Leibler divergence or $\phi$-divergence), and novel accelerated versions based on strong convex-concavity or optimistim.

**Societal impact** There is a priori no direct negative societal consequence to this work, since our methods simply return the same solutions as previous algorithms but in a simpler and more efficient way.

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
