# A Comparison to [Shi16] and [FKS21]

The focus of our paper is on developing new algorithms for convex-concave saddle-point solving via Blackwell approachability algorithms for sets beyond the simplex domain. This is what motivated the development of CBA+, which attempts to generalize the ideas from RM+ and CFR+ beyond simplex settings. A complementary question is to consider the second construction of [ABH11], which is a way to convert a no-regret algorithm into an approachability algorithm. At a very level, that construction ends up using the no-regret algorithm to select which hyperplane to force, and performing regret minimization on the choice of hyperplane. An observation made by both [Shi16] and [FKS21] is that if one uses FTRL with the Euclidean regularizer, then it corresponds to Blackwell's original approachability algorithm, and thus to regret matching on the simplex. This begs the question of what happens if one uses a different algorithm from FTRL. The most natural substitute would be OMD with a Bregman divergence derived from the Euclidean distance, which is also known as online gradient descent (OGD).

[Shi16] considers OGD on a variation of the original construction of [ABH11] with the simplex as the decision set. This setup (and the original setup from [ABH11]) uses (a subset of) the unit ball as the feasible set for the regret minimizer, and thus OGD ends up projecting the cumulated payoff vector onto the unit ball after every iteration. Therefore, the OGD setup by [Shi16] yields an algorithm that is reminiscent of RM+, but where we repeatedly renormalize the cumulated payoff vector. One consequence of this is that the stepsize used to add new payoff vectors becomes important. [FKS21] show that the [ABH11] construction can be extended to allow the regret minimizer to use any decision set $\mathcal{D}$ such that $\mathcal{K} \subseteq \mathcal{D} \subseteq \mathcal{S}^\circ$, where $\mathcal{S}$ is the target set and $\mathcal{S}^\circ$ its polar cone, and $\mathcal{K}$ is $\mathcal{S}^\circ$ intersected with the unit ball. Then, [FKS21] consider the simplex regret-minimization setting, and show that OGD instantiated on $\mathbb{R}^n_+ = \mathcal{S}^\circ$ is equivalent to RM+.

CBA+ does not generalize either of the two simplex approaches above. For [Shi16], this can be seen because of the projection and subsequent dependence on stepsize required by [Shi16]'s construction. For [FKS21], this can be seen by the fact that they obtain RM+ when applying their approach to the simplex setting, and CBA+ differs from RM+ (in fact, that paper does not attempt to derive new unaccelerated regret minimizers; the goal is to design accelerated, or predictive/optimistic, variants of RM and RM+). However, an alternative derivation of CBA+ can be accomplished by using the generalization from [FKS21] where OGD is run with the decision set $\mathcal{S}^\circ$. Instead of applying OGD on the no-regret formulation of the Blackwell formulation of regret-minimization on the simplex with target set $\mathcal{S} = \mathbb{R}^n_-$ as in [FKS21], we can apply the OGD-setup of [FKS21] to the Blackwell formulation of a general no-regret problem with decision set $\mathcal{X}$ and target set $\mathcal{C}^\circ$ where $\mathcal{C} = \text{cone}(\{\kappa\} \times \mathcal{X})$. Then, we would get an indirect proof of correctness of the CBA+ algorithm through the correctness of OGD and the two layers of reduction from regret minimization to Blackwell approachability to regret minimization. However, we believe that this approach is significantly less intuitive than understanding CBA+ directly in terms of its properties as a Blackwell approachability algorithm.

# B Proofs of Theorem 3.1

**Notations and classical results in conic optimization**  We make use of the following facts. We provide a proof here for completeness.

**Lemma B.1.** *Let $\mathcal{C} \subset \mathbb{R}^{n+1}$ a closed convex cone and $\mathcal{C}^\circ$ its polar.*

1. *If $\boldsymbol{u} \in \mathbb{R}^{n+1}$, then $\boldsymbol{u} - \pi_{\mathcal{C}^\circ}(\boldsymbol{u}) = \pi_{\mathcal{C}}(\boldsymbol{u}) \in \mathcal{C}, \langle \boldsymbol{u} - \pi_{\mathcal{C}^\circ}(\boldsymbol{u}), \pi_{\mathcal{C}^\circ}(\boldsymbol{u}) \rangle = 0$, and $\|\boldsymbol{u} - \pi_{\mathcal{C}^\circ}(\boldsymbol{u})\|_2 \leq \|\boldsymbol{u}\|_2$.*

2. *If $\boldsymbol{u} \in \mathbb{R}^{n+1}$ then*
$$d(\boldsymbol{u}, \mathcal{C}) = \max_{\boldsymbol{w} \in \mathcal{C}^\circ \bigcap B_2(1)} \langle \boldsymbol{u}, \boldsymbol{w} \rangle,$$
   *where $B_2(1) = \{\boldsymbol{w} \in \mathbb{R}^{n+1} \mid \|\boldsymbol{w}\|_2 \leq 1\}$.*

3. *If $\boldsymbol{u} \in \mathcal{C}$, then $d(\boldsymbol{u}, \mathcal{C}^\circ) = \|\boldsymbol{u}\|_2$.*

4. Assume that $\mathcal{C} = cone(\{\kappa\} \times \mathcal{X})$ with $\mathcal{X} \subset \mathbb{R}^n$ convex compact and $\kappa = \max_{\boldsymbol{x} \in \mathcal{X}} \|\boldsymbol{x}\|_2$. Then $\mathcal{C}^\circ$ is a closed convex cone. Additionally, if $\boldsymbol{u} \in \mathcal{C}$ we have $-\boldsymbol{u} \in \mathcal{C}^\circ$.

5. Let us write $\leq_{\mathcal{C}^\circ}$ the order induced by $\mathcal{C}^\circ$ : $\boldsymbol{x} \leq_{\mathcal{C}^\circ} \boldsymbol{y} \iff \boldsymbol{y} - \boldsymbol{x} \in \mathcal{C}^\circ$. Then

$$\boldsymbol{x} \leq_{\mathcal{C}^\circ} \boldsymbol{y}, \boldsymbol{x}' \leq_{\mathcal{C}^\circ} \boldsymbol{y}' \Rightarrow \boldsymbol{x} + \boldsymbol{x}' \leq_{\mathcal{C}^\circ} \boldsymbol{y} + \boldsymbol{y}', \forall\, \boldsymbol{x}, \boldsymbol{x}', \boldsymbol{y}, \boldsymbol{y}' \in \mathbb{R}^{n+1}, \tag{6}$$

$$\boldsymbol{x} + \boldsymbol{x}' \leq_{\mathcal{C}^\circ} \boldsymbol{y} \Rightarrow \boldsymbol{x} \leq_{\mathcal{C}^\circ} \boldsymbol{y}, \forall\, \boldsymbol{x}, \boldsymbol{y} \in \mathbb{R}^{n+1}, \forall\, \boldsymbol{x}' \in \mathcal{C}^\circ, \tag{7}$$

6. Assume that $\boldsymbol{x} \leq_{\mathcal{C}^\circ} \boldsymbol{y}$ for $\boldsymbol{x}, \boldsymbol{y} \in \mathbb{R}^{n+1}$. Then $d(\boldsymbol{y}, \mathcal{C}^\circ) \leq \|\boldsymbol{x}\|_2$.

*Proof.*      1. The fact that $\boldsymbol{u} - \pi_{\mathcal{C}^\circ}(\boldsymbol{u}) = \pi_{\mathcal{C}}(\boldsymbol{u}) \in \mathcal{C}, \langle \boldsymbol{u} - \pi_{\mathcal{C}^\circ}(\boldsymbol{u}), \pi_{\mathcal{C}^\circ}(\boldsymbol{u}) \rangle = 0$ follows from Moreau's Decomposition Theorem [CR13]. The fact that $\|\boldsymbol{u} - \pi_{\mathcal{C}^\circ}(\boldsymbol{u})\|_2 \leq \|\boldsymbol{u}\|_2$ is a straightforward consequence of $\langle \boldsymbol{u} - \pi_{\mathcal{C}^\circ}(\boldsymbol{u}), \pi_{\mathcal{C}^\circ}(\boldsymbol{u}) \rangle = 0$.

2. For any $\boldsymbol{w} \in \mathcal{C}^\circ \bigcap B_2(1)$ we have

$$\langle \boldsymbol{u}, \boldsymbol{w} \rangle \leq \langle \boldsymbol{u} - \pi_C(\boldsymbol{u}), \boldsymbol{w} \rangle \leq \|\boldsymbol{w}\|_2 \|\boldsymbol{u} - \pi_C(\boldsymbol{u})\|_2 \leq \|\boldsymbol{u} - \pi_C(\boldsymbol{u})\|_2.$$

Conversely, since $(\boldsymbol{u} - \pi_{\mathcal{C}}(\boldsymbol{u})) / \|\boldsymbol{u} - \pi_{\mathcal{C}}(\boldsymbol{u})\|_2 \in \mathcal{C}^\circ$, we have

$$\max_{\boldsymbol{w} \in \mathcal{C}^\circ \bigcap B_2(1)} \langle \boldsymbol{u}, \boldsymbol{w} \rangle \geq \|\boldsymbol{u} - \pi_{\mathcal{C}}(\boldsymbol{u})\|_2.$$

This shows that

$$\max_{\boldsymbol{w} \in \mathcal{C}^\circ \bigcap B_2(1)} \langle \boldsymbol{u}, \boldsymbol{w} \rangle = \|\boldsymbol{u} - \pi_{\mathcal{C}}(\boldsymbol{u})\|_2 = d(\boldsymbol{u}, \mathcal{C}).$$

3. For any $\boldsymbol{u} \in \mathbb{R}^{n+1}$, by definition we have $d(\boldsymbol{u}, \mathcal{C}^\circ) = \|\boldsymbol{u} - \pi_{\mathcal{C}^\circ}(\boldsymbol{u})\|_2$. Now if $\boldsymbol{u} \in \mathcal{C}$ we have $\pi_{\mathcal{C}^\circ}(\boldsymbol{u}) = 0$ so $d(\boldsymbol{u}, \mathcal{C}^\circ) = \|\boldsymbol{u}\|_2$.

4. Let $\boldsymbol{u} \in \mathcal{C}$. Then $\boldsymbol{u} = \alpha(\kappa, \boldsymbol{x})$ for $\alpha \geq 0, \boldsymbol{x} \in \mathcal{X}$. We will show that $-\boldsymbol{u} \in \mathcal{C}^\circ$. We have

$$\begin{aligned} -\boldsymbol{u} \in \mathcal{C}^\circ &\iff \langle -\boldsymbol{u}, \boldsymbol{u}' \rangle \leq 0, \forall\, \boldsymbol{u}' \in \mathcal{C} \\ &\iff \langle -\alpha(\kappa, \boldsymbol{x}), \alpha'(\kappa, \boldsymbol{x}') \rangle \leq 0, \forall\, \alpha' \geq 0, \forall\, \boldsymbol{x}' \in \mathcal{X} \\ &\iff \kappa^2 + \langle \boldsymbol{x}, \boldsymbol{x}' \rangle \geq 0 \\ &\iff -\langle \boldsymbol{x}, \boldsymbol{x}' \rangle \leq \kappa^2, \end{aligned}$$

and $-\langle \boldsymbol{x}, \boldsymbol{x}' \rangle \leq \kappa^2$ is true by Cauchy-Schwartz and the definition of $\kappa = \max_{\boldsymbol{x} \in \mathcal{X}} \|\boldsymbol{x}\|_2$.

5. We start by proving (6). Let $\boldsymbol{x}, \boldsymbol{x}', \boldsymbol{y}, \boldsymbol{y}' \in \mathbb{R}^{n+1}$, and assume that $\boldsymbol{x} \leq_{\mathcal{C}^\circ} \boldsymbol{y}, \boldsymbol{x}' \leq_{\mathcal{C}^\circ} \boldsymbol{y}'$. Then $\boldsymbol{y} - \boldsymbol{x} \in \mathcal{C}^\circ, \boldsymbol{y}' - \boldsymbol{x}' \in \mathcal{C}^\circ$. Because $\mathcal{C}^\circ$ is a convex set, and a cone, we have $2 \cdot \left( \frac{\boldsymbol{y} - \boldsymbol{x}}{2} + \frac{\boldsymbol{y}' - \boldsymbol{x}'}{2} \right) \in \mathcal{C}^\circ$. Therefore, $\boldsymbol{y} + \boldsymbol{y}' - \boldsymbol{x} - \boldsymbol{x}' \in \mathcal{C}^\circ$, i.e., $\boldsymbol{x} + \boldsymbol{x}' \leq_{\mathcal{C}^\circ} \boldsymbol{y} + \boldsymbol{y}'$.

   We now prove (7). Let $\boldsymbol{x}, \boldsymbol{y} \in \mathbb{R}^{n+1}, \boldsymbol{x}' \in \mathcal{C}^\circ$ and assume that $\boldsymbol{x} + \boldsymbol{x}' \leq_{\mathcal{C}^\circ} \boldsymbol{y}$. Then by definition $\boldsymbol{y} - \boldsymbol{x} - \boldsymbol{x}' \in \mathcal{C}^\circ$. Additionally, $\boldsymbol{x}' \in \mathcal{C}^\circ$ by assumption. Since $\mathcal{C}^\circ$ is convex, and is a cone, $2 \cdot \left( \frac{\boldsymbol{y} - \boldsymbol{x} - \boldsymbol{x}'}{2} + \frac{\boldsymbol{x}'}{2} \right) \in \mathcal{C}^\circ$, i.e., $\boldsymbol{y} - \boldsymbol{x} \in \mathcal{C}^\circ$. Therefore, $\boldsymbol{x} \leq_{\mathcal{C}^\circ} \boldsymbol{y}$.

6. Let $\boldsymbol{x}, \boldsymbol{y} \in \mathbb{R}^{n+1}$ such that $\boldsymbol{x} \leq_{\mathcal{C}^\circ} \boldsymbol{y}$. Then $\boldsymbol{y} - \boldsymbol{x} \in \mathcal{C}^\circ$. We have

$$d(\boldsymbol{y}, \mathcal{C}^\circ) = \min_{\boldsymbol{z} \in \mathcal{C}^\circ} \|\boldsymbol{y} - \boldsymbol{z}\|_2 \leq \|\boldsymbol{y} - (\boldsymbol{y} - \boldsymbol{x})\|_2 = \|\boldsymbol{x}\|_2.$$

$\square$

Based on Moreau's Decomposition Theorem, we will use $\pi_{\mathcal{C}}(\boldsymbol{u})$ and $\boldsymbol{u} - \pi_{\mathcal{C}^\circ}(\boldsymbol{u})$ interchangeably.

**Results for various linear averaging schemes**   We now present our convergence results for various linear averaging schemes. As a warm-up, we start with two theorems, Theorem B.2 and Theorem B.3, which show that CBA and CBA$^+$ are compatible with weighted average schemes, when *both* the decisions and the payoffs are weighted. The proofs for these theorems will be used in the proof of our main theorem, Theorem 3.1. For the sake of consiness, in all the proofs of this section we will always write $L = \max\{\|\boldsymbol{f}_t\|_2 \mid t \geq 1\}, \kappa = \max\{\|\boldsymbol{x}\|_2 \mid t \geq 1\}, \boldsymbol{v}_t = (\langle \boldsymbol{f}_t, \boldsymbol{x}_t \rangle / \kappa, -\boldsymbol{f}_t)$. We start with the following theorem.

**Theorem B.2.** *Let* $(\boldsymbol{x}_t)_{t\geq 0}$ *the sequence of decisions generated by* CBA *with weights* $(\omega_t)_{t\geq 0}$ *and let* $S_t = \sum_{\tau=1}^{t} \omega_\tau$ *for any* $t \geq 1$. *Then*

$$\frac{\sum_{t=1}^{T} \omega_t \langle \boldsymbol{f}_t, \boldsymbol{x}_t \rangle - \min_{\boldsymbol{x} \in \mathcal{X}} \sum_{t=1}^{T} \omega_t \langle \boldsymbol{f}_t, \boldsymbol{x} \rangle}{S_T} = O\left(\kappa \cdot d(\boldsymbol{u}_T, \mathcal{C}^\circ)\right).$$

*Additionally,*

$$d(\boldsymbol{u}_T, \mathcal{C}^\circ)^2 = O\left( L^2 \cdot \frac{\sum_{t=1}^{T} \omega_t^2}{\left(\sum_{t=1}^{T} \omega_t\right)^2} \right).$$

*Overall,*

$$\frac{\sum_{t=1}^{T} \omega_t \langle \boldsymbol{f}_t, \boldsymbol{x}_t \rangle - \min_{\boldsymbol{x} \in \mathcal{X}} \sum_{t=1}^{T} \omega_t \langle \boldsymbol{f}_t, \boldsymbol{x} \rangle}{S_T} = O\left( \kappa L \frac{\sqrt{\sum_{t=1}^{T} \omega_t^2}}{\sum_{t=1}^{T} \omega_t} \right).$$

*Proof.* The proof proceeds in two steps. We start by proving

$$\frac{\sum_{t=1}^{T} \omega_t \langle \boldsymbol{f}_t, \boldsymbol{x}_t \rangle - \min_{\boldsymbol{x} \in \mathcal{X}} \sum_{t=1}^{T} \omega_t \langle \boldsymbol{f}_t, \boldsymbol{x} \rangle}{S_T} = O\left(\kappa \cdot d(\boldsymbol{u}_T, \mathcal{C}^\circ)\right).$$

We have

$$d(\boldsymbol{u}_T, \mathcal{C}^\circ) = \max_{\boldsymbol{w} \in \text{cone}(\{\kappa\} \times \mathcal{X}) \cap B_2(1)} \left\langle \frac{1}{S_T} \sum_{t=1}^{T} \omega_t \boldsymbol{v}_t, \boldsymbol{w} \right\rangle \tag{8}$$

$$\geq \max_{\boldsymbol{x} \in \mathcal{X}} \left\langle \frac{1}{S_T} \sum_{t=1}^{T} \omega_t \boldsymbol{v}_t, \frac{(\kappa, \boldsymbol{x})}{\|(\kappa, \boldsymbol{x})\|_2} \right\rangle$$

$$\geq \frac{1}{S_T} \max_{\boldsymbol{x} \in \mathcal{X}} \frac{\sum_{t=1}^{T} \omega_t \langle \boldsymbol{f}_t, \boldsymbol{x}_t \rangle - \sum_{t=1}^{T} \omega_t \langle \boldsymbol{f}_t, \boldsymbol{x} \rangle}{\|(\kappa, \boldsymbol{x})\|_2}, \tag{9}$$

where (8) follows from Statement 1 in Lemma B.1, and (9) follows from CBA maintaining

$$\boldsymbol{u}_t = \left( \frac{1}{S_t} \sum_{\tau=1}^{t} \omega_\tau \frac{\langle \boldsymbol{f}_\tau, \boldsymbol{x}_\tau \rangle}{\kappa}, -\frac{1}{S_t} \sum_{\tau=1}^{t} \omega_\tau \boldsymbol{f}_\tau \right), \forall\, t \geq 1.$$

We can conclude that

$$2\kappa d(\boldsymbol{u}_T, \mathcal{C}^\circ) \geq \frac{\sum_{t=1}^{T} \omega_t \langle \boldsymbol{f}_t, \boldsymbol{x}_t \rangle - \min_{\boldsymbol{x} \in \mathcal{X}} \sum_{t=1}^{T} \omega_t \langle \boldsymbol{f}_t, \boldsymbol{x} \rangle}{S_T}.$$

We now prove that

$$d(\boldsymbol{u}_T, \mathcal{C}^\circ)^2 = O\left( L^2 \frac{\sum_{\tau=1}^{T} \omega_\tau^2}{\left(\sum_{\tau=1}^{T} \omega_\tau\right)^2} \right).$$

We have

$$d(\boldsymbol{u}_{t+1}, \mathcal{C}^\circ)^2 = \min_{\boldsymbol{z} \in \mathcal{C}^\circ} \|\boldsymbol{u}_{t+1} - \boldsymbol{z}\|_2^2 \tag{10}$$

$$\leq \|\boldsymbol{u}_{t+1} - \pi_{\mathcal{C}^\circ}(\boldsymbol{u}_t)\|_2^2$$

$$\leq \left\| \frac{S_t}{S_t + \omega_{t+1}} \boldsymbol{u}_t + \frac{\omega_{t+1}}{S_t + \omega_{t+1}} \boldsymbol{v}_{t+1} - \pi_{\mathcal{C}^\circ}(\boldsymbol{u}_t) \right\|_2^2$$

$$\leq \left\| \frac{S_t}{S_t + \omega_{t+1}} (\boldsymbol{u}_t - \pi_{\mathcal{C}^\circ}(\boldsymbol{u}_t)) + \frac{\omega_{t+1}}{S_t + \omega_{t+1}} (\boldsymbol{v}_{t+1} - \pi_{\mathcal{C}^\circ}(\boldsymbol{u}_t)) \right\|_2^2$$

$$\leq \frac{1}{S_{t+1}^2} (S_t^2 \|\boldsymbol{u}_t - \pi_{\mathcal{C}^\circ}(\boldsymbol{u}_t)\|_2^2 + \omega_{t+1}^2 \|\boldsymbol{v}_{t+1} - \pi_{\mathcal{C}^\circ}(\boldsymbol{u}_t)\|_2^2$$

$$+ 2 S_t \omega_{t+1} \langle \boldsymbol{u}_t - \pi_{\mathcal{C}^\circ}(\boldsymbol{u}_t), \boldsymbol{v}_{t+1} - \pi_{\mathcal{C}^\circ}(\boldsymbol{u}_t) \rangle)$$

$$\leq \frac{1}{S_{t+1}^2} \left( S_t^2 \|\boldsymbol{u}_t - \pi_{\mathcal{C}^\circ}(\boldsymbol{u}_t)\|_2^2 + \omega_{t+1}^2 \|\boldsymbol{v}_{t+1} - \pi_{\mathcal{C}^\circ}(\boldsymbol{u}_t)\|_2^2 \right), \tag{11}$$

where (11) follows from

$$\langle \boldsymbol{u}_t - \pi_{\mathcal{C}^\circ}(\boldsymbol{u}_t), \boldsymbol{v}_{t+1} - \pi_{\mathcal{C}^\circ}(\boldsymbol{u}_t) \rangle = 0. \tag{12}$$

This is because:

- $\langle \boldsymbol{u}_t - \pi_{\mathcal{C}^\circ}(\boldsymbol{u}_t), \boldsymbol{v}_{t+1} \rangle = 0$. This is one of the crucial component of Blackwell's approachability framework: the current decision is chosen to force a hyperplane on the aggregate payoff. To see this, first note that $\boldsymbol{u}_t - \pi_{\mathcal{C}^\circ}(\boldsymbol{u}) = \pi_{\mathcal{C}}(\boldsymbol{u}_t)$. Let us write $\boldsymbol{\pi} = (\tilde{\pi}, \hat{\boldsymbol{\pi}}) = \pi_{\mathcal{C}}(\boldsymbol{u}_t)$. Note that by definition, $\boldsymbol{x}_{t+1} = (\kappa/\tilde{\pi})\hat{\boldsymbol{\pi}}$, and $\boldsymbol{v}_{t+1} = (\langle \boldsymbol{f}_{t+1}, \boldsymbol{x}_{t+1} \rangle/\kappa, -\boldsymbol{f}_{t+1})$. Therefore,

$$\begin{aligned}
\langle \boldsymbol{u}_t - \pi_{\mathcal{C}^\circ}(\boldsymbol{u}_t), \boldsymbol{v}_{t+1} \rangle &= \langle \boldsymbol{\pi}, \boldsymbol{v}_{t+1} \rangle \\
&= \langle (\tilde{\pi}, \hat{\boldsymbol{\pi}}), (\langle \boldsymbol{f}_{t+1}, \boldsymbol{x}_{t+1} \rangle/\kappa, -\boldsymbol{f}_{t+1}) \rangle \\
&= \langle (\tilde{\pi}, \hat{\boldsymbol{\pi}}), (\langle \boldsymbol{f}_{t+1}, (\kappa/\tilde{\pi})\hat{\boldsymbol{\pi}} \rangle/\kappa, -\boldsymbol{f}_{t+1}) \rangle \\
&= \langle \hat{\boldsymbol{\pi}}, \boldsymbol{f}_{t+1} \rangle - \langle \hat{\boldsymbol{\pi}}, \boldsymbol{f}_{t+1} \rangle \\
&= 0.
\end{aligned}$$

- $\langle \boldsymbol{u}_t - \pi_{\mathcal{C}^\circ}(\boldsymbol{u}_t), \pi_{\mathcal{C}^\circ}(\boldsymbol{u}_t) \rangle = 0$ from Statement 3 of Lemma B.1 and $\boldsymbol{u}_t - \pi_{\mathcal{C}^\circ}(\boldsymbol{u}_t) = \pi_{\mathcal{C}}(\boldsymbol{u}_t) \in \mathcal{C}$.

We therefore have

$$d(\boldsymbol{u}_{t+1}, \mathcal{C}^\circ)^2 \leq \frac{1}{S_{t+1}^2} \left( S_t^2 \|\boldsymbol{u}_t - \pi_{\mathcal{C}^\circ}(\boldsymbol{u}_t)\|_2^2 + \omega_{t+1}^2 \|\boldsymbol{v}_{t+1} - \pi_{\mathcal{C}^\circ}(\boldsymbol{u}_t)\|_2^2 \right).$$

This recursion directly gives

$$d(\boldsymbol{u}_{t+1}, \mathcal{C}^\circ)^2 \leq \frac{1}{S_{t+1}^2} \sum_{\tau=1}^{t} \omega_{\tau+1}^2 \|\boldsymbol{v}_{\tau+1} - \pi_{\mathcal{C}^\circ}(\boldsymbol{u}_\tau)\|_2^2 \leq O\left( L^2 \cdot \frac{\sum_{\tau=1}^{t+1} \omega_\tau^2}{S_{t+1}^2} \right),$$

where the last inequality follows from the definition of $\boldsymbol{v}_t$ and $L$. $\qquad\square$

**Theorem B.3.** *Let* $(\boldsymbol{x}_t)_{t\geq 0}$ *the sequence of decisions generated by* CBA+ *with weights* $(\omega_t)_{t\geq 0}$ *and let* $S_t = \sum_{\tau=1}^{t} \omega_\tau$ *for any* $t \geq 1$. *Then*

$$\frac{\sum_{t=1}^{T} \omega_t \langle \boldsymbol{f}_t, \boldsymbol{x}_t \rangle - \min_{\boldsymbol{x} \in \mathcal{X}} \sum_{t=1}^{T} \omega_t \langle \boldsymbol{f}_t, \boldsymbol{x} \rangle}{S_T} = O\left( \kappa \cdot d(\boldsymbol{u}_T, \mathcal{C}^\circ) \right).$$

*Additionally,*

$$d(\boldsymbol{u}_T, \mathcal{C}^\circ)^2 = O\left( L^2 \cdot \frac{\sum_{t=1}^{T} \omega_t^2}{\left( \sum_{t=1}^{T} \omega_t \right)^2} \right).$$

*Overall,*

$$\frac{\sum_{t=1}^{T} \omega_t \langle \boldsymbol{f}_t, \boldsymbol{x}_t \rangle - \min_{\boldsymbol{x} \in \mathcal{X}} \sum_{t=1}^{T} \omega_t \langle \boldsymbol{f}_t, \boldsymbol{x} \rangle}{S_T} = O\left( \kappa L \frac{\sqrt{\sum_{t=1}^{T} \omega_t^2}}{\sum_{t=1}^{T} \omega_t} \right).$$

*Proof of Theorem B.3.* The proof proceeds in two steps. We start by proving

$$\frac{\sum_{t=1}^{T} \omega_t \langle \boldsymbol{f}_t, \boldsymbol{x}_t \rangle - \min_{\boldsymbol{x} \in \mathcal{X}} \sum_{t=1}^{T} \omega_t \langle \boldsymbol{f}_t, \boldsymbol{x} \rangle}{S_T} = O\left( \kappa \cdot d(\boldsymbol{u}_T, \mathcal{C}^\circ) \right).$$

Recall that $\boldsymbol{v}_t = (\langle \boldsymbol{f}_t, \boldsymbol{x}_t \rangle/\kappa, -\boldsymbol{f}_t)$, and let us consider $\boldsymbol{R}_t = \frac{1}{S_t} \sum_{\tau=1}^{t} \omega_\tau \boldsymbol{v}_\tau$. By definition of $\boldsymbol{R}_t$, similarly as in the proof of Theorem B.2, we have

$$\frac{\sum_{t=1}^{T} \omega_t \langle \boldsymbol{f}_t, \boldsymbol{x}_t \rangle - \min_{\boldsymbol{x} \in \mathcal{X}} \sum_{t=1}^{T} \omega_t \langle \boldsymbol{f}_t, \boldsymbol{x} \rangle}{S_T} = O\left( \kappa \cdot d(\boldsymbol{R}_T, \mathcal{C}^\circ) \right).$$

Note that at any period $t$, we have

$$S_{t+1} \boldsymbol{u}_{t+1} - S_t \boldsymbol{u}_t \leq_{\mathcal{C}^\circ} S_{t+1} \boldsymbol{R}_{t+1} - S_t \boldsymbol{R}_t. \tag{13}$$

This is simply because $\boldsymbol{u}_{t+1} = \pi_{\mathcal{C}}(\boldsymbol{u}_{t+1/2}) = \boldsymbol{u}_{t+1/2} - \pi_{\mathcal{C}^\circ}(\boldsymbol{u}_{t+1/2})$ with

$$\boldsymbol{u}_{t+1/2} = \mathsf{UPDATEPAYOFF}_{\mathsf{CBA}}(\boldsymbol{u}_t) = \frac{S_t}{S_t + \omega_{t+1}}\boldsymbol{u}_t + \frac{\omega_{t+1}}{S_t + \omega_{t+1}}\boldsymbol{v}_{t+1}.$$

Now we have

$$
\begin{aligned}
S_{t+1}\boldsymbol{R}_{t+1} - S_t\boldsymbol{R}_t - (S_{t+1}\boldsymbol{u}_{t+1} - S_t\boldsymbol{u}_t) &= \omega_{t+1}\boldsymbol{v}_{t+1} + S_t\boldsymbol{u}_t - S_{t+1}\boldsymbol{u}_{t+1/2} + S_{t+1}\pi_{\mathcal{C}^\circ}(\boldsymbol{u}_{t+1/2}) \\
&= S_{t+1}\boldsymbol{u}_{t+1/2} - S_{t+1}\boldsymbol{u}_{t+1/2} + S_{t+1}\pi_{\mathcal{C}^\circ}(\boldsymbol{u}_{t+1/2}) \\
&= S_{t+1}\pi_{\mathcal{C}^\circ}(\boldsymbol{u}_{t+1/2}) \in \mathcal{C}^\circ.
\end{aligned}
$$

From (6) in Lemma B.1, we can sum the inequalities (13). Noticing that $\boldsymbol{u}_1 = \boldsymbol{R}_1$, we can conclude that

$$\boldsymbol{u}_t \leq_{\mathcal{C}^\circ} \boldsymbol{R}_t.$$

From $\boldsymbol{u}_t \in \mathcal{C}$ and Statement 6 in Lemma B.1, we have $d(\boldsymbol{R}_t, \mathcal{C}^\circ) \leq \|\boldsymbol{u}_t\|_2$. This implies

$$\frac{\sum_{t=1}^T \omega_t \langle \boldsymbol{f}_t, \boldsymbol{x}_t \rangle - \min_{\boldsymbol{x} \in \mathcal{X}} \sum_{t=1}^T \omega_t \langle \boldsymbol{f}_t, \boldsymbol{x} \rangle}{S_T} = O\left(\kappa \|\boldsymbol{u}_T\|_2\right).$$

We now turn to proving

$$\|\boldsymbol{u}_T\|_2^2 = O\left(L^2 \frac{\sum_{t=1}^T \omega_t^2}{\left(\sum_{t=1}^T \omega_t\right)^2}\right).$$

We have

$$\|\boldsymbol{u}_{t+1}\|_2^2 = \|\boldsymbol{u}_{t+1/2} - \pi_{\mathcal{C}^\circ}(\boldsymbol{u}_{t+1/2})\|_2^2 \tag{14}$$

$$\leq \|\boldsymbol{u}_{t+1/2}\|_2^2 \tag{15}$$

$$\leq \|\frac{S_t}{S_t + \omega_{t+1}}\boldsymbol{u}_t + \frac{\omega_{t+1}}{S_t + \omega_{t+1}}\boldsymbol{v}_{t+1}\|_2^2, \tag{16}$$

where (15) follows from Statement 1 in Lemma B.1. Therefore,

$$\|\boldsymbol{u}_{t+1}\|_2^2 \leq \frac{1}{(S_t + \omega_{t+1})^2}\left(S_t^2\|\boldsymbol{u}_t\|_2^2 + \omega_{t+1}^2\|\boldsymbol{v}_{t+1}\|_2^2 + 2S_t\omega_{t+1}\langle\boldsymbol{u}_t, \boldsymbol{v}_{t+1}\rangle\right).$$

By construction and for the same reason as for (12), $\langle\boldsymbol{u}_t, \boldsymbol{v}_{t+1}\rangle = 0$. Therefore, we have the recursion

$$\|\boldsymbol{u}_{t+1}\|_2^2 \leq \frac{1}{S_{t+1}^2}\left(S_t^2\|\boldsymbol{u}_t\|_2^2 + \omega_{t+1}^2\|\boldsymbol{v}_{t+1}\|_2^2\right).$$

By telescoping the inequality above we obtain

$$d(\boldsymbol{u}_{t+1}, \mathcal{C}^\circ)^2 \leq \frac{1}{S_{t+1}^2}\left(\sum_{\tau=1}^{t+1} \omega_\tau^2\|\boldsymbol{v}_\tau\|_2^2\right).$$

By definition of $L$,

$$\|\boldsymbol{u}_{t+1}\|_2^2 = O\left(L^2 \cdot \frac{\sum_{\tau=1}^{t+1} \omega_\tau^2}{S_{t+1}^2}\right).$$

$\square$

**Linear averaging only on decisions** We are now ready to prove our main convergence result, Theorem 3.1. Our proof heavily relies on the sequence of payoffs belonging to the cone $\mathcal{C}$ at every iteration ($\boldsymbol{u}_t \in \mathcal{C}, \forall\, t \geq 1$), and for this reason it does not extend to CBA. We also note that the use of conic optimization somewhat simplifies the argument compared to the proof that RM$^+$ is compatible with linear averaging [TBJB15].

*Proof of Theorem 3.1.* Recall that $\boldsymbol{v}_t = (\langle \boldsymbol{f}_t, \boldsymbol{x}_t \rangle / \kappa, -\boldsymbol{f}_t)$. By construction and following the same argument as for the proof of Theorem B.3, we have

$$\sum_{t=1}^{T} t\langle \boldsymbol{f}_t, \boldsymbol{x}_t \rangle - \min_{\boldsymbol{x} \in \mathcal{X}} \sum_{t=1}^{T} t\langle \boldsymbol{f}_t, \boldsymbol{x} \rangle = O\left(\kappa \cdot d\left(\sum_{t=1}^{T} t\boldsymbol{v}_t, \mathcal{C}^\circ\right)\right). \tag{17}$$

Additionally, Equation (13) for uniform weights ($\omega_\tau = 1, S_\tau = \tau$) yields

$$\boldsymbol{v}_{t+1} \geq_{\mathcal{C}^\circ} (t+1)\boldsymbol{u}_{t+1} - t\boldsymbol{u}_t.$$

Therefore,

$$(t+1)\boldsymbol{v}_{t+1} \geq_{\mathcal{C}^\circ} (t+1)^2\boldsymbol{u}_{t+1} - t^2\boldsymbol{u}_t - t\boldsymbol{u}_t.$$

Summing up the previous inequalities from $t = 1$ to $t = T - 1$ and using $\boldsymbol{u}_1 = \boldsymbol{v}_1$ we obtain

$$\sum_{t=1}^{T} t\boldsymbol{v}_t \geq_{\mathcal{C}^\circ} T^2\boldsymbol{u}_T - \sum_{t=1}^{T-1} t\boldsymbol{u}_t.$$

Note that since $\sum_{t=1}^{T-1} t\boldsymbol{u}_t \in \mathcal{C}$, Statement 4 in Lemma B.1 shows that $-\sum_{t=1}^{T-1} t\boldsymbol{u}_t \in \mathcal{C}^\circ$. Now, by applying (7) in Lemma B.1, we have

$$\sum_{t=1}^{T} t\boldsymbol{v}_t \geq_{\mathcal{C}^\circ} T^2\boldsymbol{u}_T - \sum_{t=1}^{T-1} t\boldsymbol{u}_t \Rightarrow \sum_{t=1}^{T} t\boldsymbol{v}_t \geq_{\mathcal{C}^\circ} T^2\boldsymbol{u}_T.$$

Since $T^2\boldsymbol{u}_T \in \mathcal{C}$, Statement 6 shows that

$$d\left(\sum_{t=1}^{T} t\boldsymbol{v}_t, \mathcal{C}^\circ\right) \leq \|T^2\boldsymbol{u}_T\|_2.$$

By construction $\boldsymbol{u}_T$ is the output of $\mathsf{CBA}^+$ with uniform weight, so that $d(\boldsymbol{u}_T, \mathcal{C}^\circ) = \|\boldsymbol{u}_T\|_2 = O(L/\sqrt{T})$ (see Theorem B.3). Therefore, $d(\sum_{t=1}^{T} t\boldsymbol{v}_t, \mathcal{C}^\circ) = O\left(L \cdot T^{3/2}\right)$. This shows that

$$\frac{\sum_{t=1}^{T} t\langle \boldsymbol{f}_t, \boldsymbol{x}_t \rangle - \min_{\boldsymbol{x} \in \mathcal{X}} \sum_{t=1}^{T} t\langle \boldsymbol{f}_t, \boldsymbol{x} \rangle}{T(T+1)} = O\left(\kappa\frac{d\left(\sum_{t=1}^{T} t\boldsymbol{v}_t, \mathcal{C}^\circ\right)}{T(T+1)}\right) = O\left(\kappa L/\sqrt{T}\right).$$

$\square$

**Comparisons of different weighted average schemes** We conclude this section with an empirical comparisons of the different weighted average schemes (Theorem B.2, Theorem B.3, and Theorem 3.1). We also compare these algorithms with $\mathsf{RM}^+$. We present our numerical experiments on sets of random matrix game instances in Figure 4. The setting is the same as in our simulation section, Section 4. We note that $\mathsf{CBA}^+$ with linear averaging only on decisions outperforms both $\mathsf{CBA}^+$ and $\mathsf{CBA}$ with linear averaging on both decisions and payoffs, as well as $\mathsf{RM}^+$ with linear averaging on decisions.

## B.1 Geometric intuition on the projection step of $\mathsf{CBA}^+$

Figure 5 illustrates the projection step $\pi_{\mathcal{C}}(\cdot)$ of $\mathsf{CBA}^+$. At a high level, from $\boldsymbol{u}_t$ to $\boldsymbol{u}_{t+1}$, an instantaneous payoff $\boldsymbol{v}_t$ is first added to $\boldsymbol{u}_t$ (where $\boldsymbol{v}_t = (\langle \boldsymbol{f}_t, \boldsymbol{x}_t \rangle / \kappa, -\boldsymbol{f}_t)$), and then the resulting vector $\boldsymbol{u}_t^+ = \boldsymbol{u}_t + \boldsymbol{v}_t$ is projected onto $\mathcal{C}$. The projection $\pi_{\mathcal{C}}(\cdot)$ moves the vector $\boldsymbol{u}_t^+$ along the edges of the cone $\mathcal{C}^\circ$, preserving the (orthogonal) distance $d$ to $\mathcal{C}^\circ$.

## C Proof of Theorem 3.2

Let $\omega_t = t, S_T = \sum_{t=1}^{T} \omega_t = T(T+1)/2$, and

$$\bar{\boldsymbol{x}}_T = \frac{1}{S_T} \sum_{t=1}^{T} \omega_t \boldsymbol{x}_t, \quad \bar{\boldsymbol{y}}_T = \frac{1}{S_T} \sum_{t=1}^{T} \omega_t \boldsymbol{y}_t.$$

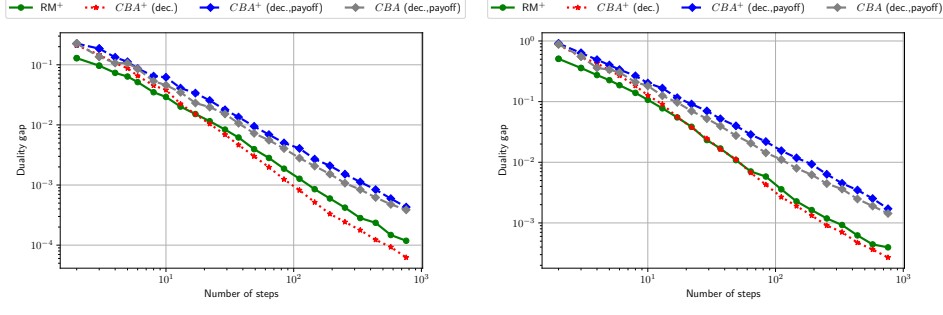

(a) Uniform distribution for payoffs.     (b) Normal distribution for payoffs.

Figure 4: Comparison of RM⁺ vs. CBA⁺ and CBA with different linear averaging schemes: only on decisions (CBA⁺ (dec.)), or on both the decisions and the payoffs $\boldsymbol{u}$ (CBA⁺ (dec.,payoff),CBA (dec.,payoff)).

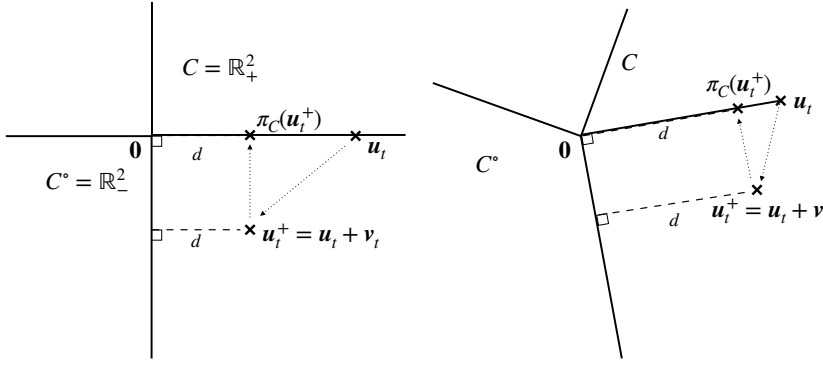

Figure 5: Illustration of $\pi_{\mathcal{C}}(\cdot)$ for $\mathcal{C} = \mathbb{R}^2_+$ (left-hand side) and $\mathcal{C}$ any cone in $\mathbb{R}^2$ (right-hand side).

Since $F$ is convex-concave, we first have

$$\max_{\boldsymbol{y}\in\mathcal{Y}} F(\bar{\boldsymbol{x}}_T, \boldsymbol{y}) - \min_{\boldsymbol{x}\in\mathcal{X}} F(\boldsymbol{x}, \bar{\boldsymbol{y}}_T) \le \frac{2}{T(T+1)} \left( \max_{\boldsymbol{y}\in\mathcal{Y}} \sum_{t=1}^T \omega_t F(\boldsymbol{x}_t, \boldsymbol{y}) - \min_{\boldsymbol{x}\in\mathcal{X}} \sum_{t=1}^T \omega_t F(\boldsymbol{x}, \boldsymbol{y}_t) \right).$$

Now,

$$\max_{\boldsymbol{y}\in\mathcal{Y}} \sum_{t=1}^T \omega_t F(\boldsymbol{x}_t, \boldsymbol{y}) - \min_{\boldsymbol{x}\in\mathcal{X}} \sum_{t=1}^T \omega_t F(\boldsymbol{x}, \boldsymbol{y}_t) = \left( \max_{\boldsymbol{y}\in\mathcal{Y}} \sum_{t=1}^T \omega_t F(\boldsymbol{x}_t, \boldsymbol{y}) - \sum_{t=1}^T \omega_t F(\boldsymbol{x}_t, \boldsymbol{y}_t) \right)$$
$$+ \left( \sum_{t=1}^T \omega_t F(\boldsymbol{x}_t, \boldsymbol{y}_t) - \min_{\boldsymbol{x}\in\mathcal{X}} \sum_{t=1}^T \omega_t F(\boldsymbol{x}, \boldsymbol{y}_t) \right).$$

Now since $F$ is convex-concave, we can use the following upper bound:

$$\max_{\boldsymbol{y}\in\mathcal{Y}} \sum_{t=1}^T \omega_t F(\boldsymbol{x}_t, \boldsymbol{y}) - \sum_{t=1}^T \omega_t F(\boldsymbol{x}_t, \boldsymbol{y}_t) \le \max_{\boldsymbol{y}\in\mathcal{Y}} \omega_t \sum_{t=1}^T \langle \boldsymbol{g}_t, \boldsymbol{y} \rangle - \sum_{t=1}^T \omega_t \langle \boldsymbol{g}_t, \boldsymbol{y}_t \rangle,$$

$$\sum_{t=1}^T \omega_t F(\boldsymbol{x}_t, \boldsymbol{y}_t) - \min_{\boldsymbol{x}\in\mathcal{X}} \sum_{t=1}^T \omega_t F(\boldsymbol{x}, \boldsymbol{y}_t) \le \sum_{t=1}^T \omega_t \langle \boldsymbol{f}_t, \boldsymbol{x}_t \rangle - \min_{\boldsymbol{x}\in\mathcal{X}} \sum_{t=1}^T \omega_t \langle \boldsymbol{f}_t, \boldsymbol{x} \rangle,$$

where $\boldsymbol{f}_t \in \partial_{\boldsymbol{x}} F(\boldsymbol{x}_t, \boldsymbol{y}_t), \boldsymbol{g}_t \in \partial_{\boldsymbol{y}} F(\boldsymbol{x}_t, \boldsymbol{y}_t)$ (recall the repeated game framework presented at the beginning of Section 2).

Now we have proved in Theorem 3.1 that

$$\frac{2}{T(T+1)} \max_{\boldsymbol{y} \in \mathcal{Y}} \sum_{t=1}^{T} \omega_t \langle \boldsymbol{g}_t, \boldsymbol{y} \rangle - \sum_{t=1}^{T} \langle \omega_t \boldsymbol{g}_t, \boldsymbol{y}_t \rangle = O\left(\kappa L / \sqrt{T}\right),$$

$$\frac{2}{T(T+1)} \sum_{t=1}^{T} \omega_t \langle \boldsymbol{f}_t, \boldsymbol{x}_t \rangle - \min_{\boldsymbol{x} \in \mathcal{X}} \sum_{t=1}^{T} \omega_t \langle \boldsymbol{f}_t, \boldsymbol{x} \rangle = O\left(\kappa L / \sqrt{T}\right).$$

Therefore, we can conclude that

$$\max_{\boldsymbol{y} \in \mathcal{Y}} F(\bar{\boldsymbol{x}}_T, \boldsymbol{y}) - \min_{\boldsymbol{x} \in \mathcal{X}} F(\boldsymbol{x}, \bar{\boldsymbol{y}}_T) = O\left(\kappa L / \sqrt{T}\right).$$

**Remark C.1.** Note that we essentially reprove the folk theorem, except that we consider *weighted* average for the decisions of both players. This is because Theorem 3.2 uses linear averaging on decisions, whereas Theorem 2.1 is written with uniform averaging on decisions.

# D  Proofs of the projections of Section 3.1

We will extensively use *Moreau's Decomposition Theorem* [CR13]: for any convex cone $\mathcal{C} \subset \mathbb{R}^{n+1}$ and $\boldsymbol{u} \in \mathbb{R}^{n+1}$, we can decompose $\boldsymbol{u} = \pi_{\mathcal{C}}(\boldsymbol{u}) + \pi_{\mathcal{C}^\circ}(\boldsymbol{u})$, where $\mathcal{C}^\circ$ is the *polar cone* of $\mathcal{C}$. Therefore, to compute $\pi_{\mathcal{C}}(\boldsymbol{u})$, it is sufficient to compute $\pi_{\mathcal{C}^\circ}(\boldsymbol{u})$, the orthogonal projection of $\boldsymbol{u}$ onto $\mathcal{C}^\circ$. We will see that in some cases, it is simpler to compute $\pi_{\mathcal{C}^\circ}(\boldsymbol{u})$ and then use $\pi_{\mathcal{C}}(\boldsymbol{u}) = \boldsymbol{u} - \pi_{C^\circ}(\boldsymbol{u})$ than directly computing $\pi_C(\boldsymbol{u})$ via solving (4).

## D.1  The case of the simplex

We consider $\mathcal{X} = \Delta(n)$. Note that in this case, $\kappa = \max_{\boldsymbol{x} \in \Delta(n)} \|\boldsymbol{x}\|_2 = 1$. The next lemma gives a closed-form expression of $\mathcal{C}^\circ$.

**Lemma D.1.** *Let $\mathcal{C} = cone(\{1\} \times \Delta(n))$. Then $\mathcal{C}^\circ = \{(\tilde{y}, \hat{\boldsymbol{y}}) \in \mathbb{R}^{n+1} \mid \max_{i=1,\dots,n} \hat{y}_i \leq -\tilde{y}\}$.*

*Proof of Lemma D.1.* Note that for $\boldsymbol{y} = (\tilde{y}, \hat{\boldsymbol{y}}) \in \mathbb{R}^{n+1}$ we have

$$
\begin{aligned}
\boldsymbol{y} \in \mathcal{C}^\circ &\iff \langle \boldsymbol{y}, \boldsymbol{z} \rangle \leq 0, \forall\, \boldsymbol{z} \in \mathcal{C} \\
&\iff \langle (\tilde{y}, \hat{\boldsymbol{y}}), \alpha(1, \boldsymbol{x}) \rangle \leq 0, \forall\, \boldsymbol{x} \in \Delta(n), \forall\, \alpha \geq 0 \\
&\iff \tilde{y} + \langle \hat{\boldsymbol{y}}, \boldsymbol{x} \rangle \leq 0, \forall\, \boldsymbol{x} \in \Delta(n) \\
&\iff \max_{\boldsymbol{x} \in \Delta(n)} \langle \hat{\boldsymbol{y}}, \boldsymbol{x} \rangle \leq -\tilde{y} \\
&\iff \max_{i=1,\dots,n} \hat{y}_i \leq -\tilde{y}.
\end{aligned}
$$

$\square$

For a given $\boldsymbol{u} = (\tilde{u}, \hat{\boldsymbol{u}})$, computing $\pi_{\mathcal{C}^\circ}(\boldsymbol{u})$ is now equivalent to solving

$$\min\{(\tilde{y} - \tilde{u})^2 + \|\hat{\boldsymbol{y}} - \hat{\boldsymbol{u}}\|_2^2 \mid (\tilde{y}, \hat{\boldsymbol{y}}) \in \mathbb{R}^{n+1}, \max_{i=1,\dots,n} \hat{y}_i \leq -\tilde{y}\}. \qquad (18)$$

Using the reformulation (18), we show that for a fixed $\tilde{y}$, the optimal $\hat{\boldsymbol{y}}(\tilde{y})$ can be computed in closed-form. It is then possible to avoid a binary search over $\tilde{y}$ and to simply use a sorting algorithm to obtain the optimal $\tilde{y}$. The next proposition summarizes our complexity result for $\mathcal{X} = \Delta(n)$.

**Proposition D.2.** *An optimal solution $\pi_{\mathcal{C}^\circ}(\boldsymbol{u})$ to (18) can be computed in $O(n \log(n))$ time.*

*Proof.* Computing $\pi_{\mathcal{C}^\circ}(\boldsymbol{u})$ is equivalent to computing

$$
\begin{aligned}
\min\ & (\tilde{y} - \tilde{u})^2 + \|\hat{\boldsymbol{y}} - \hat{\boldsymbol{u}}\|_2^2 \\
& \tilde{y} \in \mathbb{R}, \hat{\boldsymbol{y}} \in \mathbb{R}^n, \\
& \max_{i=1,\dots,n} \hat{y}_i \leq -\tilde{y}.
\end{aligned}
$$

Let us fix $\tilde{y} \in \mathbb{R}$ and let us first solve

$$
\begin{aligned}
\min \ & \|\hat{\boldsymbol{y}} - \hat{\boldsymbol{u}}\|_2^2 \\
& \hat{\boldsymbol{y}} \in \mathbb{R}^n, \\
& \max_{i=1,\dots,n} \hat{y}_i \le -\tilde{y}.
\end{aligned}
\tag{19}
$$

This is essentially the projection of $\hat{\boldsymbol{u}}$ on $(-\infty, -\tilde{y}]^n$. So a solution to (19) is $\hat{y}_i(\tilde{y}) = \min\{-\tilde{y}, \hat{u}_i\}, \forall\ i = 1, \dots, n$. Note that in this case we have $\hat{\boldsymbol{u}} - \hat{\boldsymbol{y}}(\tilde{y}) = (\hat{\boldsymbol{u}} + \tilde{y}\boldsymbol{e})^+$. So overall the projection brings down to the optimization of $\phi : \mathbb{R} \mapsto \mathbb{R}_+$ such that

$$
\phi : \tilde{y} \mapsto (\tilde{y} - \tilde{u})^2 + \|(\hat{\boldsymbol{u}} + \tilde{y}\boldsymbol{e})^+\|_2^2.
\tag{20}
$$

In principle, we could use binary search with a doubling trick to compute a $\epsilon$-minimizer of the convex function $\phi$ in $O\left(\log(\epsilon^{-1})\right)$ calls to $\phi$. However, it is possible to find a minimizer $\tilde{y}^*$ of $\phi$ using the following remark. By construction, we know that $\boldsymbol{u} - \pi_{\mathcal{C}^\circ}(\boldsymbol{u}) \in \mathcal{C}$. Here, $\mathcal{C} = \text{cone}\left(\{1\} \times \Delta(n)\right)$, and $\boldsymbol{u} - \pi_{\mathcal{C}^\circ}(\boldsymbol{u}) = \left(\tilde{u} - \tilde{y}^*, (\hat{\boldsymbol{u}} + \tilde{y}^*\boldsymbol{e})^+\right)$. This proves that

$$
\frac{(\hat{\boldsymbol{u}} + \tilde{y}^*\boldsymbol{e})^+}{\tilde{u} - \tilde{y}^*} \in \Delta(n),
$$

which in turns imply that

$$
\tilde{y}^* + \sum_{i=1}^n \max\{\hat{u}_i + \tilde{y}^*, 0\} = \tilde{u}.
\tag{21}
$$

We can use (21) to efficiently compute $\tilde{y}^*$ without using any binary search. In particular, we can sort the coefficients of $\hat{\boldsymbol{u}}$ in $O\left(n \log(n)\right)$ operations, and use (21) to find $\tilde{y}^*$. $\qquad \square$

Having obtained $\pi_{\mathcal{C}^\circ}(\boldsymbol{u})$, we can obtain $\pi_{\mathcal{C}}(\boldsymbol{u})$ by using the identity $\pi_{\mathcal{C}}(\boldsymbol{u}) = \boldsymbol{u} - \pi_{\mathcal{C}^\circ}(\boldsymbol{u})$. Note that RM and RM$^+$ are obtained by choosing the closed-form feasible point corresponding to $\tilde{y} = \tilde{u}$ in (18).

## D.2   The case of an $\ell_p$ ball

In this section we assume that $\mathcal{X} = \{\boldsymbol{x} \in \mathbb{R}^n \mid ; \|\boldsymbol{x}\|_p \le 1\}$ with $p \ge 1$ or $p = \infty$. The next lemma provides a closed-form reformulation of the polar cone $\mathcal{C}^\circ$.

**Lemma D.3.** *Let $\mathcal{X} = \{\boldsymbol{x} \in \mathbb{R}^n \mid ; \|\boldsymbol{x}\|_p \le 1\}$, with $p \ge 1$ or $p = \infty$. Then $\mathcal{C}^\circ = \{(\tilde{y}, \boldsymbol{y}) \in \mathbb{R} \times \mathbb{R}^n \mid \|\boldsymbol{y}\|_q \le -\kappa\tilde{y}\}$, with $q$ such that $1/p + 1/q = 1$.*

*Proof of Lemma D.3.* Let us write $B_p(1) = \{\boldsymbol{z} \in \mathbb{R}^n \mid \|\boldsymbol{z}\|_p \le 1\}$. Note that for $\boldsymbol{y} = (\tilde{y}, \hat{\boldsymbol{y}}) \in \mathbb{R}^{n+1}$ we have

$$
\begin{aligned}
\boldsymbol{y} \in \mathcal{C}^\circ \iff & \langle \boldsymbol{y}, \boldsymbol{z} \rangle \le 0, \forall\ \boldsymbol{z} \in \mathcal{C} \\
\iff & \langle (\tilde{y}, \hat{\boldsymbol{y}}), \alpha(\kappa, \boldsymbol{x}) \rangle \le 0, \forall\ \boldsymbol{x} \in B_p(1), \forall\ \alpha \ge 0 \\
\iff & \kappa\tilde{y} + \langle \hat{\boldsymbol{y}}, \boldsymbol{x} \rangle \le 0, \forall\ \boldsymbol{x} \in B_p(1), \\
\iff & \max_{\boldsymbol{x} \in B_p(1),} \langle \hat{\boldsymbol{y}}, \boldsymbol{x} \rangle \le -\kappa\tilde{y} \\
\iff & \|\boldsymbol{x}\|_q \le -\kappa\tilde{y},
\end{aligned}
$$

since $\|\cdot\|_q$ is the dual norm of $\|\cdot\|_p$. $\qquad \square$

The orthogonal projection problem onto $\mathcal{C}^\circ$ becomes

$$
\min\{(\tilde{y} - \tilde{u})^2 + \|\hat{\boldsymbol{y}} - \hat{\boldsymbol{u}}\|_2^2 \mid (\tilde{y}, \hat{\boldsymbol{y}}) \in \mathbb{R}^{n+1}, \|\hat{\boldsymbol{y}}\|_q \le -\kappa\tilde{y}\}.
\tag{22}
$$

For $p = 2$, (22) has a closed-form solution. For $p = 1$, a quasi-closed-form solution to (22) can be obtained efficiently using sorting. For $p = \infty$, it is more efficient to directly compute $\pi_{\mathcal{C}}(\boldsymbol{u})$. This is because the dual norm of $\|\cdot\|_\infty$ is $\|\cdot\|_1$.

**Proposition D.4.** • *For $p = 1$, $\pi_{\mathcal{C}^\circ}(\boldsymbol{u})$ can be computed in $O(n \log(n))$ arithmetic operations.*

• *For $p = \infty$, $\pi_{\mathcal{C}}(\boldsymbol{u})$ can be computed in $O(n \log(n))$ arithmetic operations.*

• *For $p = 2$, $\pi_{\mathcal{C}}(\boldsymbol{u})$ can be computed in closed-form.*

*Proof.* **The case $p = 1$.** Assume that $p = 1$. Then $\| \cdot \|_q = \| \cdot \|_\infty$ and $\kappa = 1$. We want to compute the projection of $(\tilde{u}, \hat{\boldsymbol{u}})$ on $\mathcal{C}^\circ$:

$$\min_{\boldsymbol{y} \in \mathcal{C}^\circ} \| \boldsymbol{y} - \boldsymbol{u} \|_2^2 = \min \; (\tilde{y} - \tilde{u})^2 + \| \hat{\boldsymbol{y}} - \hat{\boldsymbol{u}} \|_2^2$$
$$\tilde{y} \in \mathbb{R}, \hat{\boldsymbol{y}} \in \mathbb{R}^n, \tag{23}$$
$$\| \hat{\boldsymbol{y}} \|_\infty \leq -\tilde{y}.$$

For a fixed $\tilde{y}$, we want to compute

$$\min \| \hat{\boldsymbol{y}} - \hat{\boldsymbol{u}} \|_2^2$$
$$\hat{\boldsymbol{y}} \in \mathbb{R}^n, \tag{24}$$
$$\| \hat{\boldsymbol{y}} \|_\infty \leq -\tilde{y}.$$

The projection (24) can be computed in closed-form as

$$\hat{\boldsymbol{y}}^*(\tilde{y}) = \min\{-\tilde{y}, \max\{\tilde{y}, \hat{\boldsymbol{u}}\}\} \tag{25}$$

since this is simply the orthogonal projection of $\hat{\boldsymbol{u}}$ onto the $\ell_\infty$ ball of radius $-\tilde{y}$. Let us call $\phi : \mathbb{R} \mapsto \mathbb{R}$ such that

$$\phi(\tilde{y}) = (\tilde{y} - \tilde{u})^2 + \| \hat{\boldsymbol{y}}^*(\tilde{y}) - \hat{\boldsymbol{u}} \|_2^2.$$

Because of the closed-form expression for $\hat{\boldsymbol{y}}^*(\tilde{y})$ as in (25), we have

$$\phi : \tilde{y} \mapsto (\tilde{y} - \tilde{u})^2 + \| (\hat{\boldsymbol{u}} + \tilde{y}\boldsymbol{e})^+ \|_2^2.$$

Finding a minimizer of $\phi$ can be done in $O(n \log(n))$, with the same methods as in the proof in the previous section (Appendix D.1).

**The case $p = \infty$.** Let $p = \infty$. The problem of computing $\pi_{\mathcal{C}}(\boldsymbol{u})$, the orthogonal projection onto the cone $\mathcal{C}$, is equivalent to

$$\min_{\boldsymbol{y} \in \mathcal{C}} \| \boldsymbol{y} - \boldsymbol{u} \|_2^2 = \min \; (\tilde{y} - \tilde{u})^2 + \| \hat{\boldsymbol{y}} - \hat{\boldsymbol{u}} \|_2^2$$
$$\tilde{y} \in \mathbb{R}, \hat{\boldsymbol{y}} \in \mathbb{R}^n, \tag{26}$$
$$\| \hat{\boldsymbol{y}} \|_\infty \leq \tilde{y}.$$

Note the similarity between (26) (computing the orthogonal projection onto $\mathcal{C}$ when $p = \infty$), and (23) (computing the orthogonal projection onto $\mathcal{C}^\circ$ when $p = 1$). From Lemma D.3, we know that this is the case because $\| \cdot \|_1$ and $\| \cdot \|_\infty$ are dual norms to each other.

Therefore, the methods described for computing $\pi_{\mathcal{C}^\circ}(\boldsymbol{u})$ for $p = 1$ can be applied to the case $p = \infty$ for directly computing $\pi_{\mathcal{C}}(\boldsymbol{u})$. This gives the complexity results as stated in Proposition D.4: $\pi_{\mathcal{C}}(\boldsymbol{u})$ can be computed in $O(n \log(n))$ operations.

**The case $p = 2$.** Let $\| \cdot \|_p = \| \cdot \|_2$, then $\| \cdot \|_q = \| \cdot \|_2$. Let us fix $\tilde{y}$ and consider solving

$$\min \| \hat{\boldsymbol{y}} - \hat{\boldsymbol{u}} \|_2^2$$
$$\hat{\boldsymbol{y}} \in \mathbb{R}^n, \tag{27}$$
$$\| \hat{\boldsymbol{y}} \|_2 \leq -\tilde{y}.$$

The projection (27) can be computed in closed-form as

$$\hat{\boldsymbol{y}}^*(\tilde{y}) = (-\tilde{y}) \frac{\hat{\boldsymbol{u}}}{\| \hat{\boldsymbol{u}} \|_2},$$

since this is just the orthogonal projection of the vector $\hat{\boldsymbol{u}}$ onto the $\ell_2$-ball of radius $-\tilde{y}$. Let us call $\phi : \mathbb{R} \mapsto \mathbb{R}$ such that

$$\phi(\tilde{y}) = (\tilde{y} - \tilde{u})^2 + \| \hat{\boldsymbol{y}}^*(\tilde{y}) - \hat{\boldsymbol{u}} \|_2^2.$$

Note that here, $\tilde{y} \mapsto \hat{\boldsymbol{y}}^*(\tilde{y})$ is differentiable. Therefore $\phi : \tilde{y} \mapsto (\tilde{y} - \tilde{u})^2 + \|\hat{\boldsymbol{y}}^*(\tilde{y}) - \hat{\boldsymbol{u}}\|_2^2$ is also differentiable. First-order optimality conditions yield a closed-form solution for computing $(\tilde{y}^*, \hat{\boldsymbol{y}}^*) = \pi_{\mathcal{C}^\circ}(\boldsymbol{u})$, as

$$\tilde{y}^* = \frac{\tilde{u} - \|\hat{\boldsymbol{u}}\|_2}{2}, \hat{\boldsymbol{y}}^* = -\frac{1}{2}(\tilde{u} - \|\hat{\boldsymbol{u}}\|_2)\frac{\hat{\boldsymbol{u}}}{\|\hat{\boldsymbol{u}}\|_2}. \tag{28}$$

$\square$

## D.3   The case of an ellipsoidal confidence region in the simplex

In this section we assume that $\mathcal{X}$ is $\mathcal{X} = \{\boldsymbol{x} \in \Delta(n) \mid \|\boldsymbol{x} - \boldsymbol{x}_0\|_2 \le \epsilon_x\}$. We also assume that $\{\boldsymbol{x} \in \mathbb{R}^n \mid \boldsymbol{x}^\top \boldsymbol{e} = 1\} \bigcap \{\boldsymbol{x} \in \mathbb{R}^n \mid \|\boldsymbol{x} - \boldsymbol{x}_0\|_2 \le \epsilon_x\} \subseteq \Delta(n)$, so that we can write $\mathcal{X} = \boldsymbol{x}_0 + \epsilon\tilde{B}$, where $\tilde{B} = \{\boldsymbol{z} \in \mathbb{R}^n \mid \boldsymbol{z}^\top \boldsymbol{e} = 0, \|\boldsymbol{z}\|_2 \le 1\}$.

Suppose we took a sequence of decisions $\boldsymbol{x}_1, ..., \boldsymbol{x}_T$, which can be written as $\boldsymbol{x}_t = \boldsymbol{x}_0 + \epsilon\boldsymbol{z}_t$ for $\boldsymbol{z}_t \in \tilde{B}$. Then it is clear that for any sequence of payoffs $\boldsymbol{f}_1, ..., \boldsymbol{f}_T$, we have

$$\sum_{t=1}^T \omega_t\langle \boldsymbol{f}_t, \boldsymbol{x}_t\rangle - \min_{\boldsymbol{x}\in\mathcal{X}}\sum_{t=1}^T \omega_t\langle \boldsymbol{f}_t, \boldsymbol{x}\rangle = \epsilon_x\left(\sum_{t=1}^T \omega_t\langle \boldsymbol{f}_t, \boldsymbol{z}_t\rangle - \min_{\boldsymbol{z}\in\tilde{B}}\sum_{t=1}^T \omega_t\langle \boldsymbol{f}_t, \boldsymbol{z}\rangle\right). \tag{29}$$

Therefore, if we run CBA⁺ on the set $\tilde{B}$ to obtain $O\left(\sqrt{T}\right)$ growth of the right-hand side of (29), we obtain a no-regret algorithm for $\mathcal{X}$. We now show how to run CBA⁺ for the set $\tilde{B}$. Let $\mathcal{V} = \{\boldsymbol{v} \in \mathbb{R}^n \mid \boldsymbol{v}^\top \boldsymbol{e} = 0\}$. We use the following orthonormal basis of $\mathcal{V}$: let $\boldsymbol{v}_1, ..., \boldsymbol{v}_{n-1} \in \mathbb{R}^n$ be the vectors $\boldsymbol{v}_i = \sqrt{i/(i+1)}\,(1/i, ..., 1/i, -1, 0, ..., 0)\,, \forall\, i = 1, ..., n-1$, where the component $1/i$ is repeated $i$ times. The vectors $\boldsymbol{v}_1, ..., \boldsymbol{v}_{n-1}$ are orthonormal and constitute a basis of $\mathcal{V}$ [EPGMFBV03]. Writing $\boldsymbol{V} = (\boldsymbol{v}_1, ..., \boldsymbol{v}_{n-1}) \in \mathbb{R}^{n\times(n-1)}$, and noting that $\boldsymbol{V}^\top \boldsymbol{V} = \boldsymbol{I}$, we can write $\tilde{B} = \{\boldsymbol{V}\boldsymbol{s} \mid \boldsymbol{s} \in \mathbb{R}^{n-1}, \|\boldsymbol{s}\|_2 \le 1\}$. Now, if $\boldsymbol{x} = \boldsymbol{x}_0 + \epsilon_x\boldsymbol{z}_t$ with $\boldsymbol{z}_t \in \mathcal{V}$, we have $\boldsymbol{z}_t = \boldsymbol{V}\boldsymbol{s}_t$, for $\boldsymbol{s}_t \in \mathbb{R}^{n-1}$ and $\|\boldsymbol{s}\|_2 \le 1$. Finally,

$$\sum_{t=1}^T \omega_t\langle \boldsymbol{f}_t, \boldsymbol{x}_t\rangle - \min_{\boldsymbol{x}\in\mathcal{X}}\sum_{t=1}^T \omega_t\langle \boldsymbol{f}_t, \boldsymbol{x}\rangle = \epsilon_x\left(\sum_{t=1}^T \omega_t\langle \boldsymbol{V}^\top \boldsymbol{f}_t, \boldsymbol{s}_t\rangle - \min_{\boldsymbol{s}\in\mathbb{R}^{n-1}, \|\boldsymbol{s}\|_2\le 1}\sum_{t=1}^T \omega_t\langle \boldsymbol{V}^\top \boldsymbol{f}_t, \boldsymbol{s}\rangle\right). \tag{30}$$

Therefore, to obtain a regret minimizer for the left-hand side of (30) with observed payoffs $(\boldsymbol{f})_{t\ge 0}$, we can run CBA⁺ on the right-hand side, where the decision set is an $\ell_2$ ball and the sequence of observed payoffs is $\left(\boldsymbol{V}^\top \boldsymbol{f}_t\right)_{t\ge 0}$. In the previous section we showed how to efficiently instantiate CBA⁺ in this setting (see Proposition D.4).

**Remark D.5.** In this section we have highlighted a sequence of reformulations of the regret, from (29) to (30). We essentially showed how to instantiate CBA⁺ for settings where the decision set $\mathcal{X}$ is the intersection of an $\ell_2$ ball with a hyperplane for which we have an orthonormal basis.

# E   Additional details and numerical experiments for matrix games and EFGs

## E.1   Numerical setup

**Numerical setup for matrix games**   For the experiments on matrix games, we sample at random the matrix of payoffs $\boldsymbol{A} \in \mathbb{R}^{n\times m}$ and we let $n, m = 10$, where $n, m$ represent the number of actions of each player. We average our results over 70 instances. The decision sets $\mathcal{X}$ and $\mathcal{Y}$ are given as $\mathcal{X} = \Delta(n)$ and $\mathcal{Y} = \Delta(m)$.

**Alternation**  Alternation is a method which improves the performances of RM and RM⁺ [BMS19]. We leave proving this for CBA and CBA⁺ to future works. Using alternation, the players play in turn, instead of playing at the same time. In particular, the $y$-player may observe the current decision $\boldsymbol{x}_t$ of the $x$-player at period $t$, before choosing its own decision $\boldsymbol{y}_t$. For CBA and CBA⁺, it is implemented as follows. At period $t \geq 2$,

1. The $x$-player chooses $\boldsymbol{x}_t$ using its payoff $\boldsymbol{u}_{t-1}^x : \boldsymbol{x}_t = \mathsf{CHOOSEDECISION}(\boldsymbol{u}_{t-1}^x)$.

2. The $y$-player observes $\boldsymbol{g} \in \partial_{\boldsymbol{y}} F(\boldsymbol{x}_t, \boldsymbol{y}_{t-1})$ and updates $\boldsymbol{u}_t^y$:

$$\boldsymbol{u}_t^y = \mathsf{UPDATEPAYOFF}_{\mathsf{CBA}^+}(\boldsymbol{u}_{t-1}^y, \boldsymbol{y}_{t-1}, \boldsymbol{g}, \omega_t, \sum_{\tau=1}^{t-1} \omega_\tau).$$

3. The $y$-player chooses $\boldsymbol{y}_t$ using $\boldsymbol{u}_{t-1}^y : \boldsymbol{y}_t = \mathsf{CHOOSEDECISION}(\boldsymbol{u}_t^y)$.

4. The $x$-player observes $\boldsymbol{f} \in \partial_{\boldsymbol{x}} F(\boldsymbol{x}_t, \boldsymbol{y}_t)$ and updates $\boldsymbol{u}_t^x$:

$$\boldsymbol{u}_t^x = \mathsf{UPDATEPAYOFF}_{\mathsf{CBA}^+}(\boldsymbol{u}_t^x, \boldsymbol{x}_t, \boldsymbol{f}, \omega_t, \sum_{\tau=1}^{t-1} \omega_\tau).$$

## E.2   Comparing RM, RM⁺, CBA, and CBA⁺ on matrix games

In Figure 6 and Figure 7, we show the performances of RM, RM⁺, CBA and CBA⁺ with and without alternation, and with and without linear averaging. On the $y$-axis we show the duality gap of the current averaged decisions $(\bar{\boldsymbol{x}}_T, \bar{\boldsymbol{y}}_T)$. On the $x$-axis we show the number of iterations.

- In Figure 6a and Figure 7a, the four algorithms do not use alternation nor linear averaging, i.e., the four algorithms use uniform weights on the sequence of decisions. We note that RM⁺ is the best algorithm in this setting, outperforming CBA⁺, which performs similarly as RM.

- In Figure 6b and Figure 7b, we note that linear averaging (only on the decisions) slightly improves the performances of RM⁺ and CBA⁺. Note that RM and CBA are not known to be compatible with linear averaging (on decisions only), so we use uniform weights for RM and CBA here.

- In Figures 6c and Figure 7c, the four algorithms use alternation, but not linear averaging. The performances of the four algorithms are very similar.

- Finally, in Figure 6d and Figure 7d, RM⁺ and CBA⁺ use linear averaging on decisions *and* alternation. We see that the strongest performances are achieved by CBA⁺. Recall that RM and CBA are not known to be compatible with linear averaging (on decisions only), so we do not show their performances here.

- **Conclusion of our experiments.** We see that it is both alternation *and* linear averaging that enable the strong empirical performances of CBA⁺, rendering it capable to outperform RM⁺. Crucially, it is the "+ operation" that enables CBA⁺ (and RM⁺) to be compatible with linear averaging on the decisions only and to outperform CBA and RM.

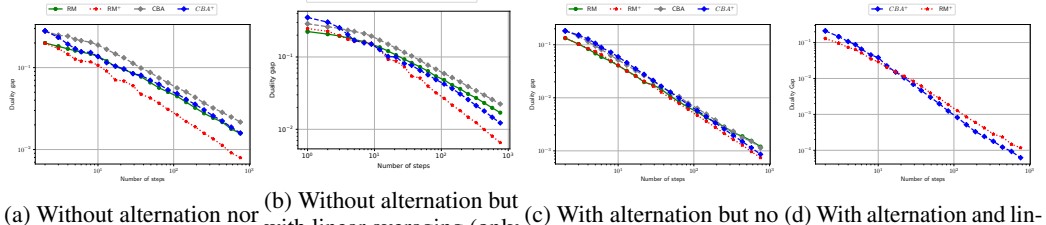

(a) Without alternation nor linear averaging.

(b) Without alternation but with linear averaging (only for RM⁺ and CBA⁺).

(c) With alternation but no linear averaging.

(d) With alternation and linear averaging.

Figure 6: Comparison of RM, RM⁺, CBA and CBA⁺ for $\mathcal{X} = \Delta(n), \mathcal{Y} = \Delta(m)$ and random matrices, with and without alternations, and with and without linear averaging. We choose $n, m = 10$ and $A_{ij} \sim U[0, 1]$ over 70 instances.

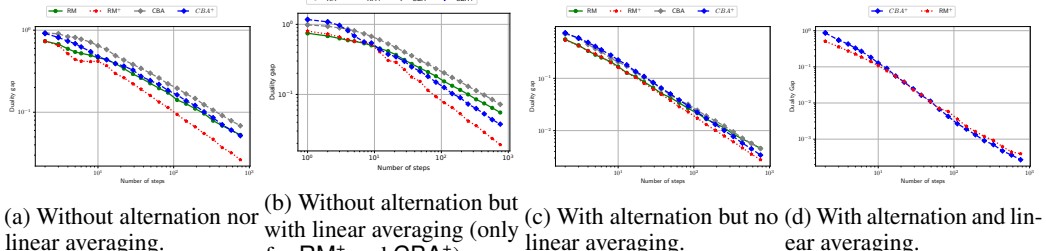

(a) Without alternation nor linear averaging.

(b) Without alternation but with linear averaging (only for RM+ and CBA+).

(c) With alternation but no linear averaging.

(d) With alternation and linear averaging.

Figure 7: Comparison of RM, RM+, CBA and CBA+ for $\mathcal{X} = \Delta(n), \mathcal{Y} = \Delta(m)$ and random matrices, with and without alternations, and with and without linear averaging. We choose $n, m = 10$ and $A_{ij} \sim N(0, 1)$ over 70 instances.

## E.3 Additional numerical experiments for matrix games

We have seen in Appendix E.2 that RM+ and CBA+ with alternation and linear averaging are outperforming RM and CBA. In Figure 1a, we have compared both RM+ and CBA+ with AdaFTRL [OP15] and AdaHedge [DRVEGK14], two algorithms that also enjoy the desirable *scale-free* property, i.e., their sequences of decisions remain invariant when the losses are scaled by a constant factor. In the next figure, we provide additional comparisons of RM+, CBA+, AdaHedge and AdaFTRL when the coefficients of the matrix of payoff are normally distributed. We found that RM+ and CBA+ are both outperforming AdaHedge and AdaFTRL, a situation similar to the case of uniform payoffs (Figure 1a.)

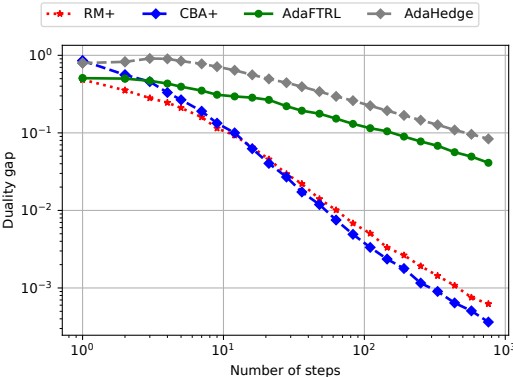

Figure 8: Comparison of RM+, CBA+, AdaHedge and AdaFTRL for $\mathcal{X} = \Delta(n), \mathcal{Y} = \Delta(m)$ and random matrices. We choose $n, m = 10$ and $A_{ij} \sim N(0, 1)$ over 70 instances.

## E.4 Additional numerical experiments for EFGs

In Section 4.1 , we have compared CBA+ (using alternation and linear averaging) and CFR+ on various EFGs instances. We present in Figure 9 additional simulations where CBA+ and CFR+ performs similarly. A description of the games can be found in [FKS21]. On the $y$-axis we show the duality gap of the current averaged decisions $(\bar{\boldsymbol{x}}_T, \bar{\boldsymbol{y}}_T)$. On the $x$-axis we show the number of iterations.

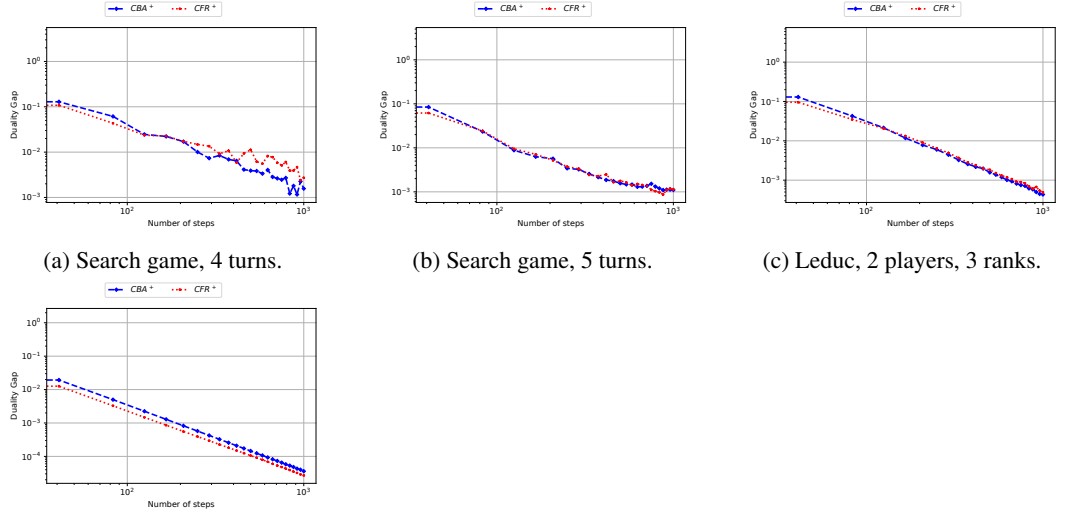

(a) Search game, 4 turns.    (b) Search game, 5 turns.    (c) Leduc, 2 players, 3 ranks.

(d) Leduc, 2 players, 6 faces.

Figure 9: Comparison of CBA$^+$ and CFR$^+$ for various Extensive-Form Games (EFG) instances.

# F OMD, FTRL and optimistic variants

## F.1 Algorithms

Let us fix some step size $\eta > 0$. For solving our instances of distributionally robust optimization, we compare Algorithm CBA$^+$ with the following four state-of-the-art algorithms:

1. Follow-The-Regularized-Leader (FTRL) [AHR09, McM11]:

$$\boldsymbol{x}_{t+1} \in \arg\min_{\boldsymbol{x}\in\mathcal{X}} \langle \sum_{\tau=1}^{t} \boldsymbol{f}_\tau, \boldsymbol{x} \rangle + \frac{1}{\eta}\|\boldsymbol{x}\|_2^2. \tag{FTRL}$$

Optimistic FTRL [RS13]: given an estimation $\boldsymbol{m}^{t+1}$ of loss at period $t + 1$, choose

$$\boldsymbol{x}_{t+1} \in \arg\min_{\boldsymbol{x}\in\mathcal{X}} \langle \sum_{\tau=1}^{t} \boldsymbol{f}_\tau + \boldsymbol{m}^{t+1}, \boldsymbol{x} \rangle + \frac{1}{\eta}\|\boldsymbol{x}\|_2^2. \tag{O-FTRL}$$

2. Online Mirror Descent (OMD) [NY83, BT03]:

$$\boldsymbol{x}_{t+1} \in \min_{\boldsymbol{x}\in\mathcal{X}} \langle \boldsymbol{f}_t, \boldsymbol{x} \rangle + \frac{1}{\eta}\|\boldsymbol{x} - \boldsymbol{x}_t\|_2^2. \tag{OMD}$$

Optimistic OMD [CYL$^+$12]: given an estimation $\boldsymbol{m}^{t+1}$ of loss at period $t + 1$,

$$\boldsymbol{z}_{t+1} \in \min_{\boldsymbol{z}\in\mathcal{X}} \langle \boldsymbol{m}_{t+1}, \boldsymbol{z} \rangle + \frac{1}{\eta}\|\boldsymbol{z} - \boldsymbol{x}_t\|_2^2,$$

Observe the loss $\boldsymbol{f}_{t+1}$ related to $\boldsymbol{z}_{t+1}$, $\qquad$ (O-OMD)

$$\boldsymbol{x}_{t+1} \in \min_{\boldsymbol{x}\in\mathcal{X}} \langle \boldsymbol{f}_{t+1}, \boldsymbol{x} \rangle + \frac{1}{\eta}\|\boldsymbol{x} - \boldsymbol{x}_t\|_2^2.$$

Note that a priori these algorithms can be written more generally using Bregman divergence (e.g., [BTN01]). We choose to work with $\|\cdot\|_2$ instead of *Kullback-Leibler divergence* as this $\ell_2$-setup is usually associated with faster empirical convergence rates [CP16, GKG19]. Additionally, following [CYL$^+$12, RS13], we use the last observed loss as the predictor for the next loss, i.e., we set $\boldsymbol{m}^{t+1} = \boldsymbol{f}_t$.

## F.2 Implementations

When $\mathcal{X}$ is the simplex or a ball based on the $\ell_2$-distance and centered at $\mathbf{0}$, there is a closed-form solution to the proximal updates for FTRL, OMD, O-FTRL and O-OMD. However, it is not clear how to compute these proximal updates for different settings, e.g., when $\mathcal{X}$ is a subset of the simplex or an $\ell_p$-ball. We present the details of our implementation below. The results in the rest of this section are reminiscent to the novel tractable proximal setups presented in [GCK20a, GCK20b].

**Computing the projection steps for min-player** For $\mathcal{X} = \{ \mathbf{x} \in \mathbb{R}^n \mid \|\mathbf{x} - \mathbf{x}_0\|_2 \leq \epsilon_x \}$, $\mathbf{c}, \mathbf{x}' \in \mathbb{R}^n$ and a step size $\eta > 0$, the proximal update becomes

$$\min_{\|\mathbf{x} - \mathbf{x}_0\|_2 \leq \epsilon_x} \langle \mathbf{c}, \mathbf{x} \rangle + \frac{1}{2\eta} \|\mathbf{x} - \mathbf{x}'\|_2^2. \tag{31}$$

This is the same arg min as

$$\min_{\|\mathbf{x} - \mathbf{x}_0\|_2 \leq \epsilon_x} \|\mathbf{x} - (\mathbf{x}' - \eta\mathbf{c})\|_2^2.$$

We can change $\mathbf{x}$ by $\mathbf{z} = (\mathbf{x} - \mathbf{x}_0)/\epsilon_x$ to solve the equivalent program

$$\min_{\|\mathbf{z}\|_2 \leq 1} \|\mathbf{z} - \frac{1}{\epsilon_x}(\mathbf{x}' - \eta\mathbf{c} - \mathbf{x}_0)\|_2^2.$$

The solution to the above program is

$$\mathbf{z} = \frac{\mathbf{x}' - \eta\mathbf{c} - \mathbf{x}_0}{\max\{\epsilon_x, \|\mathbf{x}' - \eta\mathbf{c} - \mathbf{x}_0\|_2\}}.$$

From $\mathbf{x} = \mathbf{x}_0 + \epsilon_x \mathbf{z}$ we obtain $\mathbf{x}^*$ the solution to (31)

$$\mathbf{x}^* = \mathbf{x}_0 + \epsilon_x \frac{\mathbf{x}' - \eta\mathbf{c} - \mathbf{x}_0}{\max\{\epsilon_x, \|\mathbf{x}' - \eta\mathbf{c} - \mathbf{x}_0\|_2\}}.$$

**Computing the projection steps for max-player** For $\mathcal{Y} = \{ \mathbf{y} \in \Delta(m) \mid \|\mathbf{y} - \mathbf{y}_0\|_2 \leq \epsilon_y \}$, the proximal update of the max-player from a previous point $\mathbf{y}'$ and a step size of $\eta > 0$ becomes

$$\min_{\|\mathbf{y} - \mathbf{y}_0\|_2 \leq \epsilon_y, \mathbf{y} \in \Delta(m)} \langle \mathbf{c}, \mathbf{y} \rangle + \frac{1}{2\eta} \|\mathbf{y} - \mathbf{y}'\|_2^2. \tag{32}$$

If we dualize the $\ell_2$ constraint with a Lagrangian multiplier $\mu \geq 0$ we obtain the relaxed problem $q(\mu)$ where

$$q(\mu) = -(1/2)\epsilon_y^2\mu + \min_{\mathbf{y} \in \Delta(m)} \langle \mathbf{c}, \mathbf{y} \rangle + \frac{1}{2\eta}\|\mathbf{y} - \mathbf{y}'\|_2^2 + \frac{\mu}{2}\|\mathbf{y} - \mathbf{y}_0\|_2^2. \tag{33}$$

Note that the arg min in

$$\min_{\mathbf{y} \in \Delta(m)} \langle \mathbf{c}, \mathbf{y} \rangle + \frac{1}{2\eta}\|\mathbf{y} - \mathbf{y}'\|_2^2 + \frac{\mu}{2}\|\mathbf{y} - \mathbf{y}_0\|_2^2$$

is the same arg min as in

$$\min_{\mathbf{y} \in \Delta(m)} \|\mathbf{y} - \frac{\eta}{\eta\mu + 1}\left(\frac{1}{\eta}\mathbf{y}' + \mu\mathbf{y}_0 - \mathbf{c}\right)\|_2^2. \tag{34}$$

Note that (34) is an orthogonal projection onto the simplex. Therefore, it can be solved efficiently [DSSSC08]. We call $\mathbf{y}(\mu)$ an optimal solution of (34). Then $q(\mu)$ can be rewritten

$$q(\mu) = -(1/2)\epsilon_y^2\mu + \langle \mathbf{c}, \mathbf{y}(\mu) \rangle + \frac{1}{2\eta}\|\mathbf{y}(\mu) - \mathbf{y}'\|_2^2 + \frac{\mu}{2}\|\mathbf{y}(\mu) - \mathbf{y}_0\|_2^2.$$

We can therefore binary search $q(\mu)$ as in the previous expression. An upper bound $\bar{\mu}$ for $\mu^*$ can be computed as follows. Note that

$$q(\mu) \leq -(1/2)\epsilon_y^2\mu + \langle \mathbf{c}, \mathbf{y}_0 \rangle + \frac{1}{2\eta}\|\mathbf{y}_0 - \mathbf{y}'\|_2^2.$$

Since $\mu \mapsto q(\mu)$ is concave we can choose $\bar{\mu}$ such that $q(\mu) \leq q(0)$. Using the previous inequality this yields

$$\bar{\mu} = \frac{2}{\epsilon_y^2}\left(\langle \mathbf{c}, \mathbf{y}_0 \rangle + \frac{1}{2\eta}\|\mathbf{y}_0 - \mathbf{y}'\|_2^2 - q(0)\right).$$

We choose a precision of $\epsilon = 0.001$ in our simulations. Note that these binary searches make OMD, FTRL, O-FTRL and O-OMD slower than $\mathsf{CBA}^+$ in terms of running times, since the updates in $\mathsf{CBA}^+$ only requires to compute the projection $\pi_C(\mathbf{u})$, and we have shown in Proposition D.4 and Appendix D.2 how to compute this in $O(n)$ when $\mathcal{X}$ is an $\ell_2$ ball $\mathcal{X} = \{ \mathbf{x} \in \Delta(n) \mid \|\mathbf{x} - \mathbf{x}_0\|_2 \leq \epsilon_x \}$.

**Computing the theoretical step sizes** We now give details about the choice of choice of theoretical step sizes. In theory (e.g., [BTN01]), for a player with decision set $\mathcal{X}$, we can choose $\eta_{\text{th}} = \sqrt{2}\Omega/L\sqrt{T}$ with $\Omega = \max_{\boldsymbol{x},\boldsymbol{x}' \in \mathcal{X}} \|\boldsymbol{x} - \boldsymbol{x}'\|_2$, and $L$ an upper bound on the norm of any observed loss $\boldsymbol{f}_t$: $\|\boldsymbol{f}_t\|_2 \leq L, \forall\, t \geq 1$. Note that this requires to know 1) the number of steps $T$, and 2) the upper bound $L$ on the norm of any observed loss $\boldsymbol{f}_t$, before the losses are generated. We now show how to compute $L_x$ and $L_y$ (for the $x$-player and the $y$-player) for an instance of the distributionally robust optimization problem (5).

1. For the $y$-player, the loss $\boldsymbol{f}_t$ is $\boldsymbol{f}_t = (\ell_i(\boldsymbol{x}_t))_{i \in [1,m]}$, with $\ell_i(\boldsymbol{x}) = \log(1 + \exp(-b_i \boldsymbol{a}_i^\top \boldsymbol{x}))$. For each $i \in [1, m]$ we have $|\ell_i| \leq \log(1 + \exp(|b_i|R\|\boldsymbol{a}_i\|_2))$ so that

$$L_y = \sqrt{\sum_{i=1}^m \log(1 + \exp(|b_i|R\|\boldsymbol{a}_i\|_2))^2}.$$

2. For the $x$-player we have $\boldsymbol{f}_t = \boldsymbol{A}^t \boldsymbol{y}_t$, where $\boldsymbol{A}^t$ is the matrix of subgradients of $\boldsymbol{x} \mapsto F(\boldsymbol{x}, \boldsymbol{y}_t)$ at $\boldsymbol{x}_t$:

$$A_{ij}^t = \frac{-1}{1 + \exp(b_i \boldsymbol{a}_i^\top \boldsymbol{x}_t)} b_i a_{i,j}, \forall\, (i,j) \in \{1, ..., m\} \times \{1, ..., n\}.$$

Therefore, $\|\boldsymbol{f}_t\|_2 \leq \|\boldsymbol{A}^t\|_2 \|\boldsymbol{y}_t\|_2 \leq \|\boldsymbol{A}^t\|_2$, because $\boldsymbol{y} \in \Delta(m)$. Now we have $\|\boldsymbol{A}^t\|_2 \leq \|\boldsymbol{A}^t\|_F = \sqrt{\sum_{i,j} |A_{ij}^t|^2}$. From $|A_{ij}^t| \leq |b_i a_{i,j}|$ we use

$$L_x = \sqrt{\sum_{i,j} |b_i a_{i,j}|^2}.$$

# G  Additional details and numerical experiments for distributionally robust optimization

We compare CBA$^+$ with alternation and linear averaging, OMD,FTRL,O-OMD and O-OMD for various step sizes $\eta$ where $\eta = \alpha\eta_{\text{th}}$ for $\alpha \in \{1, 100, 1,000, 10,000\}$, on additional synthetic and real data sets. We also add a comparison with adaptive step sizes.

**Data sets** We present here the characteristics of the data sets that we use in our DRO simulations. All data sets can be downloaded from the libsvm classification libraries[4]

- *Adult* data set: two classes, $m = 1,605$ samples with $n = 123$ features.

- *Australian* data set: two classes, $m = 690$ samples with $n = 14$ features.

- *Madelon* data set: two classes, $m = 2,000$ samples with $n = 500$ features.

- *Splice* data set: two classes, $m = 1,000$ samples with $n = 60$ features.

**Additional experiments with fixed step sizes** In this section we present additional numerical experiments for solving distributionally robust optimization instances in Figure 10. We use a synthetic data set, where we sample the features $a_{ij}$ as uniform random variables in $[0, 1]$. We also present results for the *adult* and the *australian* data sets from libsvm. We vary the aggressiveness of the step sizes $\eta = \alpha\eta_{\text{th}}$ by multiplying the theoretical step sizes $\eta_{\text{th}}$ by a multiplicative step factor $\alpha$. The empirical setting is the same as in Section 4. We note that our algorithm still outperforms or performs on par with the classical approaches after $10^2$ iterations, without requiring a single choice of parameter.

---

[4]https://www.csie.ntu.edu.tw/~cjlin/libsvmtools/datasets/

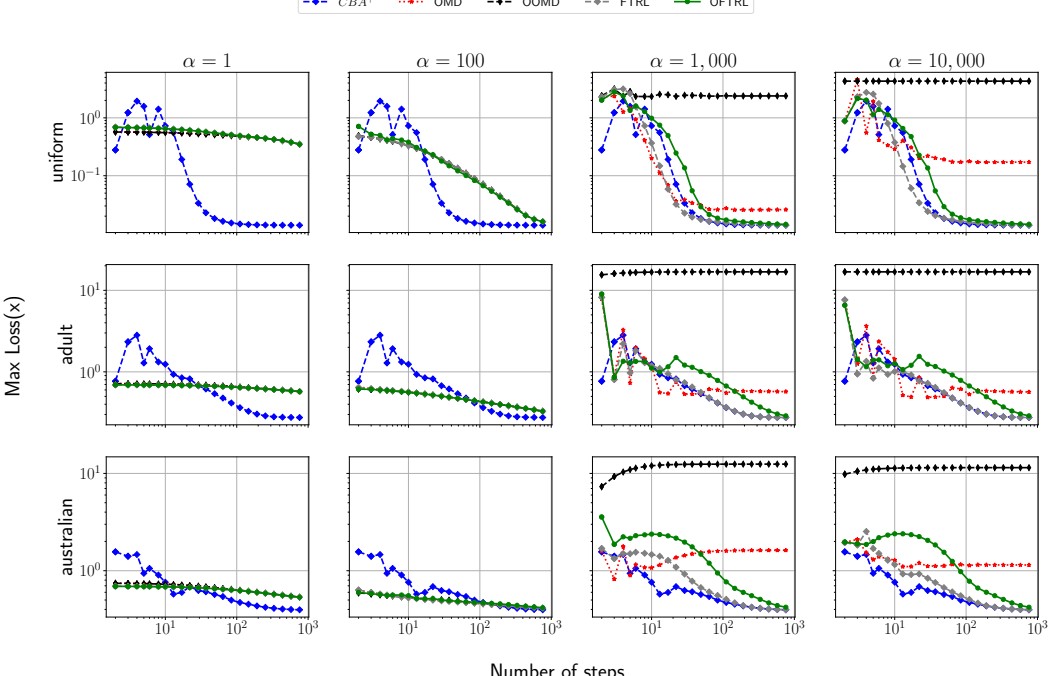

Figure 10: Comparisons of the performances of CBA⁺ with alternation and linear averaging, OMD,FTRL,O-OMD and O-FTRL on synthetic (with *uniform* distribution) and real data sets (*adult* and *australian*). We use fixed step sizes $\eta = \alpha\eta_{\text{th}}$, where $\eta_{\text{th}}$ is the theoretical step size that guarantees convergence.

**Additional experiments with adaptive step sizes** We present our additional results with adaptive step sizes in Figure 11. Given $\boldsymbol{v}_t$ the payoff observed by the player at period $t$, and following [Ora19], we choose the step sizes $(\eta_t)_{t \geq 1}$ as

$$\eta_t = 1/\sqrt{\sum_{\tau=1}^{t} \|\boldsymbol{v}_\tau\|_2^2}. \tag{35}$$

We note that CBA⁺ still outperforms, or performs on par, with the state-of-the-art approaches.

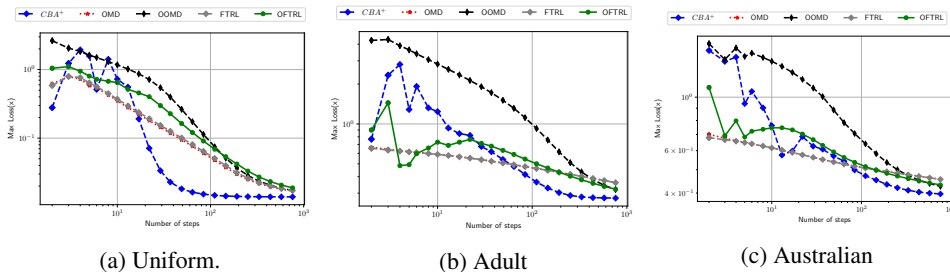

| (a) Uniform. | (b) Adult | (c) Australian |

Figure 11: Comparisons of the performances of CBA⁺ with alternation and linear averaging, OMD,FTRL,O-OMD and O-FTRL on synthetic (with *uniform* distribution) and real data sets (*adult* and *australian*). For the non-parameter free algorithms, we use the adaptive step sizes as in (35).