# OpenReview forum: "Conic Blackwell Algorithm: Parameter-Free Convex-Concave Saddle-Point Solving"
_NeurIPS.cc/2021/Conference — NeurIPS 2021 Poster_

### Official Review · Reviewer_yZ3K · 2021-07-10

**Rating:** 6
**Confidence:** 3

**Summary:**

This paper describes an online method for saddle point problems, which uses a sequence of linearizations to minimize for each player. The claim is that this method can accommodate conic domains (as opposed to previous methods which focused on simplex domains) and achieves 1/sqrt(T) average regret (the optimal rate). Discussions of efficient implementations and numerical results are also given.

**Ethical Concerns:**

There are no ethical concerns

**Limitations And Societal Impact:**

There is no negative societal impact

**Main Review:**

I am not very familiar with Blackwell algorithms, and the overall algorithmic scheme, which seemed similar to Frank-Wolfe method, is actually significantly different enough that I am not fully equipped to evaluate it in the context of similar methods. Therefore I can't judge the work's originality. In terms of significance, if the claims are as the authors state, and it is indeed a new method, then I believe it is significant.

The clarity of the main paper is very nice. The paper is easy to read, and the big picture comes through. One issue that I couldn't quite resolve, if kappa = max_{x\in X} ||x|| and the only restriction is that X is closed and convex, couldn't kappa be infinite? Does the set X need also be bounded?

To check the math, I went through lemma A1 and theorems A2, A3, and I have more significant questions. It is possible that I misunderstood most of it, and await the authors' responses, after which if the issues are reconciled I can increase the score.

 - Lemma A.1 statement 4 is not correct. If C = Rn, then C^\circ = {0}, for example.
 - line 535 I am not sure why uT can be written as 1/S_T sum_t omega_t ut. It does not seem from alg. 1 that you can recurse on x the way you can recurse on u.
 - eq. (8) there should be kappa in front of the first numerator term
 - line 537 why is 2kappa = ||(kappa, x)||_2?
 - (suggestion, not concern) most of the inequalities between (9) and (10) are in fact equalities
 - I am not sure why the statement 541 is designed to hold.
 - The statement for Thm A.2 and A.3 are exactly the same. Is one of the two a mistake?
 - line 553 I am not sure why omega_{t+1}v_{t+1} + S_tu_t = 0.
 - line 560 why is u_t^Tv_{t+1} = 0?
 - Line 562 why is ||v_tau||_2 a bounded sequence?



**Time Spent Reviewing:**

3

---

> ### Author Response · Authors · 2021-08-09
> **Response to Reviewer yZ3K**
>
> We would like to thank you for your detailed and accurate comments. We address them below and we will update our revised version accordingly. We hope that this convinces you of the soundness and significance of the paper.
>
> * Bounded decision sets $X$.
>
> As you have pointed out, we need $\kappa = \max \lbrace \lVert x \rVert \; | \; x \in X \rbrace$ to be finite for our results to apply. We will replace *$X$ closed an convex* in our paper by *$X$ compact and convex*.
>
> * Other technical comments. We respond point by point here.
>
> *  *Lemma A.1, Statement 4*:
>
> This statement holds when we  consider cones $C$ such that $C = cone(\{\kappa\} \times X )$. Indeed, it does not hold in all generality for any cone. We will highlight this more in the paper.
>
>
> * *Line 535*: you are right, this is a typo. In the equations between line 535 and line 536, $\boldsymbol{u}_t$ should be replaced by $\boldsymbol{v}_t$, defined line 530. This does not change the rest of the proof.
>
> * Thanks for pointing this out. The first component of the instantaneous payoff $( \langle f_t,x_{t} \rangle,-f_t )$ should actually be $( \frac{\langle f_t,x_t \rangle}{\kappa} ,-f_t )$. We will change this in the rest of the paper too.
>
> * We have $\lVert (\kappa,x \rVert _2 = \sqrt{\kappa^2 + \lVert x \rVert _2^2 } \leq \sqrt{\kappa^2 + \kappa^2} \leq \sqrt{2} \kappa \leq 2 \kappa.$
>
> * Thanks for mentioning this. We will update this in the manuscript.
>
> * This follows from Blackwell algorithm. To obtain inequality (10), we need  $$ \langle u_t - \pi_{C^o}(u_t),v_{t+1} - \pi_{C^o}(u_t) \rangle \leq 0.$$
>
> This can be decomposed into
> $$ \langle u_t - \pi_{C^o}(u_t),- \pi_{C^o}(u_t) \rangle \leq 0.$$
> and
> $$ \langle u_t - \pi_{C^o}(u_t),v_{t+1}  \rangle \leq 0.$$
> The first inequality is in fact an equality and follows from conic optimization (see Statement 1 in Lemma A.1).
>
> The second inequality is also an equality. It holds because of our choice of $x_{t+1}$ in Algorithms CBA and CBA+. This is one of the crucial component of Blackwell's framework - the current decision is chosen to force a hyperplane on the aggregate payoff. Let us for instance focus on CBA+. We have $u_t - \pi_{C^o}(u_t) = u_t$ because $u_t \in C \Rightarrow \pi_{C^o}(u_t)=0.$
>
> For the sake of conciseness we drop the subscript $t+1$. Let us consider the current payoff $\boldsymbol{u} = (\tilde{u},\hat{\boldsymbol{u}})$.
>
> Now by construction $v_{t+1}=(\frac{\langle f_{t+1},x_{t+1} \rangle}{\kappa} ,-f_{t+1} )$. Recall that we have chosen $x_{t+1} = (\kappa/\tilde{u})\hat{\boldsymbol{u}}$.
>
> So we have
> $$ \langle u_t ,v_{t+1}  \rangle =
> \langle (\tilde{u},\hat{\boldsymbol{u}}), \frac{\langle f_{t+1},(\kappa/\tilde{u}) \boldsymbol{\hat{u}} \rangle}{\kappa} ,-f_{t+1} ) \rangle = 0.$$
> It is straightforward to derive the same thing for CBA.
>
> Once we have obtained Inequality (10), we can telescope to obtain our bound on $d(u_t,C^o).$
>
> * Thanks for mentioning this, there is indeed a mistake in the statement of Theorem A.3, which relates to CBA+ (and not CBA, as written in our submission). We will correct this.
>
> * We have $\omega_{t+1}v_{t+1} + S_{t}u_t = S_{t+1}u_{t+1}$ by definition of the function $UPDATEPAYOFF_{CBA}$.
>
> * From our choice of $x_{t+1},$ we always have
> $$ \langle u_t - \pi_{C^o}(u_t),v_{t+1}  \rangle = 0.$$
>
>     Now since $u_t \in C$ (because of the $UPDATEPAYOFF_{CBA+}$ function), we have $\pi_{C^o}(u_t)=0$. So we have
>    $$ \langle u_t, v_{t+1} \rangle = 0.$$
>
> * Recall that we have
>  $$ v_t=\left( \frac{\langle f_t,x_t \rangle}{\kappa} ,-f_t \right).$$
>     Now if $x_t$ is bounded by $\kappa$ and $f_t$ are bounded, it is straightforward that $v_t$ is bounded too.
>
> Recall $x_t \in X$; we will add the assumption that $X$ is a compact set.
>
> Recall that $f_{t} = \nabla_{x} F(x_t,y_t)$ (see line 122). We have assumed that the losses come from a bounded set at line 170. This is the case if there exists $G_{x},G_{y} \geq 0$ such that
>     $$ \lVert \nabla_{x} F(x,y) \rVert \leq G_{x}, \lVert \nabla_{y} F(x,y) \rVert \leq G_{y}.$$
> Note that we do not need access to $G_{x},G_{y}$ to run the algorithm. In the revised version of our manuscript, we will highlight the fact that we assume bounded losses.
>
> * Conclusion.
>
> We would like to conclude by saying thanks for your detailed comments. We hope that we have addressed your concerns about the soundness and completeness of our paper.

---

> > ### Comment · Reviewer_yZ3K · 2021-08-17
> > **response to authors**
> >
> >  - re: convex + closed --> convex + compact. At this point it is not clear to me what the significance of "conic" is, then. Can the authors elaborate?
> >
> >  - Lemma A.1, Statement 4: ok, yes. In that case C is a proper cone, so it's kosher. (You may need to also explicitly state nonemptiness, which is a trivial assumption.)
> >
> >  - The second inequality is also an equality. It holds because of our choice of $x_{t+1}$  in Algorithms CBA and CBA+. This is one of the crucial component of Blackwell's framework - the current decision is chosen to force a hyperplane on the aggregate payoff. Let us for instance focus on CBA+. We have $u_t-\pi_{C^\circ}(u_t) = u_t$  because $u_t\in C \Rightarrow \pi_{C^\circ}(u_t) = 0$
> >
> > I'm not sure about this statement. The recursion on $x_t$ involves linear averages of $u_t$, but $u_t$ itself has the $\pi_C$ operation, interrupting the averaging. So I don't see how this orthogonality is maintained. That being said, the $\leq$ looks plausible (just doesn't seem it can be equality).
> >
> > I think that last statement is true only if the first $\tilde u$ were in the denominator (it's in the numerator in the first term, and denominator in the second term).
> >
> > All the comments I didn't respond to, I agree with the author's response. If these remaining issues are also all indeed minor / my mistakes, I will increase my score.

---

> > > ### Author Response · Authors · 2021-08-20
> > > **Second round of response to Reviewer yZ3K**
> > >
> > > We thank you for your quick and detailed response. Below are our responses to your new comments. We summarize your three new comments in italic.
> > >
> > > * *At this point it is not clear to me what the significance of "conic" is, then. Can the authors elaborate?*
> > >
> > > Let us detail why we decided to call our algorithm *Conic Blackwell Algorithm (CBA)*. Indeed, overall our algorithm provides a regret minimizer when the decision space is a convex, compact set $\mathcal{X}$ (which is itself not a cone). However, in order to create a regret minimizer for $\mathcal{X}$, it uses a lifted, conic set $\mathcal{C} = cone( \lbrace \kappa \rbrace \times \mathcal{X})$ and runs an instantiation of Blackwell algorithm on this cone $\mathcal{C}$. Hence the name *Conic Blackwell Algorithm (CBA)*. Hence also the need for the efficient projections onto the cone $\mathcal{C}$ for various sets $\mathcal{X}$, that we present in Section 3.1 (*Efficient implementations of $CBA^+$*).
> > >
> > > * *Lemma A.1, Statement 4.*
> > >
> > > For the exactness of the paper, we will add nonemptiness of the set of decisions $\mathcal{X}$. Thanks.
> > >
> > > * *Why do we have
> > > $ \langle u_t - \pi_{C^o}(u_t),v_{t+1} \rangle = 0$*?
> > >
> > > We will try to give a more detailed and more intuitive proof of this statement here.
> > > We first recall the notation: $$v_{t+1} \in \mathbb{R}^{n+1}, v_{t+1} = \left( \frac{\langle f_{t+1},x_{t+1} \rangle}{\kappa}, - f_{t+1} \right).$$
> > > We will write $v_{t+1}(f_{t+1},x_{t+1})$ to highlight that the instantaneous payoff $v_{t+1}$ depends on both the choice of the next decision $x_{t+1}$ and the next loss $f_{t+1}$.
> > >
> > > Blackwell algorithm essentially relies on the following fact:
> > >
> > > *Fact 1*: for any vector $\phi \in \mathcal{C} = cone( \lbrace \kappa \rbrace \times \mathcal{X}) \subset \mathbb{R}^{n+1}$, we can find $x_{t+1} \in \mathcal{X}$, such that
> > > $$\langle \phi,v(f,x_{t+1}) \rangle = 0, \forall f \in R^{n}.$$
> > > *Constructive proof.*
> > > For satisfying the above equality for a given $\phi = (\tilde{\phi},\hat{\phi})$, it is enough to choose $x_{t+1}= (\kappa/\tilde{\phi})\hat{\phi}$. By definition of $\mathcal{C} = cone( \lbrace \kappa \rbrace \times \mathcal{X})$, we have $x_{t+1} \in \mathcal{X}$. Additionally, we have, for any loss $f$,
> > > $$ \langle \phi,v(f,x_{t+1}) \rangle = \langle (\tilde{\phi},\hat{\phi}), (\frac{ \langle f, (\kappa/\tilde{\phi})\hat{\phi}) \rangle}{\kappa}, -f) \rangle = \langle (\tilde{\phi},\hat{\phi}),  (\langle f, (1/\tilde{\phi})\hat{\phi}) \rangle, -f) \rangle = \tilde{\phi} \times \langle f, (1/\tilde{\phi})\hat{\phi}) \rangle + \langle \hat{\phi},-f \rangle = \langle \hat{\phi}, f \rangle - \langle \hat{\phi},f \rangle =0. $$
> > > *End of constructive proof*.
> > >
> > > As you can see from the computation above, this orthogonality result is totally independent of the choice of the vector $\phi \in \mathcal{C}$, but relies entirely on the choice of $x_{t+1}$, given the vector $\phi$. Therefore, we can simply use the result from *Fact 1* with $\phi = u_t - \pi_{C^{o}}(u_t)$ , and the orthogonality result is always maintained, given the choice of $x_{t+1}$ as in $CHOOSEDECISION_{CBA}$ and $CHOOSEDECISION_{CBA^{+}}$.
> > >
> > >
> > > Note that the fact that the regret for the sequence $(x_t)_{t \geq 0}$ is still dominated by $O (d(u_t,C^{o}))$ for $CBA$  is proved in lines 534 - 537 in our submission. For $CBA^+$, it is harder to prove this, because of the $\pi_C$ operation for the aggregate payoff $u_t$. We prove this in lines 548 - 556.
> > >
> > > * *Conclusion.*
> > >
> > > We hope that our new responses address your concerns about the correctness of the paper. We will add more details in the revised version. We thank you for your detailed review, which greatly helped to improve the paper.

---

> > > > ### Comment · Reviewer_yZ3K · 2021-08-23
> > > > **appreciate authors' responses**
> > > >
> > > > These responses make sense to me, so I will increase my score.
> > > >
> > > > Overall, though, I do think that the authors should consider revising the paper to improve the clarity. Reading the other reviews, it seems I'm not the only one with this complaint, and I find this work very interesting; it's a shame if readability holds it back.

---

> > > > > ### Author Response · Authors · 2021-08-26
> > > > > **Final response to Reviewer yZ3K**
> > > > >
> > > > > We thank you for the time spent reviewing the paper and for the discussion. Following your comments, we will revise the paper to improve its readability.

---

### Official Review · Reviewer_fhFd · 2021-07-13

**Rating:** 6
**Confidence:** 4

**Summary:**

This paper proposed a conic Blackwell algorithm (CBA+) based on Blackwell approachability for solving convex-concave saddle point problems. The algorithm does not need to select stepsizes and achieves $O(1/\sqrt{T})$ convergence rate/average regret. The paper demonstrates strong empirical performance of CBA+ on matrix game and DRO problems.

**Ethical Concerns:**

The paper do not seem to have any ethical issues.

**Limitations And Societal Impact:**

This paper focused on theoretical work. The question is not applicable.

**Main Review:**

The paper tackles the interesting convex-concave saddle point problem. It summarizes the related literature nicely. Comparing to existing approaches like OMD, FTRL, Mirror Prox, Primal-dual approaches, CBA+ does not need to select stepsizes and achieves $O(1/\sqrt{T})$ convergence rate/average regret using a linearly increasing averaging scheme rather than a uniform average scheme. It is slower than the state-of-the-art $O(1/T)$ achieved by mirror prox though numerically better.

CBA+ builds upon the CBA algorithm [ABH11]. Both use the Blackwell approachability.  The paper also demonstrates $O(1/\sqrt{T})$ convergence rate/average of CBA algorithm. It is interesting to see algorithms originated from a different perspective than the classical mirror descent and etc. The idea of CBA+ comes from how RM+ is derived from RM algorithm in regret matching problems. Comparing to RM+, CBA+ can be applied to other constraints rather than just simplexes. The paper demonstrates strong empirical performance of CBA+ on matrix game and DRO problems. The results are quite interesting and intriguing though theoretical understanding on why CBA+ performs good in experiments is still a bit lacking.

The paper is overall well written. The proof is easy to follow, and the numerical settings are clear. Although some additional insights and illustration would be welcomed, see below.
-	The CBA+ algorithm differs from the CBA algorithm mainly in when to do projections onto the cone $\mathcal{C}$. It would be nice to further illustrate the intuition behind switching the projection order between UPDATEPAYOFF function and CHOOSEDECISION function.
-	In terms of theoretical results, both CBA and CBA+ algorithms achieve the $O(1/\sqrt{T})$ average regret when using weighted average on both the decisions and the payoffs. On the other hand, the numerical performance of CBA+ is even better than $O(1/T)$ average regret algorithm like mirror prox. It remains unclear whether the current analysis of CBA+ is tight. Demonstration of lower bounds ($O(1/\sqrt{T})$)  or ever tighter upper bounds ($O(1/T)$) for such Blackwell type algorithms would greatly enhance the contribution. Otherwise, the theoretical contribution is a bit limited.
-	In line 288, the paper says, “Numerically it also helps CBA+ but not CBA.” However, the paper did not provide results on CBA with linear increasing weighted average on decisions. It remains unclear which is the key to achieve better numerical performance, the switch of projection order or linear increasing weighted average on decisions.
-	In line 210-211, the paper says, “In practice, using linear averaging only on the decisions, as in Theorem 3.1, performs vastly better than linear averaging on both decisions and payoffs.” It would be nice to demonstrate the constants terms in the average regret to justify the numerical observations.
-	The CBA+ algorithm requires to solve projection operator onto closed convex cones optimally, it can be done on three certain cones as shown in the paper. However, for more general constraints, when exact projections to cones are expensive, what should one do?

Some minor issues:
1.	Line 499, is it $u_T$ or $u$?
2.	It should be “Blackwell approachability” rather than “Blackwell approchability”.
3.	It would be nice in Thm 3.2 to specify the selection of $w_\tau$.


**Time Spent Reviewing:**

6 hours

---

> ### Author Response · Authors · 2021-08-09
> **Response to Reviewer fhFd**
>
> We thank you for your detailed review and encouraging comments. We answer them in detail below.
>
> * Intuition on projection in CBA+.
>
> We will add some more intuition on why the geometric interpretation of the projection step for the aggregate payoff in CBA+. We summarize the main intuitions below. It is possible to obtain intuition in two ways: one geometric, one more game-theoretic.
>
> *Geometric*: It is easier to visualize things in $R^{2}$ with $C = R^2_+$ and $C^{o} = R^{2}_{-}$. The projection on $C$ then moves the payoff along the edges of $C^{o}$, maintaining the distance to $C^{o}$ and moving toward the vector $0$. Therefore, the projection step is maintaining the distance to $C^o$. Unfortunately, we can not add a picture in OpenReview but we will add a picture in $R^{2}$ in the revised version.
>
> *Game-theoretic*: The projection on $R^2_+$ eliminates the components of the payoffs that are negative; it enables CBA+ to be less *pessimistic* than CBA, which may accumulate negative payoffs on actions for a long time, leading to some actions being chosen less frequently. Somehow, the no-regret guarantee for CBA+ is valid starting from any period of *reset* of a component to $0$, i.e., when one component of the aggregated payoff is negative and the projection onto $R^2_+$ sets it to $0$. This is not the case for CBA, which never reset the component of the aggregated payoff to $0$.
>
> * Tight analysis of CBA+.
>
> The fact that the Blackwell approachability framework both attains theoretical $O(1/\sqrt{T})$ average regret while numerically performing better than methods achieving theoretical $O(1/T)$ average regret is well-known in the literature on matrix games and Extensive Form Games (EFGs) [1,2,3]. One of the main contributions of our paper is to show the generality of this Blackwell framework and its applicability to other problem instances: here, Distributionally Robust Optimization problems, but more generally, convex-concave saddle-point problems.
>     Even for the specific case of Regret Matching (RM) [4] and Regret Matching + (RM+) [1], which are instantiated on the simplex and both achieving $O(1/\sqrt{T})$ average regret, no lower-bound is known, despite the immense popularity of these two algorithms for game solving [5,6].
>     Please note that a worse-than-$1/T$ rate has been observed *empirically* for CFR+ on some cooked-up matrix games, see e.g. [7].
>     Showing lower bounds on these algorithms would be interesting, but we believe it is beyond the scope of our paper, where our goal is to demonstrate the generality and applicability of Blackwell's approachability framework to other settings than matrix games and EFGs. Even for algorithms that have been developed decades ago (e.g., OMD), it is very hard to show lower-bounds, and the only known results are very technical and appear hard to extend to other settings [10].
>
>
> * Line 288: “Numerically it also helps CBA+ but not CBA.”*
>
> We have observed that CBA with linear averaging on decisions only may not perform faster than CBA with linear averaging on both decisions and payoffs. Actually, it is not even known if CBA with linear averaging on decisions only leads to sublinear regret. The key to the better numerical performances of CBA+ are {\em both} the projection step and the linear averaging on decisions only, the latter becoming {\em compatible} with CBA+ only because of the former. This situation is analogous to RM and RM+.
>
> * Linear averaging.
>
> Thanks for mentioning this. In our revision, we have computed the constant in front of the $1/\sqrt{T}$ term for the regret.
>
> Let us write $L$ the upper bound on the gradient norm. Recall that $\kappa = \max \lbrace \lVert x \rVert | x \in X \rbrace.$ For choices of weights $\omega_{t} = t^{q}$, it is possible to obtain that for CBA and CBA+ we have
>     $$ R_{T} = O \left( \frac{(q+1) \kappa L }{\sqrt{T}} \right).$$
>     This is by combining line 537 (linking $R_{T}$ and $d(u_{T},C^{o})$) and line 543 (bound on $d(u_{T},C^{o})$). We can compare the $\kappa L$ part with the $\Omega L$ part of OMD, where $\Omega = \max \lbrace \lVert x \rVert - \lVert x’ \rVert | x,x’ \in X\rbrace$ (e.g, Equation 5.3.25 in [8]). Note that while $\kappa$ is not shift invariant, we can simply shift the set $X$ to recenter it at $0$.
>
> Finally, the choice of increasing weights $\omega_{t} = t^{q}$ adds a small constant $(q+1)$ to the regret bound. The same situation holds for OMD and other algorithms with increasing weights, see [9] for instance. Also, for RM+ with linear averaging on decisions only, a factor of 2 (which corresponds to q+1 when q=1) also appears in the regret bound (e.g., Theorem 3 in [1]).
>
> * Projection on general cones.
>
> In this paper, we intend to demonstrate the applicability and strong performances of Blackwell approachability framework to other settings than matrix games and EFGs. As we have focused on DRO problems, we provided closed-form projections for settings that naturally arise in these types of problem instances, e.g., $X =$  the simplex, $L_{p}$-balls and ellipsoidal confidence regions inside the simplex. It remains to study whether these closed-form projections can be computed efficiently for other settings.
>
> Let us use the notations of Appendix C here. For a given decision set $X$, it is always possible to use binary search over the first component $\tilde{y}$, as long as a corresponding optimal $\hat{\boldsymbol{y}}$ can be obtained efficiently. The problem of computing the projection onto $C = cone(\{\kappa \} \times X)$ then brings down to the problem of computing the orthogonal projection of a vector onto $X$. It remains to study whether this orthogonal projection can be computed efficiently for other classes of important decision sets, e.g., confidence regions based on $\phi$-divergence. If closed-form expressions are not available, it is possible to solve each of these projections with a general iterative method for computing the orthogonal projecting onto a compact convex set.
>
>
> Minor issues: thanks for pointing out these typos/inaccuracies. We will revise our paper accordingly.
>
> * Conclusion:
>
> We thank you again for your time reviewing the paper and we hope that we have addressed your concerns. We would like to conclude our response by re-emphasizing the novelty and the main contributions of our paper. The fact that Blackwell’s approachability framework could be used and actually implemented for solving general convex-concave saddle point problems had been completely overlooked until this paper and we are the first to provide an actual implementation of the CBA framework for various sets of interests other than the simplex (here, decision sets for DRO instances). We introduce CBA+, and its analysis is far from trivial (especially with linear averaging only on decisions) and requires advanced methods from conic optimization. In particular, many of our statements from Lemma A.1 appear to be new in the literature (e.g., Statements 4-5-6 in Lemma A.1). We also highlight the practical efficacy of our algorithm on several domains. Our numerical study shows that CBA+ enjoys strong empirical performances and outperforms prior approaches, even in settings where the other algorithms are granted very aggressive step sizes and may not converge. In contrast, CBA+ does not require *any* tuning to outperform the other algorithms. Additionally, the running time of CBA+ is vastly better than the other algorithms for DRO, mostly because each iteration of CBA+ is way faster than the iterations of OMD, FTRL, etc (see Lines 322-333 in our manuscript). We hope that this short summary of our contributions addresses your concerns about the intuition behind the performances of CBA+ and the novelty of our framework.
>
>
> * References
>
> [1] Oskari Tammelin, Neil Burch, Michael Johanson, and Michael Bowling. Solving heads-up limit Texas hold'em. In Twenty-Fourth International Joint Conference on Artificial Intelligence, 2015.
>
> [2] Gabriele Farina, Christian Kroer, and Tuomas Sandholm. Online convex optimization for sequential decision processes and extensive-form games. In Proceedings of the AAAI Conference on Artificial Intelligence, volume 33, pages 1917{1925, 2019.
>
> [3] Gabriele Farina, Christian Kroer, and Tuomas Sandholm. Regret circuits: Composability of regret minimizers. In International Conference on Machine Learning, pages 1863-1872, 2019.
>
> [4] Sergiu Hart and Andreu Mas-Colell. A simple adaptive procedure leading to correlated equilibrium. Econometrica, 68(5):1127{1150, 2000.
>
> [5] Matej Moravck, Martin Schmid, Neil Burch, Viliam Lisy, Dustin Morrill, Nolan Bard, Trevor Davis, Kevin Waugh, Michael Johanson, and Michael Bowling. Deepstack:Expert-level articial intelligence in heads-up no-limit poker. Science, 356(6337):508-513, 2017.
>
> [6] Noam Brown and Tuomas Sandholm. Superhuman AI for heads-up no-limit poker: Libratus beats top professionals. Science, 359(6374):418-424, 2018.
>
> [7] Gabriele Farina, Christian Kroer, and Tuomas Sandholm. Optimistic regret minimization for extensive-form games via dilated distance-generating functions. In Advances in Neural Information Processing Systems, pages 5222-5232, 2019.
>
> [8] Ben-Tal, Nemirovsky. Lectures on modern convex optimization.
>
> [9] Yuan Gao, Christian Kroer, and Donald Goldfarb. Increasing iterate averaging for solving saddle-point problems. AAAI 2021.
>
> [10] Lower complexity bounds of FOMs for convex-concave bilinear saddle-point problems, Ouyang and Xu (2021).

---

> ### Comment · Reviewer_fhFd · 2021-08-17
> **Response to Authors**
>
> Thanks for the answers. I believe that the Blackwell method has greater potential due to its tuning-free nature and good practical performance. The intuition on the projection of CBA+ is convincing and important for the algorithm design.
>
> My concern is that the answer still does not fully explain why CBA+ can achieve a better numerical performance yet with a slower theoretical convergence rate than OOMD and etc:
>  * How does the $O(\frac{(q+1)\kappa L}{\sqrt{T}})$ regret bounds of CBA+ compare with the regret bounds (also including constants) achieved by other methods like OOMD, and etc.
>  * Can the author elaborate on under what conditions OOMD, O-FTRL would perform better? For instance, what if one is pursuing a high accuracy solution?

---

> > ### Author Response · Authors · 2021-08-22
> > **Second round of response to Reviewer fhFd**
> >
> > We thank you for your quick and detailed response. Below are our responses to your new comments. We summarize your two new comments in italic.
> >
> > * *How does the* $O \left( \frac{(q+1)\kappa L}{\sqrt{T}} \right)$ *regret bounds of CBA+ compare with the regret bounds of other first-order methods?*
> >
> > * *Under what conditions would OOMD, O-FTRL perform better? For instance, what if one is pursuing a high-accuracy solution?*
> >
> > We list here the regret bound on the average regret $R_{T}$ for CBA+, OMD, FTRL, OOMD and OFTRL. We remove the term $(q+1)$ as it is specific to the weighting scheme and not to CBA+. Recall that $$\kappa = \max \lbrace \lVert x \rVert_{2} | x \in \mathcal{X} \rbrace,$$ $$\Omega =  \max \lbrace \lVert x -x'\rVert_{2} | x,x' \in \mathcal{X} \rbrace,$$ $$L = \max \lbrace \lVert f_{t} \rVert_{2} | t \geq 0 \rbrace.$$
> >
> > Let us call $R_{T}$ the average regret. For CBA+, we have the following worst-case bound on the average regret $R_{T}$:
> >
> > CBA+: $R_{T} = O \left( \kappa L  / \sqrt{T} \right).$
> >
> > We first compare this bound on $R_{T}$ with the bounds for OMD and FTRL:
> >
> > OMD [1] / FTRL [2]: $R_{T} = O \left( \Omega L / \sqrt{T} \right).$
> >
> > In this case, we just need to compare $\kappa$ and $\Omega$. As we have mentioned, for CBA+ we can always recenter the set $\mathcal{X}$ at $0$, when $\mathcal{X}$ is one of the decision sets that we have introduced in the paper (the simplex, an $\ell_{p}$-ball, or an ellipsoidal confidence region in the simplex). In this case, the *worst-case* regret bound for CBA+ and OMD/FTRL are exactly equivalent, since $\kappa \leq \Omega \leq 2 \kappa$.
> >
> > Let us now compare to OOMD and OFTRL. For these two algorithms, the regret bound is roughly as follows:
> >
> > OOMD [3] / OFTRL [4]: $R_{T} = O \left( \Omega^2 L / T  \right).$
> >
> > Thus, one potential explanation for the performance discrepancy would be the squared dependence on $\Omega$ for the optimistic algorithms, and we should expect the optimistic algorithms to be faster for some high-enough level of precision.
> >
> > However, this is not an entirely satisfying explanation, since that would imply that any of the algorithms achieving $O(\Omega L /\sqrt{T})$ average regret (OMD, FTRL) or $O \left( \kappa L  / \sqrt{T} \right)$ average regret (CBA, and CBA+) should be faster than OOMD/OFTRL for low-precision solutions. Yet,  only CBA+ (and RM+/CFR+ in the game-solving case) leads to vastly better performances than OOMD/OFTRL. It is an open question to explain this discrepancy (even in the less general setting of matrix games and EFGs, where CFR+ also seems "too fast" relative to what the theory predicts).
> >
> > Also, note that in the figures presented in the paper, we show the number of steps on the $x$-axis, and the duality gap in the $y$-axis. This neglects to take into account the time to perform an iteration. As we have highlighted in Lines 322-333, each step of OMD/FTRL/OFTRL is 2-2.5x slower than a step of CBA+, because of the projection operation in OMD/FTRL/OFTRL (see the details on our numerical implementation in Appendix E). For OOMD, each step is 3-4x slower than a step of CBA+, because of the two proximal updates at each step. Thus, in terms of time to return a good-enough solution, CBA+ is vastly faster than OMD/FTRL/OOMD/OFTRL on all the instances that we have tested.
> >
> > We would like to conclude our response by emphasizing the following remarks.
> >
> > 1) All the regret bounds on CBA+, OMD, etc. are *theoretical*, *worst-case* bounds. For most instances (in our case, four real instances and two random instances), there is no reason to believe that the algorithms will have the empirical behaviour of their theoretical worst-case bounds. This has been highlighted in the recent papers by Fabian Pedregosa and Damien Scieur, who study the average convergence rate of gradient descent (GD) as well as momentum and accelerated GD [5,6]. Analyzing the *average* performances of first-order methods such as OMD or CBA+ is a very promising future direction of research, but it is beyond the scope of this paper.
> >
> > 2) In the matrix game and Extensive-Form Games (EFGs) settings, which have received a tremendous amount of attention, leading to the recent milestones in Poker AI [7,8,9], the Blackwell algorithm (which in this case is RM+ instantiated in the CFR+ framework) routinely outperforms both OOMD and OFTRL. Note that CBA+ itself outperforms CFR+ as we show in our numerical experiments of Section 4.1. One possible explanation for these strong empirical performances is that $CBA+$ automatically adjusts to the problem instances because of the rescaling operation in $CHOOSEDECISION_{CBA+}$. In particular, there is no *conservative* choice of step-sizes to be made, because CBA+ and CFR+ are parameter-free.
> >
> > 3) Finally, note that all the past breakthroughs on matrix games and EFGs are based on algorithms with only $O(1/\sqrt{T})$ regret bounds (RM+ and CFR+). We feel that explaining the discrepancy between the theoretical and practical performance of OOMD, OFTRL, CFR+, and CBA+ is beyond the scope of our paper. In particular, there is already no explanation for this discrepancy for the past performances of RM+/CFR+ in the simpler settings of matrix games and EFGs, and if the community had insisted on a theoretical explanation for the superiority of the CFR+/RM+ algorithms over OFTRL/OOMD, then we would not have achieved any of the recent poker advances yet.
> >
> > * **Conclusion.**
> >
> > We thank you again for your time and your second response. We will incorporate in the paper the important discussion and comparison of the convergence rates of first-order methods (including the constant). This will enhance the readability of the paper. We will highlight the discrepancy between the $O(1/\sqrt{T})$ theoretical regret bounds and the strong empirical performances of CBA+. We hope that our new responses convince you of the completeness and significance of our paper and lead to an increased score.
> >
> > * **References.**
> >
> > [1] https://parameterfree.com/2019/10/01/online-mirror-descent-ii-regret-and-mirror-version/
> >
> > [2] https://parameterfree.com/2019/10/08/follow-the-regularized-leader-i-regret-equality/
> >
> > [3] Chiang, C. K., Yang, T., Lee, C. J., Mahdavi, M., Lu, C. J., Jin, R., & Zhu, S. (2012, June). Online optimization with gradual variations. In Conference on Learning Theory (pp. 6-1). JMLR Workshop and Conference Proceedings.
> >
> > [4] https://parameterfree.com/2019/11/07/optimistic-follow-the-regularized-leader/
> >
> > [5] Pedregosa, F., & Scieur, D. (2020). Average-case acceleration through spectral density estimation. ICML 2020.
> >
> > [6] Scieur, D., & Pedregosa, F. (2020). Universal average-case optimality of polyak momentum. ICML 2020.
> >
> > [7] Oskari Tammelin, Neil Burch, Michael Johanson, and Michael Bowling. Solving heads-up limit Texas hold'em. In Twenty-Fourth International Joint Conference on Artificial Intelligence, 2015.
> >
> > [8] Matej Moravck, Martin Schmid, Neil Burch, Viliam Lisy, Dustin Morrill, Nolan Bard, Trevor Davis, Kevin Waugh, Michael Johanson, and Michael Bowling. Deepstack:Expert-level articial intelligence in heads-up no-limit poker. Science, 356(6337):508-513, 2017.
> >
> > [9] Noam Brown and Tuomas Sandholm. Superhuman AI for heads-up no-limit poker: Libratus beats top professionals. Science, 359(6374):418-424, 2018.

---

> > > ### Comment · Reviewer_fhFd · 2021-09-01
> > > **Appreciate authors' response**
> > >
> > > I would like to thank the authors for the detailed explanation. As the response does not give a convincing answer to such discrepancy, I would remain my score.  Overall, I believe blackwell approachabity has great potential for new algorithm design and worst/avetage case analysis would be valuable.

---

### Official Review · Reviewer_oLRi · 2021-07-16

**Rating:** 4
**Confidence:** 4

**Summary:**

This manuscript provides a parameter-free algorithm for solving convex-concave saddle point problems. The main ingredient is that the authors propose an a Conic Blackwell Algorithm^+ (CBA^+) which is able to recover the standard $O(1/\sqrt{T})$ rate but does not require any stepsize choices. Several use cases are presented. Experiments include matrix games on a simplex and distributional robust optimization.

**Limitations And Societal Impact:**

Yes.

**Main Review:**

The proposed algorithm looks original for minimax problems, but I do not think it is of significance in the literature of solving convex-concave saddle point problems.

(i) I am not convinced by the motivation of this work. The authors mentioned in line 37--line 41 that choosing the learning rate requires to know number of periods $T$ and the upper bound $L$ on the norms of the gradients and they may be hard to obtain. I disagree with this argument. The reason is that for convex-concave saddle point problems, the parameters for both min and max variables lie in convex and compact sets with known size, so it is usually easy to compute the upper bound of the gradient $L$ based on the size of the domains. The dependency on $T$ is also avoidable, since one can use time varying learning rate such as $O(1/\sqrt{t})$ to get the same bound. From my viewpoint, the gradient norm upper bound is easy to compute at least for examples included in Section 4.

(ii) The authors should include the exact constants in big O notation (e.g., in Theorem 3.1, Theorem 3.2) and compare it with the standard algorithms (e.g., OMD, FTRL, etc.), especially the dependency in terms of domain size and the upper bound of gradient norm. It is unclear both from the main text and also the supplement.

(iii) The algorithm design and the corresponding theoretical proofs heavily rely on [Abernethy, Bartlett, Hazan, 2011]. For the algorithm design, the main differences come from the projection steps in line 6 and line 9 in Algorithm 1. For theoretical analysis, The authors should have made it clear what is the purpose of modifying these projection steps and the technical innovation in the proofs.

Experiments:

(i) In Figure 1, why only two methods are compared (e.g., CBA+, CFR+)? What about OMD with best-tuned learning rate? Also the advantage of CBA+ over CFR+ is not significant.

(ii) In section 4.2, how is $\eta_{th}$ chosen?

(iii) In Figure 2, CBA+ still cannot outperform well-tuned OFTRL.


=====POST REBUTTAL=====
Thanks for the authors' response. I still think the presentation of this manuscript needs to be improved, which may needs a major revision. The exact advantage of the regret bound of the proposed algorithm compared with other algorithm (e.g., FTRL, OMD) is not clear. It is very subtle since it is an improvement in terms of constant. I would suggest the authors to have a more precise argument about why the traditional method is inferior than the proposed methods in theory. Due to these reasons, I decide not to change my score.


**Time Spent Reviewing:**

2.5 hours

---

> ### Author Response · Authors · 2021-08-09
> **Response to Reviewer oLRi**
>
> We would like to thank you for your detailed comments. We address them below.
>
> Main reviews:
>
> (i) *Step sizes with $T$ and $L$*.
>
>  We would like to emphasize that even though the upper bound $L$ can be obtained for some min-max instances (e.g., the ones presented in our numerical experiments, where we compute the constant $L_{x}$ and $L_{y}$ in Appendix E.2), this upper bound may be *very* conservative and may lead to overly conservative step sizes. In this case, classical algorithms like OMD, FTRL, etc. may converge very slowly, as highlighted in our numerical experiments of Figure 2 (and Figure 8 in Appendix 8). In this case, one can try to fine-tune the algorithms by multiplying the conservative step sizes by $\alpha$ where $\alpha \geq 1$. However, on larger instances, it is not possible to explore for the best step sizes (like we did in our numerical experiments), as this requires running the algorithms many times and this is time- and resource-consuming. Additionally, this is often practically infeasible for very large settings, since we won't know if the step size will cause a divergence until late in the optimization process. Our framework avoids exactly this waste of resources and achieves strong empirical performances.
>
> As for the dependency in $T$, you are right that the same bound on the regret can be achieved using $1/\sqrt{t}$ instead of $1/\sqrt{T}$ in the step sizes. Note that our algorithm (CBA+) still numerically outperforms OMD, FTRL, and optimistic variants when we use this choice of step sizes (Figure 3 and Figure 9), on all of our six problem instances. We would like to emphasize that the x-axis in Figures 2-3 and Figures 8-9 are *number of iterations* and not the running times. In Lines 322-333, we have clarified that CBA+ is actually at least twice as fast as the other algorithms for DRO, mainly because of the projection steps (see Appendix E.2 for our implementation of OMD, FTRL, etc.).
>
> (ii) *Constant term in the regret*.
>
>  Thanks for mentioning this. We will add the result and our analysis in the main body.
>
> Let us write $L$ the upper bound on the gradient norm. Recall that $\kappa = \max \lbrace \lVert x \rVert | x \in X \rbrace.$ For choices of weights $\omega_{t} = t^{q}$, it is possible to obtain that for CBA and CBA+ we have
>     $$ R_{T} = O \left( \frac{(q+1) \kappa L }{\sqrt{T}} \right).$$
>     This is by combining line 537 (linking $R_{T}$ and $d(u_{T},C^{o})$) and line 543 (bound on $d(u_{T},C^{o})$). We can compare the $\kappa L$ part with the $\Omega L$ part of OMD, where $\Omega = \max \lbrace \lVert x \rVert - \lVert x’ \rVert | x,x’ \in X\rbrace$ (e.g, Equation 5.3.25 in [2]). Note that while $\kappa$ is not shift invariant, we can simply shift the set $X$ to recenter it at $0$.
>
> Finally, the choice of increasing weights $\omega_{t} = t^{q}$ adds a small constant $(q+1)$ to the regret bound. The same situation holds for OMD and other algorithms with increasing weights, see [3] for instance. Also, for RM+ with linear averaging on decisions only, a factor of 2 (which corresponds to q+1 when q=1) also appears in the regret bound (e.g., Theorem 3 in [5]).
>
> (iii) *Projection step and difference with [1]*.
>     We would like to emphasize 1) the differences between Algorithm 1 (CBA+ in our paper) and the algorithm introduced in [1] (CBA in our paper), 2) the intuition behind the projection step and the novelty of our proofs.
>     For 1), while we rely on [1] for the conic hull trick, we not only introduce the projection step for UPDATEPAYOFF$_{CBA+}$, but we also introduce linear averaging (on both payoffs and decisions, or only on decisions for CBA+). This results in vastly improving numerical performances, as highlighted in our numerical experiments. Note that [1] does not mention anywhere the potential use of weights nor their impact on the convergence rate and does not provide numerical experiments.
>     We will make this clearer in the revised version of the paper.
>
> For 2), we will add some intuition on the projection step. It is possible to obtain intuition in two ways: one geometric, one more game-theoretic.
>
> -	Geometric: It is easier to visualize things in $R^{2}$ with $C = R^2_+$ and $C^{o} = R^{2}_{-}$. The projection on $C$ then moves the payoff along the edges of $C^{o}$, maintaining the distance to $C^{o}$ and moving toward the vector $0$. Therefore, the projection step is maintaining the distance to $C^o$. Unfortunately, we can not add a picture in OpenReview but we will add a picture in $R^{2}$ in the revised version.
> -	Game-theoretic: The projection on $R^2_+$ eliminates the components of the payoffs that are negative; it enables CBA+ to be less *pessimistic* than CBA, which may accumulate negative payoffs on actions for a long time, leading to some actions being chosen less frequently. Somehow, the no-regret guarantee for CBA+ is valid starting from any period of *reset* of a component to $0$, i.e., when one component of the aggregated payoff is negative and the projection onto $R^2_+$ sets it to $0$. This is not the case for CBA, which never reset the component of the aggregated payoff to $0$.
>
>
> In terms of proofs, note that [1] entirely relies on Blackwell's original proof [4]. In contrast, our proofs are new and rely on statements from conic optimization (e.g., Statements 4-5-6 in Lemma A.1). Because they rely on somewhat more advanced tools, our proofs are slightly less complex than the corresponding results for RM+ [6].
>
>
> * Experiments:
>
>
> (i) CFR+ vastly outperforms other first-order methods (including OMD, FTRL, mirror prox, and other methods) on matrix games and extensive-form games (EFGs). This is well-documented in the literature on EFGs [5,6,7]. This is why we have chosen to only compare to CFR+ here. Note that the main goal of our paper is to show that Blackwell's framework is general enough to solve a significant class of convex-concave saddle-point problems, and not only bilinear games over simplexes (or treeplexes for EFGs). Therefore, the fact that CBA+ improves upon CFR+ only serves to reassure us that on classical settings (e.g., matrix games and EFGs), CBA+ performs at least as good as the best algorithms known for solving these instances.
>
> (ii) This is detailed in Appendix E.2, after line 767 (paragraph *Computing the theoretical step sizes*.). Thanks for mentioning this, we will add a reference to this appendix in the main body.
>
> (iii) *Figure 2: CBA+ still cannot outperform well-tuned OFTRL.*
>     This is not quite true. Please let us detail our answer below.
>
> First, note that the x-axis on the figures that we are showing here is the number of steps, and not the actual running time of the algorithms. In particular, we would like to refer here to our paragraph on *Running times compared to CBA$^+$*, lines 322-333. CBA+ is at least twice as fast as OMD, FTRL, and O-FTRL, because its projection step is quasi closed-form, whereas, for the other algorithms, a binary search is required (see Appendix E.2 for details of our implementation). In terms of computation time, CBA+ actually vastly outperforms any of the other algorithms that we have tested in our papers.
>
> Second, even if we were only considering the number of steps (and not taking into account the time spent at computing each iteration of the algorithms), OFTRL only outperforms CBA+ on one plot in Figure 2 (*madelon* data set, $\alpha=10,000$), whereas we have made comparisons on six problem instances, and have tried our best to be extensive in our choices of step sizes for the other algorithms.
>
> Third, fine-tuning the algorithms like OMD, FTRL, etc. requires time and resources; many values of $\alpha$ have to be tried, where the step sizes are $\eta = \alpha \times \eta_{\sf th}$. It may not be possible for large real-world instances, and it may be seen as a waste of time and resources. Additionally, and critically, when $\alpha=10,000$, there is no convergence guarantee for O-FTRL with the choice of fixed step sizes $\eta = \alpha \times \eta_{\sf th}$, as in this case $\eta$ is way larger than the theoretical fixed sizes $\eta_{\sf th}$ that guarantee sublinear regret. For instance, this is why O-OMD diverges on the {\em splice} and {\em normal} data sets, when $\alpha=10,000$. In contrast, CBA+ can be used without wasting time on exploring and finding the best convergence rates, and with confidence in the convergence of the algorithms. Our extensive numerical experiments for six different instances of DRO problems (Figures 2-3 in Section 4, Figures 8-9 in Appendix F) show that it is hard to ``fine-tune'' the other algorithms to achieve similar strong performances that CBA+ achieves without any tuning.
>
>
> We thank you again for your time reviewing the paper and we hope that we have addressed your concerns.
>
> * References
>
> [1] J Abernethy, PL Bartlett, and E Hazan. Blackwell approachability and no-regret learning are equivalent. In Proceedings of the 24th Annual Conference on Learning Theory, pages 27-46. JMLR Workshop and Conference Proceedings, 2011.
>
> [2] Ben-Tal, Nemirovsky. Lectures on modern convex optimization.
>
> [3] Yuan Gao, Christian Kroer, and Donald Goldfarb. Increasing iterate averaging for solving saddle-point problems. AAAI 2021.
>
> [4] David Blackwell. An analog of the minimax theorem for vector payoffs. Pacific Journal of Mathematics, 6(1):1–8, 1956.
>
> [5] Oskari Tammelin, Neil Burch, Michael Johanson, and Michael Bowling. Solving heads-up limit Texas hold'em. In Twenty-Fourth International Joint Conference on Artificial Intelligence, 2015.
>
> [6] Gabriele Farina, Christian Kroer, and Tuomas Sandholm. Online convex optimization for sequential decision processes and extensive-form games. In Proceedings of the AAAI Conference on Artificial Intelligence, volume 33, pages 1917{1925, 2019.
>
> [7] Gabriele Farina, Christian Kroer, and Tuomas Sandholm. Regret circuits: Composability of regret minimizers. In International Conference on Machine Learning, pages 1863-1872, 2019.

---

> ### Author Response · Authors · 2021-09-01
> **Response to comments post rebuttal**
>
> We thank you for your response to our comments.
>
> The exact advantages of our algorithm CBA+ compared to prior approaches are its simplicity and its very strong empirical performances for various important domains and real instances. Focusing only on the theoretical regret bound fails to understand the main point of the paper: that our method achieves these bounds while simultaneously leading to a highly performant method for solving saddle-point problems in practice. From a philosophy of science perspective, we feel that the practical, empirical performances of an algorithm are as important as its theoretical properties. We would also like to point out that if the rest of the field had adopted the perspective of focusing solely on theoretical rates, then we would not have achieved any of the recent impressive results on solving large-scale games. These are all based on algorithms with only $O(1/\sqrt{T})$ regret bounds (RM+ and CFR+). We feel that if the community had insisted on a theoretical explanation for the empirical superiority of CFR+/RM+ over classical methods (OFTRL/OOMD/OMD/FTRL), then we would not have achieved any of the recent poker advances yet. Indeed, we still do not have a theoretical explanation for the superiority of CFR+ over these other methods.
>
> We thank you again for your update and for your time reviewing the paper.

---

### Official Review · Reviewer_BS7v · 2021-07-16

**Rating:** 7
**Confidence:** 3

**Summary:**

# Summary

In light of successful applications of regret matching (RM) -- an instance of Blackwell approachability -- and its variant regret matching+ (RM+) for saddle point optimization  in zero-sum matrix games with simplex decision sets, the paper proposes generalizations beyond this simple setting to address convex concave problems with compact decisions sets. In the way that RM+ improves RM the conic Blackwell algorithm+ is introduced to improve over the approach of conic blackwell approachability (CBA) from [1]. Furthermore, CBA and CBA+ require non trivial Euclidean projections, in this work explicit closed form or efficient approximations are introduced. CBA and CBA+ are extensively tested on a variety of domains, for which CBA+ seems to demonstrate improvements over CBA similar to how RM+ improves RM. Importantly, like RM and RM+, CBA and CBA+ are parameter-free requiring no tuning.

**Ethics Review Area:**

["I don’t know"]

**Limitations And Societal Impact:**

Limitations are discussed throughout the paper and in the conclusions. Potential negative societal impact is also addressed.

**Main Review:**

# Main Review

## Summary of Review

Overall, the  paper makes for a good contribution to the area of no-regret learning and convex/ concave constrained saddle point optimization. However, improvements can be made with respect to connections with related work, including experiments more closely aligned with the theory, and improving clarity regarding connections to CFR and methods for solving extensive-form games (EFGs).



## Originality and Connections with Previous Work

The approach on regret minimization (or online linear optimization) being reduced to an approachability game for convex sets actually predates the construction from [1] referenced in the paper. In [2] the method of Lagrangian hedging (LH) explicitly does this, in fact it is more general, the decision set $\mathcal{X}$ need not be a cone (for example the simplex is not a cone) but they also consider the trick of embedding the set $\mathcal{X}$ in a higher dimensional space and taking the conic hull. This flexibility allows for LH to exactly recover RM on the simplex, whereas it is unclear whether the construction in [1] and hence the CBA algorithm recovers RM when applied on the simplex. Provided that CBA and CBA+ are strongly influenced by RM and RM+ it would be useful to mention whether or not CBA can recover RM.

### On the originality of CBA+
It is likely CBA+ is novel however it might be strongly related to the OGD based approachability algorithm of [3] if not the same. In [3] however, they do not need to apply the conic hull trick of [1] and do a projection step similar to CBA+. Some discussion is needed to compare the two algorigthms.
 In fact [3] shows an explicit connection with follow the regularized leader and Blackwell's algorithm which is equivalent to CBA if one applies the conic hull trick (this connection was also rediscovered in [4]), it would be nice to include a discussion on how CBA fits in the closely related work of LH [2], [3] and [4].

## Quality
Overall the paper is well written except for some clarity issues discussed below. However, I have one issue regarding the experiments. It seems that in all cases CBA+ is used with alternation while there is no theory to support its convergence. I do not think it is necessary to have such a result in this paper but there should be at least some experiments in the main body that are consistent with the theory. What happens if alternation is not used with CBA+ is all the performance lost? For example I think it would be useful to see to RM+, CBA+, and other algorithms without alternation. Also it would be nice to see the explicit benefit of having the  averaging of Theorem 3.1 (again without alternation). How much performance is lost without the linear averaging?

## Clarity
The paper is mostly clear except for references to sequential games and EFGs.  For example statements like line 231 "Simplex  $\Delta(n)$ is the classical setting used for matrix games and extensive-form games." This is very confusing since CBA and CBA+ is often compared to CFR and CFR+, indeed there are experiments in the appendix that make such comparisons. This is confusing because CFR minmizes regret with respect to sequence form which in general is not a simplex (a treeplex instead [5]), however, there seems to be no discussion on implementing CBA or CBA+ on the sequence form. One can model and EFGs using the strategic form and hence over the simplex but this is exponential in the number of game states. Is CBA or CBA+ operating over the sequence form or the classical strategic form? More clarification is needed, and as it stands it in unclear how CBA or CBA+ is used to solve EFGs.

Furthermore, if CBA or CBA+ operates over the sequence form then there needs to be discussion on how it compares with the LH algorithms of [3] (see tech report for more details) a regret matching like algorithm is  devised to work over the sequence form and it is also parameter free approachability strategy designed to minmize regret and hence solve EFGs. I would expect some connection with LH with the Euclidean norm to be closely related to CBA, tho I believe a CBA+ like algorithm that works on the sequence is completely novel.

# Significance


# Other suggested improvements

Below I list some suggested improvements:
1. I believe your framework and results apply to the case when $F$ is not even differentiable but subdifferentiable, this would improve your results as other stated competitors such as mirror-prox cannot achieve a $O(1/T)$ rate in this setting.
2. Related to the previous point, there is a missing assumption regarding the $O(1/T)$ rate of mirror prox and optimistic methods, the game needs to be smooth in the Lipschitz gradient sense, see for example section 5.2.3 in [6].
3. Regarding the $O(1/T)$ rate in smooth games of optimistic FTRL and OMD  please include a citation to [7]
4. On the topic or rates, it should be mentioned that in matrix games it is possible to achieve linear convergence (a.k.a exponential convergence) with a first order  algorithm which is as computationally expensive as mirror prox see the MAIO algorithm in [10].
5. Regarding statements on the stepsize selection of FTRL and / or OMD, it is not true that they require "knowing a bound L on the norm of the instantaneous payoffs" it suffices to select $\eta_t \in O(\frac{1}{\sqrt{t}})$ to achieve an anytime regret bound of $O(\sqrt{T})$ see for example [8] or [9].


# References

1. Jacob Abernethy, Peter L Bartlett, and Elad Hazan. Blackwell approachability and no-regret
learning are equivalent. In Proceedings of the 24th Annual Conference on Learning Theory,
pages 27–46. JMLR Workshop and Conference Proceedings, 2011.

2. Gordon, G.J., 2007. No-regret algorithms for online convex programs. In Advances in Neural Information Processing Systems (pp. 489-496). (for tech report see [here](https://www.cs.cmu.edu/~ggordon/ggordon.CMU-CALD-05-112.no-regret.pdf))

3. Shimkin, N., 2016. An online convex optimization approach to Blackwell's approachability. The Journal of Machine Learning Research, 17(1), pp.4434-4456.

4. Gabriele Farina, Christian Kroer, and Tuomas Sandholm. Faster game solving via predictive
blackwell approachability: Connecting regret matching and mirror descent. In Proceedings of
the AAAI Conference on Artificial Intelligence. AAAI, 2021.

5. Samid Hoda, Andrew Gilpin, Javier Pena, and Tuomas Sandholm. “Smooth-ing techniques for computing Nash equilibria of sequential games.” In:Mathematics of Operations Research35.2 (2010), pp. 494–512.

6. Bubeck, S., 2014. Convex optimization: Algorithms and complexity. arXiv preprint arXiv:1405.4980.

7. Syrgkanis, V., Agarwal, A., Luo, H. and Schapire, R.E., 2015. Fast Convergence of Regularized Learning in Games. Advances in Neural Information Processing Systems, 28, pp.2989-2997.

8. Xiao, L., 2010. Dual Averaging Methods for Regularized Stochastic Learning and Online Optimization. Journal of Machine Learning Research, 11(88), pp.2543-2596.

9. Orabona, F., 2019. A modern introduction to online learning. arXiv preprint arXiv:1912.13213.

10. Munos, R., Perolat, J., Lespiau, J.B., Rowland, M., De Vylder, B., Lanctot, M., Timbers, F., Hennes, D., Omidshafiei, S., Gruslys, A. and Azar, M.G., 2020, November. Fast Computation of Nash Equilibria in Imperfect Information Games. In International Conference on Machine Learning (pp. 7119-7129). PMLR.


**Time Spent Reviewing:**

10

---

> ### Author Response · Authors · 2021-08-09
> **Response to Reviewer BS7v**
>
> We thank you for the time spent reviewing our paper. We appreciate your detailed, accurate, and helpful comments. We present our response below.
>
> * Originality of our work.
>
>  Thanks for mentioning the results on Lagrangian Hedging (LH) from [2], we will appropriately reference it in our revised version. Note that the RM decision can be seen as an approximate version of the $CHOOSEDECISION_{CBA}$. In particular, the $CHOOSEDECISION_{CBA}$ function requires computing a projection onto the cone $C$ (see line 196 of our manuscript). If a certain approximate solution of this projection problem is used, then the same decision as RM is chosen. This is explained in lines 628-630 of our manuscript (Appendix C.1). We will make this more explicit in the main body of the paper.
>
> * On the originality of CBA+.
>
> Thanks for this reference. We agree that CBA+ is related to the OGD-based approach of [3].
> That said, there are some differences beyond the conic hull trick from [1].
> First, [3] explicitly focuses on the probability simplex as the decision space, whereas we are interested in more general convex compact decision sets (we think that OGD approach from [3] should be generalizable to convex compact sets, and will add a discussion of that).
> Secondly, we show that CBA+ works with linear averaging, which is crucial from a practical perspective (it's likely that one could show the same for the OGD approach from [3]).
> We will add a careful discussion to the paper of all these connections and differences.
>
> Going back to both [2] and [3], we will also add a discussion about the convex hull trick of [1] and what the resulting difference is, and its pros and cons.
>
> * Quality.
>
>  We will add experiments to compare the individual benefits of alternation and the individual benefits of linear averaging for CBA+. If alternation is not used on any algorithm, CBA+ still outperforms CBA and RM but it is outperformed by RM+. In the setting without alternation, linear averaging greatly improves the numerical performances of CBA+ (respectively, RM+) compared to CBA+ without linear averaging (respectively, RM+ without linear averaging). We will add some more detailed experiments on this in our revised manuscript; thanks for suggesting this improvement.
>
> * Clarity.
>
>  Thanks for mentioning these clarity issues in our paper. Indeed, CFR/CFR+ minimizes regret over treeplexes, but CFR decomposes the treeplex problem into a set of simplex regret minimization problems; this is what we meant by the simplex being an important regret minimization problem even for EFGs.
> For EFGs, we used *CBA* and *CBA+* as simplex regret minimizers, as part of the CFR regret decomposition. This lets us compare CBA and CBA+ directly to RM and RM+ for EFG solving.
> We will make this clearer.
>
> As evident from the response above, CBA and CBA+ do not operate over the sequence form in our current setup, though that is a very interesting future direction. In particular, all our theory works for that problem (since it's a compact convex set), except that we now come back to the question of finding an efficient projection operation in order to run the algorithm. Whether such a projection exists is something we will investigate in future work, but it may not be a trivial question.
>
> * Other suggested improvements.
>
> Thanks for your suggestion. We comment on these potential improvements below.
>
> 1) *Sub-differentiable function F.* You are correct, thanks for mentioning this. We will update the paper to work for this more general setting.
>
> 2) *Assumptions of optimistic methods.* Thanks, we will add this assumption.
>
>  3) *Missing citation.* We will add this, thanks.
>
> 4) *Linear convergence.*
>
>     First of all, we would like to point out that these linear rate results have been rediscovered many times. The earliest such result that we know of which would apply to matrix games and EFG solving is Paul Tseng's paper from 1995 [4], but more generally these ideas go back to Hoffman's seminal result [5], see e.g. Karimi et al. for a recent discussion [6].
>
>     They are thus a well-known phenomenon, but they are typically treated as a distinct form of guarantee/a different analysis regime. This is because they rely on unknown constants that can be extremely large (all these results only show the existence of a constant, but not bounds on it). In fact, getting a handle on these constants, which are sometimes known as condition numbers, is a research area of its own, see e.g. [7].
>
>     Numerically, our experience is that this exponential rate is often hard to see, potentially due to it kicking in late in the optimization process, when very close to the solution.
>
>     The specific paper you reference does show an example of such a result, but it is a setting rather different from mirror prox. Note from Theorem 2 of [8] that they specifically need the $L_2$ regularizer in order for the existence of the Hoffman-type constant to exist. Therefore, you will need projection onto the simplex at every iteration in order to invoke their result (whereas mirror prox can use the entropy regularizer).
>
>     More generally, their result in Lemma 1 (which is what enables the exponential rate) is actually a special case of Tseng's 1995 result [4], which showed that this will work when the gradient mapping $F$ of the variational inequality description of a saddle-point problem is affine, and the decision set is polyhedral. Tseng also showed that this works for the extragradient method, which is equivalent to mirror prox with $L_2$ as the regularizer.
>     Tseng's paper thus shows that, actually, even for EFGs it's possible to get a linear rate (such a rate was also rediscovered in [9]), again as long as we are fine using accelerated first-order methods *with the $L_2$ regularizer*.
>
>     We will mention the existence of these linear-rate results for extragradient-type methods (as well as the recent results on linear-rate guarantees on optimistic OGD and optimistic MWU [10]), with what we view as the appropriate caveats above.
>
>   5) *Step size for FTRL/OMD.* Thanks for pointing at this inaccuracy. We were referring to the theoretical constant step sizes. We will add a comment that $\eta_{t} = O (1/\sqrt{t})$ is sufficient. Note that the difficulty is to choose the constant $\alpha > 0$ for which $\eta_{t} = \alpha/\sqrt{t}$. This may require trying various values of $\alpha$, which is time- and resource-consuming (and often practically infeasible for very large games, since we won't know if the stepsize will cause a divergence until late in the optimization process). Our framework avoids exactly this waste of resources and achieves strong empirical performances.
>
> We thank you again for your detailed responses that will lead to a stronger revised manuscript. We hope that we have addressed your concerns.
>
> * References
>
> [1] Jacob Abernethy, Peter L Bartlett, and Elad Hazan. Blackwell approachability and no-regret learning are equivalent. In Proceedings of the 24th Annual Conference on Learning Theory, pages 27–46. JMLR Workshop and Conference Proceedings, 2011.
>
> [2] Gordon, G.J., 2007. No-regret algorithms for online convex programs. In Advances in Neural Information Processing Systems (pp. 489-496).
>
> [3] Shimkin, N., 2016. An online convex optimization approach to Blackwell's approachability. The Journal of Machine Learning Research, 17(1), pp.4434-4456.
>
> [4] Paul Tseng. On linear convergence of iterative methods for the variational inequality problem. Journal of Computational and Applied Mathematics, 60(1-2):237-252, 1995.
>
> [5] A.J Ho
> man and R.M. Karp. On nonterminating stochastic games. Journal of the Institute of Management Science. Application and Theory Series, 12:359-370, 1966.
>
> [6] Hamed Karimi, Julie Nutini, and Mark Schmidt. Linear convergence of gradient and proximal-gradient methods under the polyak- lojasiewicz condition. In Joint European Conference on Machine Learning and Knowledge Discovery in Databases, pages 795-811. Springer, 2016.
>
> [7] Boris S Mordukhovich, Javier F Pena, and Vera Roshchina. Applying metric regularity to compute a condition measure of a smoothing algorithm for matrix games. SIAM Journal on Optimization, 20(6):3490-3511, 2010.
>
> [8] Munos, R., Perolat, J., Lespiau, J.B., Rowland, M., De Vylder, B., Lanctot, M., Timbers, F., Hennes, D., Omidshafiei, S., Gruslys, A. and Azar, M.G., 2020, November. Fast Computation of Nash Equilibria in Imperfect Information Games. In International Conference on Machine Learning (pp. 7119-7129). PMLR.
>
> [9] Andrew Gilpin, Javier Pena, and Tuomas Sandholm. First-order algorithm with o(ln($1/\epsilon$)) convergence for $\epsilon$-equilibrium in two-person zero-sum games. Mathematical programming, 133(1):279-298, 2012.
>
> [10] Chung-Wei Lee, Haipeng Luo, Chen-Yu Wei, and Mengxiao Zhang. Linear last-iterate convergence for matrix games and stochastic games. arXiv e-prints.

---

> > ### Comment · Reviewer_BS7v · 2021-08-25
> > **Response to Authors**
> >
> > Thank you for the detailed rebuttal. I acknowledge that the authors have answered all of my concerns, I have a few comments for some of the responses which I discuss below but overall am satisfied with the author's response. After giving some thought about the work I have realized there is some missing related work that should be mentioned and discussed, namely the work of Francesco Orabona and Dàvid Pàl [101] and possibly others, I go into detail below and look forward to the author's response.
> >
> >  ## RM as approximate CBA
> > Thank you for the clarification, I believe that viewing RM as CBA with an approximate $CHOOSEDECISION_{CBA}$ function is a little forced but I think this stems from the limitations of the approach in [1] and not this paper. It is my understanding that under approximations of $CHOOSEDECISION_{CBA}$ the paper presents no guarantees, could you confirm this?
> >
> > ## CBA+ vs OGD in [3]
> > Thank you for the response regarding this issue, I do now see that [3] is limited to games with simplex decision sets and as you say it is probably extendable to the more general case. However, since the main focus of the experiments is with simplex decision sets then it should be clearly stated when CBA+ and the approach of [3] are the same (without linear averaging of course). If we consider regret minimization over the simplex would CBA+ and OGD from [3] be the same? Maybe one needs to lift the target set to a cone before applying the approach of [3]?
> >
> > ## Experiments and Ablation Study
> > Thanks for the response, I have nothing else to add.
> >
> > ## Clarity CBA/CBA+ in EFGs
> > Thank you for the clarification.
> >
> > ## Linear Convergence Related Work
> > I was not aware of all these related works, I thank the authors for mentioning them. I agree that there are some caveats with respect to these rates.
> >
> > # Other Possible Important Related Work
> >
> > I have come to realize that there should also be a discussion regarding the parameter-free and scale-free algorithms from online optimization such as those from [101]. In [101] adaptive no regret algorithms are presented that are scale-invariant to the norms of the losses, and they are able to show meaningful regret guarantees even in unbounded domains. I would expect that the algorithms in [101] to be competitors of CBA and CBA+ regarding being parameter-free and capable of solving saddle point problems (via the folk-theorem). I would still expect say CBA+ to outperform these algorithms, however, I think these algorithms need to be mentioned as alternatives since they also require no tuning. Furthermore, the algorithms under study in [101] include FTRL and it is mentioned by the authors that it is difficult to tune FTRL even with a schedule of stepsizes, I would expect that this might not hold for the algorithms in [101]. For example, some of the algorithms in [101] can be viewed as a generalization of ada-hedge [102] to general decision sets, I would like to know the author's opinion on these algorithms and if the authors believe the following comment in the paper still applies (it might be so):
> >
> > > Even when considering adaptive step sizes, or fixed step sizes that are up to 10,000 larger than those predicted by theory, our CBA+ algorithm performs better.
> >
> > I think a discussion with the adaptive and scale-free approaches from online optimization such as those from [101] are needed and would improve the paper.
> >
> > [1]    Jacob Abernethy, Peter L Bartlett, and Elad Hazan. Blackwell approachability and no-regret learning are equivalent. In Proceedings of the 24th Annual Conference on Learning Theory, pages 27–46. JMLR Workshop and Conference Proceedings, 2011.
> >
> > [2]    Gordon, G.J., 2007. No-regret algorithms for online convex programs. In Advances in Neural Information Processing Systems (pp. 489-496). (for tech report see [here](http://www.cs.cmu.edu/~ggordon/ggordon.CMU-CALD-05-112.no-regret.pdf))
> >
> >  [3]   Shimkin, N., 2016. An online convex optimization approach to Blackwell's approachability. The Journal of Machine Learning Research, 17(1), pp.4434-4456.
> >
> > [101] Orabona, Francesco, and Dávid Pál. "Scale-free algorithms for online linear optimization." International Conference on Algorithmic Learning Theory. Springer, Cham, 2015.
> >
> > [102] De Rooij, Steven, et al. "Follow the leader if you can, hedge if you must." The Journal of Machine Learning Research 15.1 (2014): 1281-1316.

---

> > > ### Author Response · Authors · 2021-08-30
> > > **Second response to Reviewer BS7v**
> > >
> > > We thank you for your quick and detailed response. Below are our responses to your new comments.
> > >
> > > ## *RM as approximate CBA.*
> > >
> > > You are correct that our proofs of the convergence of $CBA$ and $CBA+$ are based on exact computations of $CHOOSEDECISION_{CBA}$ and $CHOOSEDECISION_{CBA+}$. It is a promising next step to investigate the convergence of $CBA$ and $CBA+$ when we use approximate updates, especially since this can save some computation time at each step of the algorithms. This is the main bottleneck in $CBA$ and $CBA+$: even though we show in Section 3.1 that for some important decision sets $\mathcal{X} \in R^{n}$, the exact computation of $CHOOSEDECISION_{CBA}$ can be made in $O(n \log(n))$, an approximate closed-form update would yield a $O(n)$ complexity for each iteration. We leave this for future works.
> > >
> > > ## *CBA+ vs OGD in [3].*
> > >
> > > We would like to clarify that we do not view the simplex domain as the primary setting of interest in our paper. The experiments in our manuscript have a number of simplex settings because we were interested in seeing how well CBA+ captures the performance of RM+ (with linear averaging) in that setting. The main point of the paper, though, is to give a method that then generalizes that performance beyond simplex settings (among others, instances of distributionally robust optimization).
> > > We agree that if you lift the target set to a cone and apply OGD from [3], you probably get the same algorithm for simplex regret minimization. We will make sure to work this out rigorously and we will clarify this in the paper.
> > >
> > > ## *Other Possible Important Related Work.*
> > >
> > > We agree that a comparison to [101] (and similar results) is something that should be added, and we will do so in our revised manuscript. Regarding numerical performance, we expect those algorithms to suffer from the same drawbacks as other variations on adaptive stepsizing for FTRL/OMD for the simplex setting. For instance, in the simplex setting, the practical performance of NormalHedge does not seem competitive with RM+-like algorithms [103].
> > >
> > > More generally, the fact that nobody uses these algorithms in lieu of RM+ in practical large-scale poker solving is also a form of evidence that they are not competitive. That said, it could be different beyond the simplex setting, and we will try to add some additional comparisons (beyond the adaptive stepsizing experiments that we already have in the paper, see Figure 3 and Figure 9).
> > >
> > > ## *Conclusion.*
> > >
> > > We thank you for your detailed review, which greatly helped to improve the paper. We will add in the paper a detailed discussion on the connection between our algorithms and the results in [3]. We will also add a comparison to [101]. We would like to conclude with two comments.
> > >
> > > ### A note on the empirical performances of $CBA+$.
> > >
> > > Regarding the empirical performances of $CBA+$, all figures in our paper show on the $x$-axis the number of steps, and on the $y$-axis the optimality gap. Because the computational effort at each step varies greatly across the algorithms, the $x$-axis does not reflect the actual running time of the algorithms. We would like to refer here to our paragraph on *Running times compared to CBA*, lines 322-333. In terms of time at each iteration, CBA+ is at least twice as fast as OMD, FTRL, and OFTRL, because its projection step is quasi closed-form, whereas, for the other algorithms, a binary search is required (see Appendix E.2 for details of our implementation). It is even $3-4x$ times faster than OOMD, that requires two projections at each iteration. In terms of running time, CBA+ vastly outperforms any of the other algorithms that we have tested in our papers.
> > >
> > > ### Novelty of our work.
> > >
> > > Finally, we would like to emphasize that apart from the $CBA+$ framework (algorithm, efficient projections, instantiation, code available and numerical experiments), one of our main contributions is to highlight that the Blackwell approachability framework can be applied to (very efficiently) solve *general* convex-concave saddle-point problems *in practice*.
> > >
> > > In particular, despite the very good empirical performances of $RM+$ on the simplex in the linear objective setting, we are not aware of an actual instantiation and implementation of this approach for solving other types of problems than matrix games and Extensive-Form Games (EFGs).
> > >
> > > Note that many papers on no-regret algorithms do not provide simulations and codes to highlight the performances of their algorithms. Let us for example focus on the papers that you cite in your last response. The authors in [101] do not provide simulations to show the performances of their algorithms. The situation is similar for [3], and even the original paper [1] on $CBA$ does not provide numerical experiments (it cannot, because it does not explain how to solve the projection problem at every iteration). Similarly, the authors in [102] run simulations on made-up examples of dimension $n=2$.  We do not claim that this diminishes the relevance and significance of these papers. But in contrast with most of the literature, we also provide detailed simulations that highlight the strong speedups achieved by CBA+ compared to classical algorithms, and we emphasize its applicability to various important problems. To the best of our knowledge, we are the first to implement and instantiate the Blackwell approachability framework (through $CBA+$) to solve Distributionally Robust Optimization instances. We hope that our paper can help generalize the use and implementation of the Blackwell approachability framework to other optimization settings than the classical matrix games and EFGs applications.
> > >
> > > ### Final words.
> > >
> > > We thank you again for your time, we enjoyed the discussion. We hope that our new responses convince you of the completeness and significance of our paper and lead to an increased score.
> > >
> > > ## References.
> > >
> > > [1] Jacob Abernethy, Peter L Bartlett, and Elad Hazan. Blackwell approachability and no-regret learning are equivalent. In Proceedings of the 24th Annual Conference on Learning Theory, pages 27–46. JMLR Workshop and Conference Proceedings, 2011.
> > >
> > > [3] Shimkin, N., 2016. An online convex optimization approach to Blackwell's approachability. The Journal of Machine Learning Research, 17(1), pp.4434-4456.
> > >
> > > [101] Orabona, Francesco, and Dávid Pál. "Scale-free algorithms for online linear optimization." International Conference on Algorithmic Learning Theory. Springer, Cham, 2015.
> > >
> > > [102] De Rooij, Steven, et al. "Follow the leader if you can, hedge if you must." The Journal of Machine Learning Research 15.1 (2014): 1281-1316.
> > >
> > > [103] Brown, Noam, and Tuomas Sandholm. "Solving imperfect-information games via discounted regret minimization." Proceedings of the AAAI Conference on Artificial Intelligence. Vol. 33. No. 01. 2019.

---

> > > > ### Comment · Reviewer_BS7v · 2021-08-30
> > > > **Second Response to Authors**
> > > >
> > > > Thank you for your response. I feel that my two remaining concerns from my previous post have been or will be addressed, namely:
> > > >
> > > > 1. A comparison between CBA+ and OGD approach from [3]
> > > > 2. A discussion and experiments comparing the adaptive CBA and CBA+ methods against those in [101]
> > > >
> > > > Finally, I acknowledge that highlighting the effectiveness of RM+ style approaches (e.g CBA+) beyond the simplex setting is of value. For these reasons, I've increased my score from a 6 to a 7.
> > > >
> > > > [3] Shimkin, N., 2016. An online convex optimization approach to Blackwell's approachability. The Journal of Machine Learning Research, 17(1), pp.4434-4456.
> > > >
> > > > [101] Orabona, Francesco, and Dávid Pál. "Scale-free algorithms for online linear optimization." International Conference on Algorithmic Learning Theory. Springer, Cham, 2015.

---

> > > > > ### Author Response · Authors · 2021-08-30
> > > > > **Final response to Reviewer BS7v**
> > > > >
> > > > > We thank you for the time spent reviewing the paper and for the discussion. We will revise the paper accordingly.

---

### Decision · Program_Chairs · 2021-09-27

**Decision:**

Accept (Poster)

**Comment:**

The paper introduces parameter and scale free algorithms for convex concave saddle point problems, based on the Blackwell approachability algorithm. The resulting algorithmic schemes exhibit standard 1/sqrt(T) regret, but in comparison to existing algorithms do not require the knowledge of the problem parameters. The techniques for proving convergence rely on results from conic optimization. Numerical results on real and synthetic data demonstrate the efficacy of the introduced algorithms.

Overall, the reviews appreciated the novelty and the simplicity of the framework. However, there were several issues raised by the reviews relating to relationship with related work and the overall clarity. The authors should take these comments into account when preparing the revised version of the paper.